# TopoTune: a framework for generalized combinatorial complex neural networks

## Abstract

Graph Neural Networks (GNNs) excel in learning from relational datasets, processing node and edge features in a way that preserves the symmetries of the graph domain. However, many complex systems—such as biological or social networks—involve multiway complex interactions that are more naturally represented by higher-order topological domains. The emerging field of Topological Deep Learning (TDL) aims to accommodate and leverage these higher-order structures. Combinatorial Complex Neural Networks (CCNNs), fairly general TDL models, have been shown to be more expressive and better performing than GNNs. However, differently from the graph deep learning ecosystem, TDL lacks a principled and standardized framework for easily defining new architectures, restricting its accessibility and applicability. To address this issue, we introduce *Generalized CCNNs* (GCCNs), a novel simple yet powerful family of TDL models that can be used to systematically transform any (graph) neural network into its TDL counterpart. We prove that GCCNs generalize and subsume CCNNs, while extensive experiments on a diverse class of GCCNs show that these architectures consistently match or outperform CCNNs, often with less model complexity. In an effort to accelerate and democratize TDL, we introduce TopoTune, a lightweight software for defining, building, and training GCCNs with unprecedented flexibility and ease.

## 1 Introduction

Graph Neural Networks (GNNs) (Scarselli et al., 2008; Corso et al., 2024) have demonstrated remarkable performance in several relational learning tasks by incorporating prior knowledge through graph structures (Kipf & Welling, 2017; Zhang & Chen, 2018). However, constrained by the pairwise nature of graphs, GNNs are limited in their ability to capture and model higher-order interactions—crucial in complex systems like particle physics, social interactions, or biological networks (Lambiotte et al., 2019). *Topological Deep Learning* (TDL) (Bodnar, 2023) precisely emerged as a framework that naturally encompasses multi-way relationships, leveraging beyond-graph combinatorial topological domains such as simplicial and cell complexes, or hypergraphs (Papillon et al., 2023).[1]

In this context, Hajij et al. (2023; 2024a) have recently introduced *combinatorial complexes*, fairly general objects that are able to model *arbitrary* higher-order interactions along with a *hierarchical* organization among them–hence generalizing (for learning purposes) most of the combinatorial topological domains within TDL, including graphs. The elements of a combinatorial complex are *cells*, being nodes or groups of nodes, which are categorized by *ranks*. The simplest cell, a single node, has rank zero. Cells of higher ranks define relationships between nodes: rank one cells are edges, rank two cells are faces, and so on. Hajij et al. (2023) also proposes *Combinatorial Complex Neural Networks* (CCNNs), machine learning architectures that leverage the versatility of combinatorial complexes to naturally model higher-order interactions. For instance, consider the task of predicting the solubility of a molecule from its structure. GNNs model molecules as graphs, thus considering atoms (nodes) and bonds (edges) (Gilmer et al., 2017). By contrast, CCNNs model molecules as combinatorial complexes, hence considering atoms (nodes, i.e., cells of rank zero), bonds (edges, i.e., cells of rank one), and also important higher-order structures such as rings or functional groups (i.e., cells of rank two) (Battiloro et al., 2024).

---

[1]Simplicial and cell complexes model *specific* higher-order interactions organized *hierarchically*, while hypergraphs model *arbitrary* higher-order interactions but *without any hierarchy*.

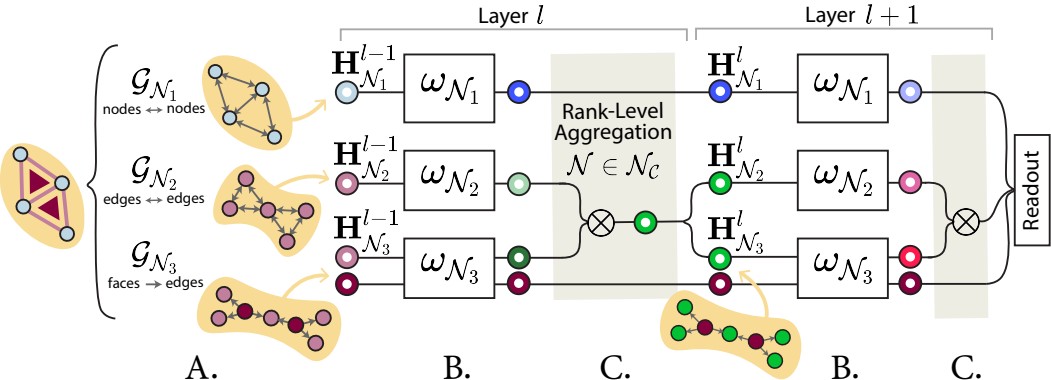

Figure 1: **Generalized Combinatorial Complex Network (GCCN).** The input complex $\mathcal{C}$ has neighborhoods $\mathcal{N}_{\mathcal{C}} = \{\mathcal{N}_1, \mathcal{N}_2, \mathcal{N}_3\}$. **A.** The complex is expanded into three augmented Hasse graphs $\mathcal{G}_{\mathcal{N}_i}$, $i = \{1, 2, 3\}$, each with features $H_{\mathcal{N}_i}$ represented as a colored disc. **B.** A GCCN layer dedicates one base architecture $\omega_{\mathcal{N}_i}$ (GNN, Transformer, MLP, etc.) to each neighborhood. **C.** The output of all the architectures $\omega_{\mathcal{N}_i}$ is aggregated rank-wise, then updated. In this example, only the complex's edge features (originally pink) are aggregated across multiple neighborhoods ($\mathcal{N}_2$ and $\mathcal{N}_3$).

**TDL Research Trend.** To date, research in TDL has largely progressed by taking existing GNNs architectures (convolutional, attentional, message-passing, etc.) and generalizing them one-by-one to a specific TDL counterpart, whether that be on hypergraphs (Feng et al., 2019; Chen et al., 2020a; Yadati, 2020), on simplicial complexes (Roddenberry et al., 2021; Yang & Isufi, 2023; Ebli et al., 2020; Giusti et al., 2022a; Battiloro et al., 2023; Bodnar et al., 2021b; Maggs et al., 2024), on cell complexes (Hajij et al., 2020; Giusti et al., 2022b; Bodnar et al., 2021a), or on combinatorial complexes (Battiloro et al., 2024; Eitan et al., 2024). Although overall valuable and insightful, such a fragmented research trend is slowing the development of standardized methodologies and software for TDL, as well as limiting the analysis of its cost-benefits trade-offs (Papamarkou et al., 2024). We argue that these two relevant aspects are considerably hindering the use and application of TDL beyond the community of experts.

**Current Efforts and Gaps for TDL Standardization.** TopoX (Hajij et al., 2024b) and TopoBench-mark (Telyatnikov et al., 2024) have become the reference Python libraries for *developing* and *benchmarking* TDL models, respectively. However, despite their potential in defining and implementing novel standardized methodologies in the field, the current focus of these packages is on replicating and analyzing existing message-passing CCNNs. Works like Jogl et al. (2022b,a) have instead focused on making TDL accessible and reproducible by porting models to the graph domain. They do so via principled transformations from combinatorial topological domains to graphs. However, although these architectures over the resulting graph-expanded representations are as expressive as their TDL counterparts (using the Weisfeiler-Lehman criterion (Xu et al., 2019a)), they are neither formally equivalent to nor a generalization of their TDL counterparts. Due to loss of topological information during the graph expansion, the GNNs on the resulting graph do not preserve the same topological symmetry as their TDL counterparts.

**Contributions.** This works seeks to accelerate TDL research and increase its accessibility and standardization for outside practitioners. To that end, we introduce a novel joint methodological and software framework that easily enables the development of new TDL architectures in a principled way—overcoming the limitations of existing works. We outline our main contributions and specify which of the field's open problems (as defined in Papamarkou et al. (2024)) they help answer:

- **Systematic Generalization.** We propose the first method to systematically generalize ***any neural network*** to its topological counterpart with minimal adaptation. Specifically, we define a novel expansion mechanism that transforms a combinatorial complex into a collection of graphs, enabling the training of TDL models as an ensemble of synchronized models. To our knowledge, this is the

first method which is designed to work across many topological domains. (Open problems 6, 11: need for foundational, cross-domain TDL.)

- **General Architectures.** Our method induces a novel wide class of TDL architectures, *Generalized Combinatorial Complex Networks* (GCCNs), portrayed in Fig. 1. GCCNs *(i)* formally generalize CCNNs, *(ii)* are cell permutation equivariant, and *(iii)* are as expressive as CCNNs. (Open problem 9: consolidating TDL advantages in a unified theory.)

- **Implementation.** We provide TopoTune, a lightweight PyTorch module for designing and implementing GCCNs fully integrated into TopoBenchmark (Telyatnikov et al., 2024). Using TopoTune, both newcomers and expert TDL practitioners can, for the first time, easily define and iterate upon TDL architectures. (Open problems 1, 4: need for accessible TDL, need for software.)

- **Benchmarking.** Using TopoTune, we create a broad class of GCCNs using four base GNNs and one base Transformer over two combinatorial topological spaces (simplicial and cell complexes). A wide range of experiments on graph-level and node-level benchmark datasets shows GCCNs generally outperform existing CCNNs, often with smaller model sizes. Some of these results are obtained with GCCNs that cannot be reduced to standard CCNNs, further underlining our methodological contribution. We will provide all code and experiment scripts in the camera-ready paper. (Open problem 3: need for standardized benchmarking.)

**Outline.** Section 2 provides necessary background. Section 3 motivates and positions our work in the current TDL literature. Section 4 introduces and discusses GCCNs. Section 5 introduces and describes TopoTune. Finally, Section 6 showcases extensive numerical experiments and comparisons.

## 2 BACKGROUND

To properly contextualize our work, we revisit in this section the fundamentals of combinatorial complexes and CCNNs—closely following the works of Hajij et al. (2023) and Battiloro et al. (2024)—as well as the notion of augmented Hasse graphs. Appendix A provides a brief introduction to all topological domains used in TDL, such as simplicial and cell complexes.

**Combinatorial Complex.** A *combinatorial complex* is a triple $(\mathcal{V}, \mathcal{C}, \mathrm{rk})$ consisting of a set $\mathcal{V}$, a subset $\mathcal{C}$ of the powerset $\mathcal{P}(\mathcal{V}) \backslash \{\emptyset\}$, and a rank function $\mathrm{rk} : \mathcal{C} \to \mathbb{Z}_{\geq 0}$ with the following properties:

1. for all $v \in \mathcal{V}$, $\{v\} \in \mathcal{C}$ and $\mathrm{rk}(\{v\}) = 0$;

2. the function $\mathrm{rk}$ is order-preserving, i.e., if $\sigma, \tau \in \mathcal{C}$ satisfy $\sigma \subseteq \tau$, then $\mathrm{rk}(\sigma) \leq \mathrm{rk}(\tau)$.

The elements of $\mathcal{V}$ are the nodes, while the elements of $\mathcal{C}$ are called cells (i.e., group of nodes). The rank of a cell $\sigma \in \mathcal{C}$ is $k := \mathrm{rk}(\sigma)$, and we call it a $k$-cell. $\mathcal{C}$ simplifies notation for $(\mathcal{V}, \mathcal{C}, \mathrm{rk})$, and its dimension is defined as the maximal rank among its cell: $\dim(\mathcal{C}) := \max_{\sigma \in \mathcal{C}} \mathrm{rk}(\sigma)$.

**Neighborhoods.** Combinatorial complexes can be equipped with a notion of neighborhood among cells. In particular, a neighborhood $\mathcal{N} : \mathcal{C} \to \mathcal{P}(\mathcal{C})$ on a combinatorial complex $\mathcal{C}$ is a function that assigns to each cell $\sigma$ in $\mathcal{C}$ a collection of "neighbor cells" $\mathcal{N}(\sigma) \subset \mathcal{C} \cup \emptyset$. Examples of neighborhood functions are *adjacencies*, connecting cells with the same rank, and *incidences*, connecting cells with different consecutive ranks. Usually, up/down incidences $\mathcal{N}_{I,\uparrow}$ and $\mathcal{N}_{I,\downarrow}$ are defined as

$$\mathcal{N}_{I,\uparrow}(\sigma) = \{\tau \in \mathcal{C} \,|\, \mathrm{rk}(\tau) = \mathrm{rk}(\sigma) + 1, \sigma \subset \tau\}, \quad \mathcal{N}_{I,\downarrow}(\sigma) = \{\tau \in \mathcal{C} \,|\, \mathrm{rk}(\tau) = \mathrm{rk}(\sigma) - 1, \tau \subset \sigma\}. \tag{1}$$

Therefore, a $k + 1$-cell $\tau$ is a neighbor of a $k$-cell $\sigma$ w.r.t. to $\mathcal{N}_{I,\uparrow}$ if $\sigma$ is contained in $\tau$; analogously, a $k - 1$-cell $\tau$ is a neighbor of a $k$-cell $\sigma$ w.r.t. to $\mathcal{N}_{I,\downarrow}$ if $\tau$ is contained in $\sigma$. These incidences induce up/down adjacencies $\mathcal{N}_{A,\uparrow}$ and $\mathcal{N}_{A,\downarrow}$ as

$$\mathcal{N}_{A,\uparrow}(\sigma) = \{\tau \in \mathcal{C} \,|\, \mathrm{rk}(\tau) = \mathrm{rk}(\sigma), \exists \delta \in \mathcal{C} : \mathrm{rk}(\delta) = \mathrm{rk}(\sigma) + 1, \tau \subset \delta, \text{ and } \sigma \subset \delta\},$$
$$\mathcal{N}_{A,\downarrow}(\sigma) = \{\tau \in \mathcal{C} \,|\, \mathrm{rk}(\tau) = \mathrm{rk}(\sigma), \exists \delta \in \mathcal{C} : \mathrm{rk}(\delta) = \mathrm{rk}(\sigma) - 1, \delta \subset \tau, \text{ and } \delta \subset \sigma\}. \tag{2}$$

Therefore, a $k$-cell $\tau$ is a neighbor of a $k$-cell $\sigma$ w.r.t. to $\mathcal{N}_{A,\uparrow}$ if they are both contained in a $k + 1$-cell $\delta$; analogously, a $k$-cell $\tau$ is a neighbor of a $k$-cell $\sigma$ w.r.t. to $\mathcal{N}_{A,\downarrow}$ if they both contain a $k - 1$-cell $\delta$. Other neighborhood functions can be defined for specific applications (Battiloro et al., 2024).

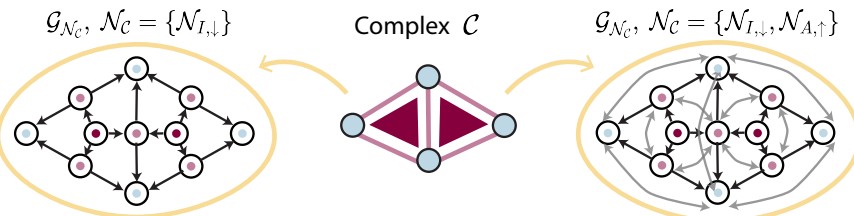

Figure 2: **Augmented Hasse graphs.** Expansions of a combinatorial complex $\mathcal{C}$ (middle) into two augmented Hasse graphs: (left) the Hasse graph induced by $\mathcal{N}_\mathcal{C} = \{\mathcal{N}_{I,\downarrow}\}$; (right) the augmented Hasse graph induced by $\mathcal{N}_\mathcal{C} = \{\mathcal{N}_{I,\downarrow}, \mathcal{N}_{A,\uparrow}\}$. Information on cell rank is discarded (we retain rank color for illustrative purposes).

**Combinatorial Complex Message-Passing Neural Networks.** Let $\mathcal{C}$ be a combinatorial complex, and $\mathcal{N}_\mathcal{C}$ a collection of neighborhood functions. The $l$-th layer of a CCNN updates the embedding $\mathbf{h}_\sigma^l \in \mathbb{R}^{F^l}$ of cell $\sigma$ as

$$\mathbf{h}_\sigma^{l+1} = \phi\left(\mathbf{h}_\sigma^l, \bigotimes_{\mathcal{N} \in \mathcal{N}_\mathcal{C}} \bigoplus_{\tau \in \mathcal{N}(\sigma)} \psi_{\mathcal{N},\text{rk}(\sigma)}\left(\mathbf{h}_\sigma^l, \mathbf{h}_\tau^l\right)\right) \in \mathbb{R}^{F^{l+1}}, \tag{3}$$

where $\mathbf{h}_\sigma^0 := \mathbf{h}_\sigma$ are the initial features, $\bigoplus$ is an intra-neighborhood aggregator, $\bigotimes$ is an inter-neighborhood aggregator. The functions $\psi_{\mathcal{N},\text{rk}(\cdot)} : \mathbb{R}^{F^l} \to \mathbb{R}^{F^{l+1}}$ and the update function $\phi$ are learnable functions, which are typically homogeneous across all neighborhoods and ranks. In other words, the embedding of a cell is updated in a learnable fashion by first aggregating messages with neighboring cells per each neighborhood, and then by further aggregating across neighborhoods. We remark that by this definition, all CCNNs are message-passing architectures. Moreover, they can only leverage neighborhood functions that consider all ranks in the complex.

**Augmented Hasse Graphs.** In TDL, a Hasse graph is a graph expansion of a combinatorial complex. Specifically, it represents the incidence structure $\mathcal{N}_{I,\downarrow}$ by representing each cell (node, edge, face) as a node and drawing edges between cells that are incident to each other. For example, if three edges bound a face, then in the Hasse graph, the three nodes representing the three edges will each share an edge with the node representing the face. Going beyond just considering $\mathcal{N}_{I,\downarrow}$, given a collection of *multiple* neighborhood functions, every combinatorial complex $\mathcal{C}$ can be expanded into a unique graph representation. We refer to this representation as an augmented Hasse graph (Hajij et al., 2023). Formally, let $\mathcal{N}_\mathcal{C}$ be a collection of neighborhood functions on $\mathcal{C}$: the augmented Hasse graph $\mathcal{G}_{\mathcal{N}_\mathcal{C}}$ of $\mathcal{C}$ induced by $\mathcal{N}_\mathcal{C}$ is a directed graph $\mathcal{G}_{\mathcal{N}_\mathcal{C}} = (\mathcal{C}, \mathcal{E}_{\mathcal{N}_\mathcal{C}})$ with cells as nodes, and edges given by

$$\mathcal{E}_{\mathcal{N}_\mathcal{C}} = \{(\tau, \sigma) | \sigma, \tau \in \mathcal{C}, \exists\, \mathcal{N} \in \mathcal{N}_\mathcal{C} : \tau \in \mathcal{N}(\sigma)\}. \tag{4}$$

The augmented Hasse graph of a combinatorial complex is thus obtained by considering the cells as nodes, and inserting directed edges among them if the cells are neighbors in $\mathcal{C}$. Fig. 2 shows an example of a combinatorial complex as well as *i)* a Hasse graph and *ii)* an augmented Hasse graph. Notably, such a representation of a combinatorial complex discards all information about cell rank.

# 3 MOTIVATION AND RELATED WORKS

As outlined in the introduction, TDL lacks a comprehensive framework for easily creating and experimenting with novel topological architectures—unlike the more established GNN field. This section outlines some previous works that have laid important groundwork in addressing this challenge.

**Formalizing CCNNs on graphs.** The position paper (Veličković, 2022) proposed that any function over a higher-order domain can be computed via message passing over a transformed graph, but without specifying how to design GNNs that reproduce CCNNs. Later, (Hajij et al., 2023) proposed that, given a combinatorial complex $\mathcal{C}$ and a collection of neighborhoods $\mathcal{N}_\mathcal{C}$, a message-passing GNN that runs over the augmented Hasse graph $\mathcal{G}_{\mathcal{N}_\mathcal{C}}$ is equivalent to a specific CCNN as in (3) running over $\mathcal{C}$ using: i) $\mathcal{N}_\mathcal{C}$ as collection of neighborhoods; ii) same intra- and inter-aggregations, i.e., $\bigoplus = \bigotimes$; and iii) no rank- and neighborhood-dependent message functions, i.e., $\psi_{\mathcal{N},\text{rk}(\cdot)} = \psi \; \forall \mathcal{N} \in \mathcal{N}_\mathcal{C}$.

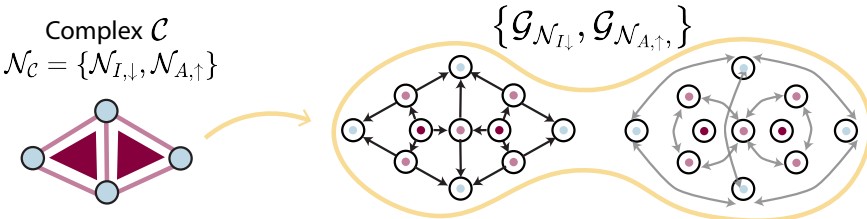

Figure 3: **Ensemble of strictly augmented Hasse Graphs.** Given a complex $\mathcal{C}$ with neighborhood structure including both incidence and upper adjacency (left), this graph expansion (right) produces one augmented Hasse graph for *each* neighborhood.

**Retaining expressivity, but not topological symmetry.** Jogl et al. (2022a;b) demonstrate that GNNs on augmented Hasse graphs $\mathcal{G}_{\mathcal{N}_\mathcal{C}}$ are as expressive as CCNNs on $\mathcal{C}$ (using the WL criterion), suggesting that some CCNNs can be simulated with standard graph libraries. [2]. However, as the authors state, such GNNs do not structurally distinguish between cells of different ranks or neighborhoods, collapsing topological relationships into a single representation. For instance, in a molecule (cellular complex), two bonds (edges) may simultaneously share multiple neighborhoods: lower-adjacent through a shared atom (node) and upper-adjacent through a shared ring (face). A GNN on $\mathcal{G}_{\mathcal{N}_\mathcal{C}}$ collapses these distinctions, applying the same weights to all connections and losing the structural symmetries encoded in the domain. While this may suffice for preserving expressivity, it is inherently a very different computation than that of TDL models.

**The Particular Case of Hypergraphs.** Hypergraph neural networks have long relied on graph expansions (Telyatnikov et al., 2023), which has allowed the field to leverage advances in the graph domain and, by extension, a much wider breadth of models (Antelmi et al., 2023; Papillon et al., 2023). Most hypergraph models are expanded into graphs using the star (Zhou et al., 2006; Solé et al., 1996), the clique (Bolla, 1993; Rodríguez, 2002; Gibson et al., 2000), or the line expansion (Bandyopadhyay et al., 2020). As noted by Agarwal et al. (2006), many hypergraph learning algorithms leverage graph expansions.

The success story of hypergraph neural networks motivates further research on new graph-based expansions that generalize and subsume current CCNNs. These expansions could, at the same time, encompass current CCNNs *and* exploit progress in the GNN field. Therefore, returning to our core goal of accelerating and democratizing TDL while preserving its theoretical properties, we propose a two-part approach: a **novel graph-based methodology** able to generate general architectures (Section 4), and a **lightweight software framework** to easily and widely implement it (Section 5).

## 4 GENERALIZED COMBINATORIAL COMPLEX NEURAL NETWORKS

We propose Generalized Combinatorial Complex Neural Networks (GCCNs), a novel broad class of TDL architectures. GCCNs overcome the limitations of previous graph-based TDL architectures by leveraging the notions of *strictly augmented Hasse graphs* and *per-rank neighborhoods*.

**Ensemble of Strictly Augmented Hasse Graphs.** This graph expansion method (see Fig. 3) extends from the the established definition of an augmented Hasse graph (see Fig. 2). Specifically, given a combinatorial complex $\mathcal{C}$ and a collection of neighborhood functions $\mathcal{N}_\mathcal{C}$, we expand it into $|\mathcal{N}_\mathcal{C}|$ graphs, each of them representing a neighborhood $\mathcal{N} \in \mathcal{N}_\mathcal{C}$. In particular, the *strictly augmented Hasse graph* $\mathcal{G}_\mathcal{N} = (\mathcal{C}_\mathcal{N}, \mathcal{E}_\mathcal{N})$ of a neighborhood $\mathcal{N} \in \mathcal{N}_\mathcal{C}$ is a directed graph whose nodes $\mathcal{C}_\mathcal{N}$ and edges $\mathcal{E}_\mathcal{N}$ are given by:

$$\mathcal{C}_\mathcal{N} = \{\sigma \in \mathcal{C} \,|\, \mathcal{N}(\sigma) \neq \emptyset\}, \; \mathcal{E}_\mathcal{N} = \{(\tau, \sigma) \,|\, \tau \in \mathcal{N}(\sigma)\}. \tag{5}$$

Following the same arguments from Hajij et al. (2023), a GNN over the strictly augmented Hasse graph $\mathcal{G}_\mathcal{N}$ induced by $\mathcal{N}$ is equivalent to a CCNN running over $\mathcal{C}$ and using $\mathcal{N}_\mathcal{C} = \{\mathcal{N}\}$ up to the (self-)update of the cells in $\mathcal{C}/\mathcal{C}_\mathcal{N}$.

---

[2]The same authors generalize these ideas to non-standard message-passing GNNs (Jogl et al., 2024)

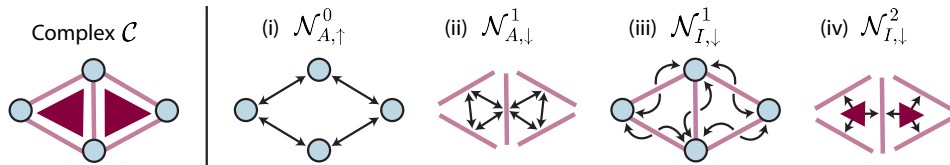

Figure 4: **Per-rank neighborhoods**. Given a complex $\mathcal{C}$ (left), we illustrate four examples of per-rank neighborhoods (right). In each case, they only include rank-specific cells.

**Per-rank Neighborhoods.** The standard definition of adjacencies and incidences given in Section 2 implies that they are applied to each cell regardless of its rank. For instance, consider a combinatorial complex of dimension two with nodes (0-cells), edges (1-cells), and faces (2-cells).

- Employing the down incidence $\mathcal{N}_{I,\downarrow}$ as in (1) means the edges must exchange messages with their endpoint nodes, and faces must exchange messages with the edges on their sides. It is impossible for edges to exchange messages while faces do not.

- Employing the up adjacency $\mathcal{N}_{A,\uparrow}$ as in (2) means the nodes must exchange messages with other edge-connected nodes, and edges must exchange messages the other edges bounding the same faces. It is impossible for nodes to exchange messages while edges do not.

This limitation increases the computational burden of standard CCNNs while not always increasing the learning performance, as we will show in the numerical results. For this reason, we introduce *per-rank neighborhoods*, depicted in Fig. 4. Formally, a per-rank neighborhood function $\mathcal{N}^r$ is a neighborhood function that, regardless of its definition, maps a cell $\sigma$ to the empty set if $\sigma$ is not a $r$-cell (i.e., a cell of rank $r$). For example, the up/down $r$-*incidences* $\mathcal{N}_{I,\uparrow}^r$ and $\mathcal{N}_{I,\downarrow}^r$ are defined as

$$\mathcal{N}_{I,\uparrow}^r(\sigma) = \begin{cases} \{\tau \in \mathcal{C} \mid \mathrm{rk}(\tau) = \mathrm{rk}(\sigma) + 1, \sigma \subset \tau\} \text{ if } \mathrm{rk}(\sigma) = r \\ \emptyset \text{ otherwise} \end{cases}, \tag{6}$$

$$\mathcal{N}_{I,\downarrow}^r(\sigma) = \begin{cases} \{\tau \in \mathcal{C} \mid \mathrm{rk}(\tau) = \mathrm{rk}(\sigma) - 1, \sigma \subset \tau\} \text{ if } \mathrm{rk}(\sigma) = r \\ \emptyset \text{ otherwise} \end{cases}, \tag{7}$$

and the up/down $r$-*adjacencies* $\mathcal{N}_{A,\uparrow}^r$ and $\mathcal{N}_{A,\downarrow}^r$ can be obtained analogously. So, it is now straightforward to model a setting in which:

- Employing only $\mathcal{N}_{I,\downarrow}^1$ (Fig. 4(iii)) allows edges to exchange messages with their bounding nodes but not triangles with their bounding edges.

- Employing only $\mathcal{N}_{A,\uparrow}^0$ (Fig. 4(i)) allows nodes to exchange messages with their edge-connected nodes but not edges do not exchange messages with other edges that are part of their same faces.

**Generating Graph-based TDL Architectures.** We use these notions to define a novel graph-based methodology for generating principled TDL architectures. Given a combinatorial complex $\mathcal{C}$ and a set $\mathcal{N}_{\mathcal{C}}$ of neighborhoods, the method works as follows (see also Fig. 1):

**A.** $\mathcal{C}$ is expanded into an *ensemble* of strictly augmented Hasse graphs—one for each $\mathcal{N} \in \mathcal{N}_{\mathcal{C}}$.

**B.** Each strictly augmented Hasse graph $\mathcal{G}_{\mathcal{N}}$ and the features of its cells are independently processed by a base model.

**C.** An aggregation module $\bigotimes$ synchronizes the cell features across the different strictly augmented Hasse graphs (as the same cells can belong to multiple strictly augmented Hasse graphs).

This method enables an ensemble of synchronized models per layer— the $\omega_{\mathcal{N}}$s—each of them applied to a specific strictly augmented Hasse graph.[3] Additionally, such a pipeline confers unprecedented flexibility in choosing a subset of neighborhoods of interest, allowing the consideration of *per-rank neighborhoods* within TDL. The rest of this section formalizes the architectures induced by this methodology and describes their theoretical properties.

---

[3]Contrary to past CCNN simulation works that apply a model to the singular, whole augmented Hasse graph.

**Generalized Combinatorial Complex Networks.** We formally introduce a broad class of novel TDL architectures called Generalized Combinatorial Complex Networks (GCCNs), depicted in Fig. 1. Let $\mathcal{C}$ be a combinatorial complex containing $|\mathcal{C}|$ cells and $\mathcal{N}_{\mathcal{C}}$ a collection of neighborhoods on it. Assume an arbitrary labeling of the cells in the complex, and denote the $i$-th cell with $\sigma_i$. Denote by $\mathbf{H} \in \mathbb{R}^{|\mathcal{C}| \times F}$ the feature matrix collecting some embeddings of the cells on its rows, i.e., $[\mathbf{H}]_i = \mathbf{h}_{\sigma_i}$, and by $\mathbf{H}_{\mathcal{N}} \in \mathbb{R}^{|\mathcal{C}_{\mathcal{N}}| \times F}$ the submatrix containing just the embeddings of the cells belonging to the strictly augmented Hasse graph $\mathcal{G}_{\mathcal{N}}$ of $\mathcal{N}$. The $l$-th layer of a GCCN updates the embeddings of the cells $\mathbf{H}^l \in \mathbb{R}^{|\mathcal{C}| \times F^l}$ as

$$\mathbf{H}^{l+1} = \phi \left( \mathbf{H}^l, \bigotimes_{\mathcal{N} \in \mathcal{N}_{\mathcal{C}}} \omega_{\mathcal{N}}(\mathbf{H}^l_{\mathcal{N}}, \mathcal{G}_{\mathcal{N}}) \right) \in \mathbb{R}^{|\mathcal{C}| \times F^{l+1}}, \tag{8}$$

where $\mathbf{H}^0$ collects the initial features, and the update function $\phi$ is a learnable row-wise update function, i.e., $[\phi(\mathbf{A}, \mathbf{B})]_i = \phi([\mathbf{A}]_i, [\mathbf{B}]_i)$. The neighborhood-dependent sub-module $\omega_{\mathcal{N}} : \mathbb{R}^{|\mathcal{C}_{\mathcal{N}}| \times F^l} \rightarrow \mathbb{R}^{|\mathcal{C}_{\mathcal{N}}| \times F^{l+1}}$, which we refer to as the *neighborhood message function*, is a learnable (matrix) function that takes as input the whole strictly augmented Hasse graph of the neighborhood, $\mathcal{G}_{\mathcal{N}}$ and the embeddings of the cells that are part of it, and gives as output a processed version of them. Finally, the inter-neighborhood aggregation module $\bigotimes$ synchronizes the possibly multiple neighborhood messages arriving on a single cell across multiple strictly augmented Hasse graphs into a single message. In this way, the embedding of a cell collects information about the whole relational structures induced by each (nonempty) neighborhood. GCCNs enjoy increased flexibility over CCNS (eq. 3) as their neighborhoods are allowed to be rank-dependent and the corresponding $\omega_{\mathcal{N}}$'s are not necessarily message-passing based.

> **Theoretical properties of GCCNs.**
>
> 1. **Generality.** GCCNs formally generalize CCNNs.
>
>    **Proposition 1.** *Let $\mathcal{C}$ be a combinatorial complex. Let $\mathcal{N}_{\mathcal{C}}$ be a collection of neighborhoods on $\mathcal{C}$. Then, there exists a GCCN that exactly reproduces the computation of a CCNN over $\mathcal{C}$ using $\mathcal{N}_{\mathcal{C}}$.*
>
> 2. **Permutation Equivariance.** Generalizing CCNNs, GCCNs layers are equivariant with respect to the relabeling of cells in the combinatorial complex.
>
>    **Proposition 2.** *A GCCN layer is cell permutation equivariant if the neighborhood message function is node permutation equivariant and the inter-neighborhood aggregator is cell permutation invariant.*
>
> 3. **Expressivity**. The expressiveness of TDL models is tied to their ability to distinguish non-isomorphic graphs. Variants of the Weisfeiler-Leman (WL) test, like the cellular WL for cell complexes (Bodnar et al., 2021a), set upper bounds on their corresponding TDL models' expressiveness, as the WL test does for GNNs (Xu et al., 2019a).
>
>    **Proposition 3.** *GCCNs are strictly more expressive than CCNNs.*
>
> The proofs are provided in Appendix B.1, B.2, and B.3, respectively.

Given Proposition 1, GCCNs allow us to define general TDL models using any neighborhood message function $\omega_{\mathcal{N}}$, such as any GNN. Not only does this framework avoid having to approximate CCNN computations, as is the case in previous works [4] (Jogl et al., 2022b;a; 2023), but it also enjoys the same permutation equivariance as regular CCNNs (Proposition 2). We show in Appendix C that the resulting time complexity of a GCCN is a compromise between a typical GNN and a CCNN. Differently from the work in (Hajij et al., 2023), the fact that GCCNs can have arbitrary neighborhood message functions implies that non message-passing TDL models can be readily defined (e.g., by using non message-passing models as neighborhood message functions). Moreover, the fact that the whole strictly augmented Hasse graphs are given as input enables also the usage of multi-layer GNNs as neighborhood message functions. To the best of our knowledge, GCCNs are the only objects in the literature that encompass all the above properties.

---

[4]These models employ GNNs running on one augmented Hasse graph, i.e. a GCCN that, given a collection of neighborhoods $\mathcal{N}_{\mathcal{C}}$, uses a single neighborhood $\mathcal{N}_{tot}$ defined, for a cell $\sigma$, as $\mathcal{N}_{tot}(\sigma) = \bigcup_{\mathcal{N} \in \mathcal{N}_{\mathcal{C}}} \mathcal{N}(\sigma)$.

## 5 TOPOTUNE

Our proposed methodology, together with its resulting GCCNs architectures, addresses the challenge of systematically generating principled, general TDL models. Here, we introduce TopoTune, a software module for defining and benchmarking GCCN architectures on the fly—a vehicle for accelerating and democratizing TDL research. TopoTune is made available as part of TopoBenchmark Telyatnikov et al. (2024). This section details TopoTune's main features.

**Change of Paradigm.**  TopoTune introduces a new perspective on TDL through the concept of "neighborhoods of interest," enabling unprecedented flexibility in architectural design. Previously fixed components of CCNNs become hyperparameters of our framework. Even the choice of topological domain becomes a mere variable, representing a new paradigm in the design and implementation of TDL architectures.

**Accessible TDL.**  Using TopoTune, a practitioner can instantiate customized GCCNs simply by modifying a few lines of a configuration file. In fact, it is sufficient to specify $(i)$ a collection of per-rank neighborhoods $\mathcal{N}_\mathcal{C}$, $(ii)$ a neighborhood message function $\omega_\mathcal{N}$, and optionally $(iii)$ some architectural parameters—e.g., the number $l$ of GCCN layers.[5] For the neighborhood message function $\omega_\mathcal{N}$, the same configuration file enables direct import of models from standard PyTorch libraries, including PyTorch Geometric (Fey & Lenssen, 2019) and Deep Graph Library (Chen et al., 2020b). TopoTune's simplicity provides both newcomers and TDL experts with an accessible tool for defining higher-order topological architectures.

**Accelerating TDL Research.**  TopoTune is fully integrated into TopoBenchmark (Telyatnikov et al., 2024), a comprehensive package offering a wide range of standardized methods and tools for TDL. Practitioners can access ready-to-use models, training pipelines, tasks, and evaluation metrics, including leading open-source models from TopoX (Hajij et al., 2024b). In addition, TopoBenchmark features the largest collection of *topological liftings* currently available—transformations that map graph datasets into higher-order topological domains. Together, TopoBenchmark and TopoTune organize the vast design space of TDL into an accessible framework, providing unparalleled versatility and standardization for practitionners.

## 6 EXPERIMENTS

We present experiments showcasing a broad class of GCCN's constructed with TopoTune. These models consistently match, outperform, or finetune existing CCNNs, often with smaller model sizes. TopoTune's integration into the TopoBenchmark experiment infrastructure ensures a fair comparison with CCNNs from the literature, as data processing, domain lifting, and training are homogeonized.

### 6.1 EXPERIMENTAL SETUP

We generate our class of GCCNs by considering ten possible choices of neighborhood structure $\mathcal{N}_\mathcal{C}$ (including both regular and per-rank, see Appendix E.1) and five possible choices of $\omega_\mathcal{N}$: GCN (Kipf & Welling, 2017), GAT (Velickovic et al., 2017), GIN (Xu et al., 2019b), GraphSAGE (Hamilton et al., 2017), and Transformer (Vaswani et al., 2017). We import these models directly from PyTorch Geometric (Fey & Lenssen, 2019) and PyTorch (Paszke et al., 2019). TopoTune enables running GCCNs on both an ensemble of strictly augmented Hasse graphs (eq. 5) and a single augmented Hasse graph (eq. 4). While CCNN results reflect extensive hyperparameter tuning by Telyatnikov et al. (2024), we fix GCCN training hyperparameters using the TopoBenchmark default configuration.

**Datasets.**  We include a wide range of benchmark tasks (see Appendix E.2) commonly used in the graph and topological domains. MUTAG, PROTEINS, NCI01, and NCI09 (Morris et al., 2020) are graph-level classification tasks about molecules or proteins. ZINC (Irwin et al., 2012) (subset) is a graph-level regression task related to molecular solubility. At the node level, the Cora, CiteSeer, and PubMed tasks (Yang et al., 2016) involve classifying publications (nodes) within citation networks. We consider two cases of combinatorial complexes, simplicial and cellular complexes. We leverage TopoBenchmark's data lifting processes to infer higher-order relationships in these datasets. We only use node features to construct edge and face features.

---

[5]We provide a detailed pseudo-code for TopoTune module in Appendix D.

## 6.2 RESULTS AND DISCUSSION

Table 1: Cross-domain, cross-task, cross-expansion, and cross-$\omega_\mathcal{N}$ comparison of GCCN architectures with top-performing CCNNs benchmarked on TopoBenchmark (Telyatnikov et al., 2024). Best result is in **bold** and results within 1 standard deviation are highlighted blue . Experiments are run with 5 seeds. We report accuracy for classification tasks and MAE for regression.

| Model | Graph-Level Tasks | | | | | Node-Level Tasks | | |
|---|---|---|---|---|---|---|---|---|
| | MUTAG (↑) | PROTEINS (↑) | NCI1 (↑) | NCI109 (↑) | ZINC (↓) | Cora (↑) | Citeseer (↑) | PubMed (↑) |
| Cellular | | | | | | | | |
| CCNN (Best Model on TopoBenchmark) | 80.43 ± 1.78 | 76.13 ± 2.70 | 76.67 ± 1.48 | 75.35 ± 1.50 | 0.34 ± 0.01 | 87.44 ± 1.28 | 75.63 ± 1.58 | 88.64 ± 0.36 |
| GCCN $\omega_\mathcal{N}$ = GAT | 83.40 ± 4.85 | 74.05 ± 2.16 | 76.11 ± 1.69 | 75.62 ± 0.76 | 0.38 ± 0.03 | 88.39 ± 0.65 | 74.62 ± 1.95 | 87.68 ± 0.33 |
| GCCN $\omega_\mathcal{N}$ = GCN | 85.11 ± 6.73 | 74.41 ± 1.77 | 76.42 ± 1.67 | 75.62 ± 0.94 | 0.36 ± 0.01 | 88.51 ± 0.70 | 75.41 ± 2.00 | 88.18 ± 0.26 |
| GCCN $\omega_\mathcal{N}$ = GIN | **86.38 ± 6.49** | 72.54 ± 3.07 | 77.65 ± 1.11 | **77.19 ± 0.21** | **0.19 ± 0.00** | 87.42 ± 1.85 | 75.13 ± 1.17 | 88.47 ± 0.27 |
| GCCN $\omega_\mathcal{N}$ = GraphSAGE | 85.53 ± 6.80 | 73.62 ± 2.72 | **78.23 ± 1.47** | 77.10 ± 0.83 | 0.24 ± 0.00 | 88.57 ± 0.58 | 75.89 ± 1.84 | 89.40 ± 0.57 |
| GCCN $\omega_\mathcal{N}$ = Transformer | 83.83 ± 6.49 | 70.97 ± 4.06 | 73.00 ± 1.37 | 73.20 ± 1.05 | 0.45 ± 0.02 | 84.61 ± 1.32 | 75.05 ± 1.67 | 88.37 ± 0.22 |
| GCCN $\omega_\mathcal{N}$ = Best GNN, 1 Aug. Hasse graph | 85.96 ± 7.15 | 73.73 ± 2.95 | 76.75 ± 1.63 | 76.94 ± 0.82 | 0.31 ± 0.01 | 87.24 ± 0.58 | 74.26 ± 1.47 | 88.65 ± 0.55 |
| Simplicial | | | | | | | | |
| CCNN (Best Model on TopoBenchmark) | 76.17 ± 6.63 | 75.27 ± 2.14 | 76.60 ± 1.75 | 77.12 ± 1.07 | 0.36 ± 0.02 | 82.27 ± 1.34 | 71.24 ± 1.68 | 88.72 ± 0.50 |
| GCCN $\omega_\mathcal{N}$ = GAT | 79.15 ± 4.09 | 74.62 ± 1.95 | 74.86 ± 1.42 | 74.81 ± 1.14 | 0.57 ± 0.03 | 88.33 ± 0.67 | 74.65 ± 1.93 | 87.72 ± 0.36 |
| GCCN $\omega_\mathcal{N}$ = GCN | 74.04 ± 8.30 | 74.91 ± 2.51 | 74.20 ± 2.17 | 74.13 ± 0.53 | 0.53 ± 0.05 | 88.51 ± 0.70 | 75.41 ± 2.00 | 88.19 ± 0.24 |
| GCCN $\omega_\mathcal{N}$ = GIN | 85.96 ± 4.66 | 72.83 ± 2.72 | 76.67 ± 1.62 | 75.76 ± 1.28 | 0.35 ± 0.01 | 87.27 ± 1.63 | 75.05 ± 1.27 | 88.54 ± 0.21 |
| GCCN $\omega_\mathcal{N}$ = GraphSAGE | 75.74 ± 2.43 | 74.70 ± 3.10 | 76.85 ± 1.50 | 75.64 ± 1.94 | 0.50 ± 0.02 | 88.57 ± 0.59 | **75.92 ± 1.85** | 89.34 ± 0.39 |
| GCCN $\omega_\mathcal{N}$ = Transformer | 74.04 ± 4.09 | 70.97 ± 4.06 | 70.39 ± 0.96 | 69.99 ± 1.13 | 0.64 ± 0.01 | 84.4 ± 1.16 | 74.6 ± 1.88 | 88.55 ± 0.39 |
| GCCN $\omega_\mathcal{N}$ = Best GNN, 1 Aug. Hasse graph | 74.04 ± 5.51 | 74.48 ± 1.89 | 75.02 ± 2.24 | 73.91 ± 3.9 | 0.56 ± 0.02 | 87.56 ± 0.66 | 74.5 ± 1.61 | 88.61 ± 0.27 |
| Hypergraph | | | | | | | | |
| CCNN (Best Model on TopoBenchmark) | 80.43 ± 4.09 | **76.63 ± 1.74** | 75.18 ± 1.24 | 74.93 ± 2.50 | 0.51 ± 0.01 | **88.92 ± 0.44** | 74.93 ± 1.39 | **89.62 ± 0.25** |

**GCCNs outperform CCNNs.** Table 1 portrays a cross-comparison between top-performing CCNN models and our class of GCCNs. GCCNs outperform CCNNs in the simplicial and cellular domains across all datasets. Notably, GCCNs in these domains achieve comparable results to hypergraph CCNNs, a feat unattainable by existing CCNNs in node-level tasks. Out of the 16 domain/dataset combinations considered in our experiments, GCCNs outperform the best counterpart CCNN by $> 1\sigma$ in 11 cases. Evidence supports that GCCN's architectural novelties contribute to this performance: *(i)* Representing complexes as ensembles of augmented Hasse graphs, rather than a single augmented Hasse graph, consistently improves results (Table 1). *(ii)* Some GCCNs with per-rank neighborhood structures outperform not only CCNNs but also other GCCNs with regular neighborhoods. For example, on MUTAG, a cellular GCCN with a lightweight, per-rank neighborhood structure makes it 19% the size of the best cellular CCNN on this task.

**GCCNs perform competitively to CCNNs with fewer parameters.** GCCNs are generally more parameter efficient than existing CCNNs in simplicial and cellular domains, and in some instances (MUTAG, NCI1, NCI09), even in the hypergraph domain. Even as GCCNs become more resource-intensive for large graphs with high-dimensional embeddings—as seen in node-level tasks—they maintain a competitive edge. For instance, on the Citeseer dataset, a GCCN ($\omega_\mathcal{N}$ = GraphSAGE) outperforms the best existing CCNN while being 28% smaller. We refer to Table 4. Training times provided in Appendix G show that GCCNs train at comparable speeds on smaller datasets, and slow down for larger datasets, most likely due to TopoTune's on-the-fly graph expansion. In future work, we expect that performing this expansion during preprocessing will address this lag.

**Generalizing existing CCNNs to GCCNs improves performance.** TopoTune makes it easy to iterate upon and improve preexisting CCNNs by replicating their architecture in a GCCN setting. For example, TopoTune can generate a counterpart GCCN by replicating a CCNN's neighborhood structure, aggregation, and training scheme. We show in Table 2 that counterpart GCCNs often achieve comparable or better results than SCCN (Yang et al., 2022) and CWN (Bodnar et al., 2021a) just by sweeping over additional choices of $\omega_\mathcal{N}$ (same as in Table 1). In the single augmented Hasse graph regime, GCCN models are consistently more lightweight, up to half their size (see Table 5).

Table 2: We compare existing CCNNs with $\omega_{\mathcal{N}}$-modified GCCN counterparts. We show the result for best choice of $\omega_{\mathcal{N}}$. Experiments are run with 5 seeds.

| Model | MUTAG | PROTEINS | NCI1 | NCI109 | Cora | Citeseer | PubMed |
|---|---|---|---|---|---|---|---|
| SCCN Yang et al. (2022) | | | | | | | |
| Benchmark results Telyatnikov et al. (2024) | 70.64 ± 5.90 | 74.19 ± 2.86 | **76.60 ± 1.75** | **77.12 ± 1.07** | 82.19 ± 1.07 | 69.60 ± 1.83 | **88.18 ± 0.32** |
| GCCN, on ensemble of strictly aug. Hasse graphs | **82.13 ± 4.66** | **75.56 ± 2.48** | 75.6 ± 1.28 | 74.19 ± 1.44 | **88.06 ± 0.93** | 74.67 ± 1.24 | 87.70 ± 0.19 |
| GCCN, on 1 aug. Hasse graph | 69.79 ± 4.85 | 74.48 ± 2.67 | 74.63 ± 1.76 | 70.71 ± 5.50 | 87.62 ± 1.62 | **74.86 ± 1.7** | 87.80 ± 0.28 |
| CWN Bodnar et al. (2021a) | | | | | | | |
| Benchmark results Telyatnikov et al. (2024) | 80.43 ± 1.78 | **76.13 ± 2.70** | 73.93 ± 1.87 | 73.80 ± 2.06 | 86.32 ± 1.38 | **75.20 ± 1.82** | **88.64 ± 0.36** |
| GCCN, on ensemble of strictly aug. Hasse graphs | **84.26 ± 8.19** | 75.91 ± 2.75 | 73.87 ± 1.10 | 73.75 ± 0.49 | 85.64 ± 1.38 | 74.89 ± 1.45 | 88.40 ± 0.46 |
| GCCN, on 1 aug. Hasse graph | 81.70 ± 5.34 | 75.05 ± 2.39 | **75.14 ± 0.76** | **75.39 ± 1.01** | **86.44 ± 1.33** | 74.45 ± 1.59 | 88.56 ± 0.55 |

**TopoTune finds parameter-efficient GCCNs.** By easily exploring a wide landscape of possible GCCNs for a given task, TopoTune helps identify models that maximize performance while minimizing model size. Fig. 5 illustrates this trade-off by comparing the performance and size of selected GCCNs (see Appendix H for more). On the PROTEINS dataset, two GCCNs using per-rank neighborhood structures (orange and purple) achieve performance within 2% of the best result while requiring as little as 48% of the parameters. This reduction is due to fewer neighborhoods $\mathcal{N}$, resulting in fewer $\omega_{\mathcal{N}}$ blocks per GCCN layer. Similarly, on ZINC, lightweight neighborhood structures (orange and dark green) deliver competitive results with reduced parameter costs. Node-level tasks, however, see less benefit, likely due to the larger graph sizes and higher-dimensional input features.

**Impactfulness of GNN choice is dataset specific.** Fig. 5 also provides insights into the impact of neighborhood message functions. On ZINC, GIN clearly outperforms all other models, which do not even appear in the plot's range. In the less clear-cut cases of PROTEINS and Citeseer, we observe a trade-off between neighborhood structure and message function complexity. We find that more complex base models (GIN, GraphSAGE) on lightweight neighborhood structures perform comparably to simpler base models (GAT, GCN) on more complete neighborhood structures.

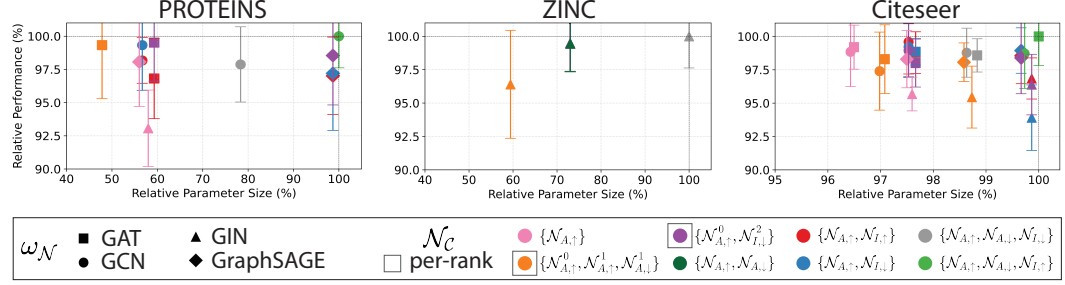

Figure 5: **GCCN performance versus size.** We compare various GCCNs across three datasets on the cellular domain, two graph-level (left, middle) and one node-level (right). Each GCCN (point) has a different neighborhood structure $\mathcal{N}_{\mathcal{C}}$, some of which can only be represented as per-rank structures ($\square$ in legend), and message function $\omega_{\mathcal{N}}$. The amount of layers is kept constant according to the best performing model. The axes are scaled relative to this model.

## 7 CONCLUSION

This work introduces a simple yet powerful graph-based methodology for constructing Generalized Combinatorial Complex Neural Networks (GCCNs), TDL architectures that generalize and subsume standard CCNNs. Additionally, we introduce TopoTune, the first lightweight software module for systematically and easily implementing new TDL architectures across many topological domains. In doing so, we have addressed, either in part or in full, 7 of the 11 open problems of the field defined by some of its leaders in Papamarkou et al. (2024). Future work includes customizing GCCNs for application-specific and potentially sparse or multimodal datasets, and leveraging software from state-of-the-art GNNs. We hope TopoTune will also help bridge the gap with other fields such as attentional learning and $k$-hop higher-order GNNs (Morris et al., 2019; Maron et al., 2019).

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
