

Figure 6: **Topological Deep Learning Domains.** Nodes in blue, (hyper)edges in pink, and faces in dark red. Figure adopted from Papillon et al. (2023).

## A  DOMAINS OF TOPOLOGICAL DEEP LEARNING

We summarize the different discrete domains leveraged within TDL and, in doing so, contextualize how combinatorial complexes generalize all of them. To that end, we will closely follow the description of Papillon et al. (2023), using as well its very clarifying Figure 6. We recommend this survey for a high-level overview of TDL literature, and the more extensive work of Hajij et al. (2023) for a detailed formulation of the field. We also refer to Appendix C of Battiloro et al. (2024) for a concise mathematical description of each domain. From left to right in Figure 6, the different domains in TDL are:

### TRADITIONAL DISCRETE DOMAINS

**Set / Pointcloud.** A collection of points called *nodes* without any additional structure.

**Graph.** A set of points (nodes) connected with edges that denote pairwise relationships.

### SET + PART-WHOLE RELATIONS

**Simplicial Complex.** A generalization of a graph that incorporates hierarchical part-whole relations through the multi-scale construction of cells. Nodes are rank 0-cells that can be combined to form edges (rank 1 cells). Edges are, in turn, combined to form faces (rank 2 cells), which are combined to form volumes (rank 3 cells), and so on. In particular, each cell $\sigma$ in a simplicial complex must contain all lower dimensional cells $\tau$ such that $\tau \subseteq \sigma$. Therefore, faces must be triangles, volumes must be tetrahedrons, and so forth.

**Cellular Complex.** A generalization of an simplicial complex in which cells are not limited to simplexes, but may instead take any shape: faces can involve more than three nodes, volumes more than four faces, and so on. This flexibility endows CCs with greater expressivity than simplicial complexes (Bodnar et al., 2021a), but still edges only connect pairs of nodes.

### SET + SET-TYPE RELATIONS

**Hypergraph:** A generalization of a graph, in which higher-order edges called hyperedges can connect arbitrary sets of two or more nodes. Connections in HGs represent set-type relationships, in which participation in an interaction is not implied by any other relation in the system. This makes HGs an ideal choice for data with abstract and arbitrarily large interactions of equal importance, such as semantic text and citation networks.

### SET + PART-WHOLE AND SET-TYPE RELATIONS

**Combinatorial Complex:** A structure that combines features of hypergraphs and cellular complexes. Like a hypergraph, edges may connect any number of nodes. Like a cellular complex, cells can be combined to form higher-ranked structures. Hence, combinatorial complexes generalize all other topological domains.

# B PROOFS

## B.1 PROOF OF GENERALITY

The proof is straightforward. It is sufficient to set $\omega_{\mathcal{N}}(\mathbf{H}_{\mathcal{N}}^l, \mathcal{G}_{\mathcal{N}})$ to $\{\bigoplus_{y \in \mathcal{N}(\sigma)} \psi_{\mathcal{N}, \mathrm{rk}(\sigma)}(\mathbf{h}_\sigma^l, \mathbf{h}_\tau^l)\}_{\sigma \in \mathcal{C}}$ in (8) as all $y \in \mathcal{N}(\sigma)$ are part of the node set $\mathcal{C}_{\mathcal{N}}$ of the strictly augmented Hasse graph of $\mathcal{N}$ by definition.

## B.2 PROOF OF EQUIVARIANCE

As for GNNs, an amenable property for GCCNNs is the awareness w.r.t. relabeling of the cells. In other words, given that the order in which the cells are presented to the networks is arbitrary -because CCs, like (undirected) graphs, are purely combinatorial objects-, one would expect that if the order changes, the output changes accordingly. To formalize this concept, we need the following notions.

**Matrix Representation of a Neighborhood.** Assume again to have a combinatorial complex $\mathcal{C}$ containing $C := |\mathcal{C}|$ cells and a neighborhood function $\mathcal{N}$ on it. Assume again to give an arbitrary labeling to the cells in the complex, and denote the $i$-th cell with $\sigma_i$. The matrix representation of the neighborhood function is a matrix $\mathbf{N}_{\mathcal{N}} \in \mathbb{R}^{C \times C}$ such that $\mathbf{N}_{i,j} = 1$ if the $\sigma_j \in \mathcal{N}(\sigma_i)$ or zero otherwise. We notice that the submatrix $\widetilde{\mathbf{N}}_{\mathcal{N}} \in \mathbb{R}^{|\mathcal{C}_{\mathcal{N}}| \times |\mathcal{C}_{\mathcal{N}}|}$ obtained by removing all the zero rows and columns is the adjacency matrix of the strictly augmented Hasse graph $\mathcal{G}_{\mathcal{C}_{\mathcal{N}}}$ induced by $\mathcal{N}$.

**Permutation Equivariance.** Let $\mathcal{C}$ be combinatorial complex, $\mathcal{N}_{\mathcal{C}}$ a collection of neighborhoods on it, and $\mathbf{N} = \{\mathbf{N}_{\mathcal{N}}\}_{\mathcal{N} \in \mathcal{N}_{\mathcal{C}}}$ the set collecting the corresponding neighborhood matrices. Let $\mathbf{P} \in \mathbb{R}^{C \times C}$ be a permutation matrix. Finally, denote by $\mathbf{PH}$ the permuted embeddings and by $\{\mathbf{PN}_{\mathcal{N}}\mathbf{P}^T\}_{\mathcal{N} \in \mathcal{N}_{\mathcal{C}}}$, the permuted neighborhood matrices. We say that a function $f : (\mathbf{H}^l, \mathbf{B}) \mapsto \mathbf{H}^{l+1}$ is cell permutation equivariant if $f\left(\mathbf{PH}^l, \{\mathbf{PN}_{\mathcal{N}}\mathbf{P}^T\}_{\mathcal{N} \in \mathcal{N}_{\mathcal{C}}}\right) = \mathbf{P}f\left(\mathbf{H}^l, \{\mathbf{N}_{\mathcal{N}}\}_{\mathcal{N} \in \mathcal{N}_{\mathcal{C}}}\right)$ for any permutation matrix $\mathbf{P}$. Intuitively, the permutation matrix changes the arbitrary labeling of the cells, and a permutation equivariant function is a function that reflects the change in its output.

*Proof of Proposition 2.* We follow the approach from (Bodnar et al., 2021a). Given any permutation matrix $\mathbf{P}$, for a cell $\sigma_i$, let us denote its permutation as $\sigma_{\mathbf{P}(i)}$ with an abuse of notation. Let $\mathbf{h}_{\sigma_i}^{l+1}$ be the output embedding of cell $\sigma_i$ for the $l$-th layer of a GCCN taking $(\mathbf{H}^l, \{\mathbf{N}_{\mathcal{N}}\}_{\mathcal{N} \in \mathcal{N}_{\mathcal{C}}})$ as input, and $\mathbf{h}_{\sigma_{\mathbf{P}(i)}}^{l+1}$ be the output embedding of cell $\sigma_{\mathbf{P}(i)}$ for the same GCCN layer taking $\left(\mathbf{PH}^l, \{\mathbf{PN}_{\mathcal{N}}\mathbf{P}^T\}_{\mathcal{N} \in \mathcal{N}_{\mathcal{C}}}\right)$ as input. To prove the permutation equivariance, it is sufficient to show that $\mathbf{h}_{\sigma_i}^{l+1} = \mathbf{h}_{\sigma_{\mathbf{P}(i)}}^{l+1}$ as the update function $\phi$ is row-wise, i.e., it independently acts on each cell. To do so, we show that the (multi-)set of embeddings being passed to the neighborhood message function, aggregation, and update functions are the same for the two cells $\sigma_i$ and $\sigma_{\mathbf{P}(i)}$. The neighborhood message functions act on the strictly augmented Hasse graph of $\mathcal{G}_{\mathcal{C}_{\mathcal{N}}}$ of $\mathcal{N}$, thus we work with the submatrix $\widetilde{\mathbf{N}}_{\mathcal{N}}$. The neighborhood message function is assumed to be *node* permutation equivariant, i.e., denoting again the embeddings of the cells in $\mathcal{G}_{\mathcal{C}_{\mathcal{N}}}$ with $\mathbf{H}_{\mathcal{C}_{\mathcal{N}}}^l \in \mathbb{R}^{|\mathcal{C}_{\mathcal{N}}| \times F^l}$ and identifying $\mathcal{G}_{\mathcal{C}_{\mathcal{N}}}$ with $\widetilde{\mathbf{N}}_{\mathcal{N}}$, it holds that $\omega_{\mathcal{N}}(\mathbf{P}_{\mathcal{C}_{\mathcal{N}}}\mathbf{H}_{\mathcal{C}_{\mathcal{N}}}^l, \mathbf{P}_{\mathcal{C}_{\mathcal{N}}}\widetilde{\mathbf{N}}_{\mathcal{N}}\mathbf{P}_{\mathcal{C}_{\mathcal{N}}}^T) = \mathbf{P}_{\mathcal{C}_{\mathcal{N}}}\omega_{\mathcal{N}}(\mathbf{H}_{\mathcal{C}_{\mathcal{N}}}^l, \widetilde{\mathbf{N}}_{\mathcal{N}})$, where $\mathbf{P}_{\mathcal{C}_{\mathcal{N}}}$ is the submatrix of $\mathbf{P}$ given by the rows and the columns corresponding to the cells in $\mathcal{G}_{\mathcal{C}_{\mathcal{N}}}$. This assumption, together with the assumption that the inter-neighborhood aggregation is assumed to be *cell* permutation invariant, i.e. $\bigotimes_{\mathcal{N} \in \mathcal{N}_{\mathcal{C}}} \mathbf{P}_{\mathcal{C}_{\mathcal{N}}}\omega_{\mathcal{N}}(\mathbf{H}_{\mathcal{C}_{\mathcal{N}}}^l, \widetilde{\mathbf{N}}_{\mathcal{N}}) = \bigotimes_{\mathcal{N} \in \mathcal{N}_{\mathcal{C}}} \omega_{\mathcal{N}}(\mathbf{H}_{\mathcal{C}_{\mathcal{N}}}^l, \widetilde{\mathbf{N}}_{\mathcal{N}})$, trivially makes the overall composition of the neighborhood message function with the inter-neighborhood aggregation *cell* permutation invariant. This fact, together with the fact that the (labels of) the neighbors of the cell $\sigma_i$ in $\mathcal{N}$ are given by the nonzero elements of the $i$-th row of $\mathbf{N}_{\mathcal{N}}$, or the corresponding row of $\widetilde{\mathbf{N}}_{\mathcal{N}}$, and that the columns and rows of $\widetilde{\mathbf{N}}_{\mathcal{N}}$ are permuted in the same way the rows of the feature matrix $\mathbf{H}_{\mathcal{C}_{\mathcal{N}}}^l$ are permuted, implies

$$[\widetilde{\mathbf{N}}_{\mathcal{N}}]_{i,j} = [\mathbf{P}_{\mathcal{C}_{\mathcal{N}}}\widetilde{\mathbf{N}}_{\mathcal{N}}\mathbf{P}_{\mathcal{C}_{\mathcal{N}}}^T]_{\mathbf{P}_{\mathcal{C}_{\mathcal{N}}}(i), \mathbf{P}_{\mathcal{C}_{\mathcal{N}}}(j)}, \tag{9}$$

thus that $\sigma_i$ and $\sigma_{\mathbf{P}(i)}$ receive the same neighborhood message from the neighboring cells in $\mathcal{N}$, for all $\mathcal{N} \in \mathcal{N}_{\mathcal{C}}$.

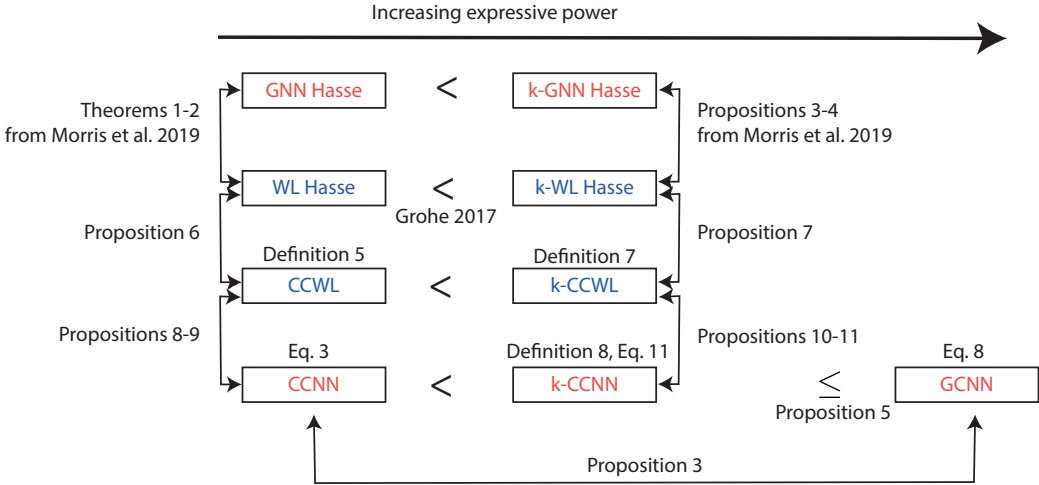

Figure 7: Graphical summary of the definitions and propositions related to the expressivity of CCNNs and GCNNs and of the different WL tests. Neural networks expressivity is in red, and WL test expressivity is in blue.

### B.3 PROOF OF EXPRESSIVITY

We provide the theory required to prove Proposition 3, i.e., to prove that GCNNs are strictly more expressive than CCNNs. The definitions and propositions from this subsection are summarized in Figure 7. This figure serves as a graphical reading guide for the subsection.

#### B.3.1 HOMOMORPHISM AND ISOMORPHISM INDUCED BY NEIGHBORHOODS

We first recall the notion of homomorphism of a combinatorial complexes (CC) from (Hajij et al., 2023) and generalize it to the notions of homomorphism and of isomorphism of combinatorial complexes induced by a neighborhood $\mathcal{N}$.

**Definition 1** (CC-Homomorphism (Hajij et al., 2023)). A homomorphism from a CC $(\mathcal{V}_1, \mathcal{C}_1, \mathrm{rk}_1)$ to a CC $(\mathcal{V}_2, \mathcal{C}_2, \mathrm{rk}_2)$, also called a CC-homomorphism, is a function $f : \mathcal{C}_1 \to \mathcal{C}_2$ that satisfies the following conditions:

1. If $\sigma, \tau \in \mathcal{C}_1$ satisfy $\sigma \subseteq \tau$, then $f(\sigma) \subseteq f(\tau)$.

2. If $\sigma \in \mathcal{C}_1$, then $\mathrm{rk}_1(\sigma) \geq \mathrm{rk}_2(f(\sigma))$.

Definition 1 proposes a CC-homomorphism that respects the incidence structures of the CCs, denoted by the symbol $\subseteq$ in the definition above. We generalize Definition 1 by allowing CC-homomorphisms to take into account a labeling of the cells and to be defined in terms of general neighborhood structures beyond incidence. We first define a labeled combinatorial complex.

**Definition 2** (Labeled Combinatorial Complex). A labeled combinatorial complex $(\mathcal{C}, \ell)$ is a CC $\mathcal{C}$ equipped with a cell coloring $\ell : \mathcal{C} \mapsto \Sigma$ with arbitrary codomain $\Sigma$. We say that $\ell(\sigma)$ is a label or color of cell $\sigma \in \mathcal{C}$.

Next, we provide our definitions of homomorphisms.

**Definition 3** (CC-Homomorphism induced by $(\mathcal{N}_1, \mathcal{N}_2)$). A homomorphism from a CC $(\mathcal{V}_1, \mathcal{C}_1, \mathrm{rk}_1)$ with neighborhood $\mathcal{N}_1$ to a CC $(\mathcal{V}_2, \mathcal{C}_2, \mathrm{rk}_2)$ with neighborhood $\mathcal{N}_2$, also called a CC-homomorphism induced by $(\mathcal{N}_1, \mathcal{N}_2)$, is a function $f : \mathcal{C}_1 \to \mathcal{C}_2$ that satisfies: If $\sigma, \tau \in \mathcal{C}_1$ are such that $\tau \in \mathcal{N}_1(\sigma)$, then $f(\tau) \in \mathcal{N}_2(f(\sigma))$. A labeled CC-homomorphism induced by $(\mathcal{N}_1, \mathcal{N}_2)$ is a CC-homomorphism induced by $(\mathcal{N}_1, \mathcal{N}_2)$ that additionally respects labeling of the cells, that is: if $\sigma, \tau \in \mathcal{C}_1$ have the same label, then $f(\sigma), f(\tau) \in \mathcal{C}_2$ also have the same label.

We prove that a CC-homomorphism induced by $(\mathcal{N}_1, \mathcal{N}_2)$ is equivalent to a homomorphism of the respective strictly augmented Hasse graphs $\mathcal{G}_{\mathcal{N}_1}$, $\mathcal{G}_{\mathcal{N}_2}$.

**Proposition 4.** *For every CC-homomorphism $f$ from $\mathcal{C}_1$ to $\mathcal{C}_2$ induced by $(\mathcal{N}_1, \mathcal{N}_2)$, there exists a unique graph homomorphism between their respective strictly augmented Hasse graphs $\mathcal{G}_{\mathcal{N}_1}$ and $\mathcal{G}_{\mathcal{N}_2}$.*

*Proof.* Consider $f$ a CC-homomorphism from $\mathcal{C}_1$ to $\mathcal{C}_2$ induced by $(\mathcal{N}_1, \mathcal{N}_2)$ as in Definition 3. Define the function $\tilde{f}$ from nodes of $\mathcal{G}_{\mathcal{N}_1}$ to the nodes of $\mathcal{G}_{\mathcal{N}_2}$ corresponding to $f$, i.e., $\tilde{f} : \mathcal{C}_{\mathcal{N}_1} \mapsto \mathcal{C}_{\mathcal{N}_2}$ defined as $\tilde{f}(\tilde{\sigma}) = f(\sigma)$ where $\tilde{\sigma}$ is the node in $\mathcal{G}_{\mathcal{N}_1}$ corresponding to the cell $\sigma$ in $\mathcal{C}_1$, and $f(\sigma)$ is the node in $\mathcal{G}_{\mathcal{N}_2}$ corresponding to the cell $f(\sigma)$ in $\mathcal{C}_2$. We show that $\tilde{f}$ is a graph homomorphism from $\mathcal{G}_{\mathcal{N}_1}$ to $\mathcal{G}_{\mathcal{N}_2}$, i.e., a function from the nodes of $\mathcal{G}_{\mathcal{N}_1}$ to the nodes of $\mathcal{G}_{\mathcal{N}_2}$ that preserves edges.

By definition of the CC-homomorphism induced by $(\mathcal{N}_1, \mathcal{N}_2)$, we have: if $\tau \in \mathcal{N}_1(\sigma)$ then $f(\tau) \in \mathcal{N}_2(f(\sigma))$. Recognizing that $\mathcal{N}_1$ defines edges of $\mathcal{G}_{\mathcal{N}_1}$, and $\mathcal{N}_2$ defines edgess of $\mathcal{G}_{\mathcal{N}_2}$, we have: if $(\tilde{\sigma}, \tilde{\tau})$ is an edge in $\mathcal{G}_{\mathcal{N}_1}$, then $(\tilde{f}(\tilde{\sigma}), \tilde{f}(\tilde{\tau}))$ is an edge in $\mathcal{G}_{\mathcal{N}_2}$. Thus, a CC-homomorphism induced by $(\mathcal{N}_1, \mathcal{N}_2)$ gives a homomorphism of the strictly augmented Hasse graphs.

Conversely, if $\tilde{f}$ is a graph homomorphism from $\mathcal{G}_{\mathcal{N}_1}$ to $\mathcal{G}_{\mathcal{N}_2}$, then we similarly construct a CC-homomorphism $f$ between $\mathcal{C}_1$ and $\mathcal{C}_2$. This concludes the proof. $\square$

Lastly, we can define a notion of CC-isomorphism induced by neighborhood structures.

**Definition 4** (CC-Isomorphism induced by $(\mathcal{N}_1, \mathcal{N}_2)$). A isomorphism from a CC $(\mathcal{V}_1, \mathcal{C}_1, \mathrm{rk}_1)$ with neighborhood $\mathcal{N}_1$ to a CC $(\mathcal{V}_2, \mathcal{C}_2, \mathrm{rk}_2)$ with neighborhood $\mathcal{N}_2$, also called a CC-isomorphism induced by $(\mathcal{N}_1, \mathcal{N}_2)$, is an invertible CC-homomorphism induced by $(\mathcal{N}_1, \mathcal{N}_2)$ whose inverse is a CC-isomorphism induced by $(\mathcal{N}_2, \mathcal{N}_1)$. A labeled CC-isomorphism induced by $(\mathcal{N}_1, \mathcal{N}_2)$ is a CC-isomorphism that additionally respects labels.

### B.3.2 WEISFEILER-LEMAN (WL) TESTS ON COMBINATORIAL COMPLEXES

We propose two WL tests, called CCWL and (set-based) $k$-CCWL that generalize the WL and the (set-based) $k$-WL tests to labeled combinatorial complexes. We start with the generalization of the WL test to labeled combinatorial complexes.

**Definition 5** (The CC Weisfeiler-Leman (CCWL) test on labeled combinatorial complexes). Let $(\mathcal{C}, \ell)$ be a labeled combinatorial complex. Let $\mathcal{N}$ be a neighborhood on $\mathcal{C}$. The scheme proceeds as follows:

- Initialization: Cells $\sigma$ are initialized with the labels given by $\ell$, i.e.: for all $\sigma \in \mathcal{C}$, we set: $c^0_{\sigma, \ell} = \ell(\sigma)$.

- Refinement: Given colors of cells at iteration $t$, the refinement step computes the color of cell $\sigma$ at the next iteration $c^{t+1}_{\sigma, \ell}$ using a perfect HASH function as follows:

$$c^t_{\mathcal{N}}(\sigma) = \left\{\left\{ c^t_{\sigma', \ell} \mid \forall \sigma' \in \mathcal{N}(\sigma) \right\}\right\},$$
$$c^{t+1}_{\sigma, \ell} = \mathrm{HASH}\left( c^t_{\sigma, \ell}, c^t_{\mathcal{N}}(\sigma) \right).$$

- Termination: The algorithm stops when an iteration leaves the coloring unchanged.

Next, we generalize the set-based $k$-WL test to labeled combinatorial complexes, called the $k$-CCWL test. The set-based $k$-WL test is employed in (Morris et al., 2019) where colors are defined on $k$-sets of nodes, as opposed to $k$-tuples of nodes in the standard $k$-WL test. Specifically, we denote $[\mathcal{C}]^k$ the set of $k$-sets formed with cells of $\mathcal{C}$. We generalize the definition of neighborhood of $k$-sets of vertices from (Morris et al., 2019) to neighborhood of $k$-sets of cells.

**Definition 6** (Neighborhood of $k$-sets of cells). Given a $k$-set of cells $s = \{\sigma_1, \ldots, \sigma_k\}$ in $[\mathcal{C}]^k$, we define its neighborhood as the function $\mathcal{N}_k : [\mathcal{C}]^k \mapsto \mathcal{P}([\mathcal{C}]^k)$ defined as:

$$\mathcal{N}_k(s) = \left\{ t \in [\mathcal{C}]^k \mid |s \cap t| = k - 1 \right\}. \tag{10}$$

Figure 8: Notations for Proposition 5. Denote $|\mathcal{C}|$ the number of cells. The number of $k$-sets that contain a given cell $\sigma$ is equal to $\binom{|\mathcal{C}|-1}{k-1}$. The feature on one $k$-set that contains a given cell has dimension $d$. Thus, $\mathbf{H}^l \in \mathbb{R}^{|\mathcal{C}| \times F}$ for $F = \binom{|\mathcal{C}|-1}{k-1}d$. We note that the $k$-sets for one row do not correspond to the $k$-sets of another row. However, for every row, there is the same number of $k$-sets that contain the cell $\sigma$ characteristic of that row.

**Definition 7** (The CC $k$-Weisfeiler-Leman ($k$-CCWL) test on combinatorial complexes). Let $(\mathcal{C}, \ell)$ be a labeled combinatorial complex. Let $\mathcal{N}$ be a neighborhood on $\mathcal{C}$. The scheme proceeds as follows:

- Initialization: Every $k$-set $s$ in $[\mathcal{C}]^k$ is initialized with a color that corresponds to the CC-isomorphism type of the sub-CC defined by $s = \{\sigma_1, \ldots, \sigma_k\}$ induced by $\mathcal{N}|_s$ where $\mathcal{N}|_s$ is the neighborhood $\mathcal{N}$ restricted to $s$. This means that two $k$-sets $s$ and $s'$ get the same color if and only if there is a labeled CC-isomorphism (for labeling function $\ell$) between the sub-CCs corresponding to the cells in $s$ and $s'$, respectively.

- Refinement: Given colors of $k$-sets at iteration $t$, the refinement step computes the color of the $k$-set $s$ at the next iteration $c_{s,\ell}^{t+1}$ using a perfect HASH function, as follows:

$$c_{\mathcal{N}_k(s),\ell}^t = \left\{\left\{ c_{s',\ell}^t \mid \forall s' \in \mathcal{N}_k(s) \right\}\right\},$$
$$c_{s,\ell}^{t+1} = \text{HASH}\left( c_{s,\ell}^t, c_{\mathcal{N}_k(s),\ell}^t \right).$$

- Termination: The algorithm stops when an iteration leaves the coloring unchanged.

Two combinatorial complexes are deemed non-isomorphic according to the CCWL and $k$-CCWL respectively, if their color histograms differ upon termination of the scheme. If the histograms are the same, we cannot conclude.

### B.3.3 Definitions of $k$-GNNs and $k$-CCNNs

We generalize the definition of $k$-GNNs by (Morris et al., 2019) into a definition of $k$-CCNNs.

**Definition 8** ($k$-CCNNs). Let $(\mathcal{C}, \ell)$ be a labeled CC. In each $k$-CCNN layer $t$, the feature vector $h_k^{(t)}(s) \in \mathbb{R}^d$ for each $k$-set $s$ in $[\mathcal{C}]^k$ is updated into $h_k^{(t+1)}(s)$ as follows:

$$h_k^{(t+1)}(s) = U\left( h_k^{(t)}(s) \cdot W_1^{(t)} + \sum_{u \in \mathcal{N}_k(s)} h_k^{(t)}(u) \cdot W_2^{(t)} \right) \in \mathbb{R}^d, \tag{11}$$

where $W_1^{(t)}, W_2^{(t)}$ are matrices of parameters for layer $t$, $\mathcal{N}_k$ the neighborhood structure on $k$-sets, and $U$ is an update function.

Then, we show that $k$-CCNNs of Definition 8 form a subclass of GCNNs.

**Proposition 5.** *GCNNs generalize and subsume $k$-CCNNs.*

*Proof.* Let be given a $k$-CCNN defined by equation 11. We show that we can recover equation 11 by an appropriate choice of feature dimensionality $F$, update function $\phi$ and sub-module $\omega_\mathcal{N}$ in equation 8 defining GCNNs, and thus that any $k$-CCNN can be expressed as a GCNN.

For simplicity of notations, we assume that the layers of the $k$-CCNN have same feature dimensionality, denoted $d$. Given that there are $\binom{|\mathcal{C}|-1}{k-1}$ $k$-sets containing a given cell $\sigma$, we define $F = \binom{|\mathcal{C}|-1}{k-1}d$ to be the feature dimensionality of the layers of a GCNN. Denote $\mathbf{H}^l(\sigma)$ the row of $\mathbf{H}^l$ containing $F$-dimensional feature corresponding to cell $\sigma$, as well as $\mathbf{H}^l(\sigma, s)$ the subrow containing the $d$-dimensional feature corresponding to one $k$-set $s$ to which $\sigma$ belongs. Figure 8 illustrates these notations. We then define:

$$\mathbf{H}^{l+1} = \phi\left(\mathbf{H}^l, \omega(\mathbf{H}^l)\right)$$

by defining $\phi$ and $\omega$ on $(\sigma, s)$-blocks of the matrix $\mathbf{H}^l$. Specifically, we have:

$$\mathbf{H}^{l+1}(\sigma, s) = \phi_{(\sigma,s)}\left(\mathbf{H}^l(\sigma, s), \omega_{(\sigma,s)}(\mathbf{H}^l)\right)$$

$$= U\left(\mathbf{H}^l(\sigma, s) \cdot W_1^{(t)} + \sum_{u \in \mathcal{N}_k(s)} \mathbf{H}^l(\sigma, u) \cdot W_2^{(t)}\right),$$

where:

$$\omega_{(\sigma,s)}(\mathbf{H}^l) = \sum_{u \in \mathcal{N}_k(s)} \mathbf{H}^l(\sigma, u) \cdot W_2^{(t)}, \qquad \phi_{(\sigma,s)}(A, B) = U(A \cdot W_1^{(1)} + B). \tag{12}$$

In other words, we first use $\omega$ defined as a sequence of $\omega_{(\sigma,s)}$ to update each $(\sigma, s)$-block of $\mathbf{H}^l$ into an auxiliary feature $B = \tilde{\mathbf{H}}^l$. Then, we use $\phi$ as a sequence of $\phi_{(\sigma,s)}$ to perform a block-wise operations. Thus, we have built a GCNN that reproduces the computations of the $k$-CCNN. Therefore, GCNNs generalize and subsume $k$-CCNNs. $\square$

### B.3.4 RELATIONSHIPS BETWEEN CCWL/GCWL TESTS AND CCNNS/GCNNS

We prove relationships between the expressivity of the WL tests and the expressivity of the corresponding neural networks. We first recall results on WL tests on graphs and GNNs (Morris et al., 2019). In what follows, $(G, \ell)$ is a labeled graph, and $W^{(t)}$ denote the parameters of a GNN up to layer $t$. We encode the initial labels $\ell(v)$, for a vertex $v$, by vectors $h^{(0)}(v) \in \mathbb{R}^{1 \times d}$.

**WL/GNNs and $k$-WL/$k$-GNNs** Theorem 1 in (Morris et al., 2019) states that, for every encoding of the graph labels $\ell(v)$ as $d$-vectors $h^{(0)}(v)$, and for every choice of parameters $W^{(t)}$, the coloring $c(t)_\ell$ of the WL test always refines the coloring $h(t)$ induced by the GNN parameterized by $W^{(t)}$. Theorem 2 in (Morris et al., 2019) states that there exists parameter matrices $W^{(t)}$ such that GNNs have exactly the same power as the WL test. Consequently, we say that GNNs have the same expressivity as the WL test. Similarly, Propositions 3 and 4 from (Morris et al., 2019) show that $k$-GNNs have the same expressivity as the WL test.

**CCWL/CCNNs and $k$-CCWL/GCNNs** We generalize the equivalence between WL tests and GNNs to the framework of CCs. First, we prove two propositions establishing equivalence of WL tests between CCs and Hasse graphs.

**Proposition 6** (CCWL and WL on the Hasse graph). *Let $(\mathcal{C}, \ell)$ be a labeled CC. Let $\mathcal{N}$ be one neighborhood on this CC and $\mathcal{G}_\mathcal{N}$ the associated strictly augmented Hasse graph. The CCWL test defined in Def. 5 is equivalent to the WL test defined on $\mathcal{G}_\mathcal{N}$.*

*Proof.* We prove the equivalence between the CCWL and the WL on $\mathcal{G}_\mathcal{N}$.

*Equivalence of initializations.* The CCWL test initializes cell colors using the labels given by $\ell$. The labeling function $\ell$ labels cells of $\mathcal{C}$ and therefore its restriction to $\mathcal{C}_\mathcal{N}$ labels nodes of the associated Hasse graph $\mathcal{G}_\mathcal{N}$. This turns $\mathcal{G}_\mathcal{N}$ into a labeled graph $(\mathcal{G}_\mathcal{N}, \ell_{\mathcal{C}_\mathcal{N}})$. We initialize the WL test on $\mathcal{G}_\mathcal{N}$ with colors from $\ell_{C_\mathcal{N}}$.

*Equivalence of refinements.* By construction of the strictly augmented Hasse graph $\mathcal{G}_\mathcal{N}$, nodes in $\mathcal{G}_\mathcal{N}$ are cells in $\mathcal{C}_\mathcal{N}$ and edges in $\mathcal{G}_\mathcal{N}$ are neighbors in $\mathcal{C}_\mathcal{N}$ for the neighborhood $\mathcal{N}$. Thus, the refinement equation of the CCWL test is equal to the refinement equation of the WL test on $\mathcal{G}_\mathcal{N}$. This proves that CCWL and the WL on $\mathcal{G}_\mathcal{N}$ are equivalent. □

**Proposition 7** ($k$-CCWL and $k$-WL on the Hasse graph)**.** *Let $(\mathcal{C}, \ell)$ be a labeled CC. Let $\mathcal{N}$ be one neighborhood on this $CC$ and $\mathcal{G}_\mathcal{N}$ the associated strictly augmented Hasse graph. The $k$-CCWL defined in Def. 7 is equivalent to the $k$-WL test on $\mathcal{G}_\mathcal{N}$.*

*Proof.* We prove the equivalence between the $k$-CCWL and the $k$-WL on $\mathcal{G}_\mathcal{N}$.

*Equivalence of initializations.* The $k$-CCWL test initializes colors of $k$-sets based on the CC-isomorphism class of every sub-CC defined by every $k$-set. Using Proposition 4, the CC-isomorphism class of a sub-CC $s$ corresponds to the graph isomorphism class on the associated subgraph in the strictly augmented Hasse graph. We initialize the $k$-WL test on $\mathcal{G}_\mathcal{N}$ with colors on $k$-sets associated with this isomorphism class.

*Equivalence of refinements.* By construction of the strictly augmented Hasse graph $\mathcal{G}_\mathcal{N}$, $k$-sets of nodes in $\mathcal{G}_\mathcal{N}$ are $k$-sets of cells in $\mathcal{C}_\mathcal{N}$, and the neighborhoods of $k$-sets of nodes defined in (Morris et al., 2019) are the neighborhoods of $k$-sets of cells defined in Definition 6. Thus, the refinement equation of the $k$-CCWL test is equal to the refinement equation of the $k$-WL test on $\mathcal{G}_\mathcal{N}$.

This proves that $k$-CCWL and the $k$-WL on $\mathcal{G}_\mathcal{N}$ are equivalent. □

Given the equivalence between the computations in $\mathcal{C}$ and in $\mathcal{G}_\mathcal{N}$ provided by Proposition 6, we can pull the results from Theorems 1 and 2 from (Morris et al., 2019) and provide the following propositions.

**Proposition 8.** *Let $(\mathcal{C}, \ell)$ be a labeled CC. Then for all $t \geq 0$ and for all choices of initial colorings $h^{(0)}$ consistent with $\ell$, and weights $\mathbf{W}^{(t)}$, $c_\ell^{(t)} \sqsubseteq h^{(t)}$, i.e., the coloring $c_l^{(t)}$ induced by the CCWL test refines the coloring induced by the CCNN $h^{(t)}$.*

**Proposition 9.** *Let $(\mathcal{C}, \ell)$ be a labeled CC. Then for all $t \geq 0$ there exists a sequence of weights $\mathbf{W}^{(t)}$, and a CCNN architecture such that $c_\ell^{(t)} \equiv h^{(t)}$., i.e., the coloring of the CCWL and the CCNN are equivalent.*

Consequently, CCNNs have the same power as the CCWL. Next, by Proposition 7, we can also pull the results from Propositions 3 and 4 from (Morris et al., 2019) and provide the following propositions.

**Proposition 10.** *Let $(\mathcal{C}, \ell)$ be a labeled CC and let $k \geq 2$. Then, for all $t \geq 0$, for all choices of initial colorings $h_k^{(0)}$ consistent with $\ell$ and for all weights $\mathbf{W}^{(t)}$, $c_{s,k,\ell}^{(t)} \sqsubseteq h_k^{(t)}$.*

**Proposition 11.** *Let $(\mathcal{C}, \ell)$ be a labeled CC and let $k \geq 2$. Then, for all $t \geq 0$ there exists a sequence of weights $\mathbf{W}^{(t)}$, and a $k$-CCNN architecture such that $c_{s,k,\ell}^{(t)} \equiv h_k^{(t)}$.*

Consequently, $k$-CCNNs have the same power as the $k$-CCWL.

### B.3.5    PROOF

We now provide the proof for Proposition 3 that states that GCCNs are strictly more expressive than CCNNs.

*Proof.* We prove that GCNNs are strictly more powerful than CCNNs in distinguishing non-isomorphic combinatorial complexes. We leverage the propositions of this subsection summarized on Figure 7.

By Proposition 5, GCNN have at least the same expressive power as $k$-CCNNs. By Propositions 10-11, $k$-CCNNs have the same expressive power as the $k$-CCWL. By Proposition 7, the $k$-CCWL test is equivalent to the $k$-WL test on the associated strictly augmented Hasse graph. It is known (e.g., (Grohe, 2017)) that the $k$-WL test on graph is strictly more powerful than the WL test. Thus, the $k$-WL test on the strictly augmented Hasse graph is strictly more powerful than the WL test on that

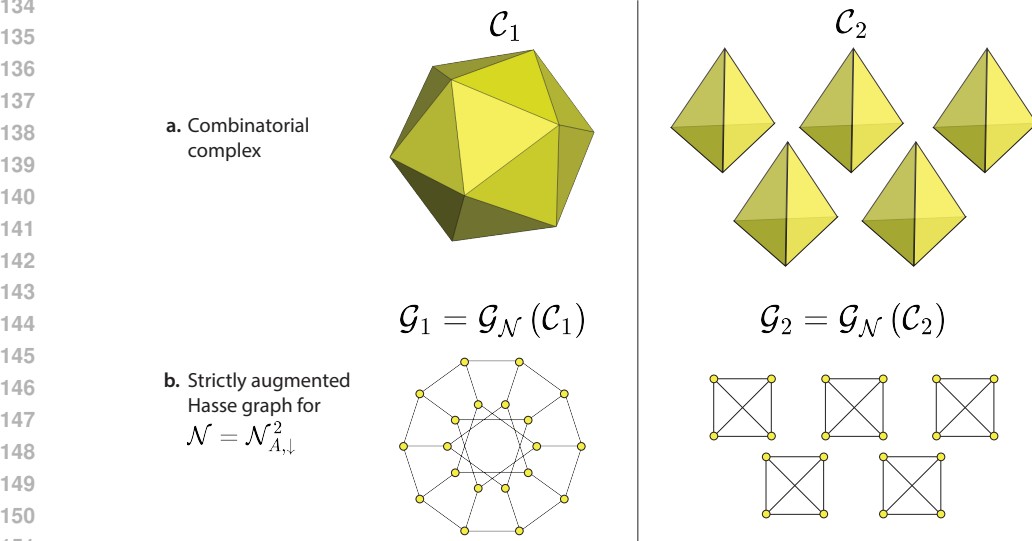

Figure 9: **a.** Pair of combinatorial complexes: $\mathcal{C}_1$ is an icosahedron polygon, and $\mathcal{C}_2$ is five tetrahedrons. **b.** Strictly augmented Hasse graphs corresponding to each combinatorial complex, given a choice of neighborhood $\mathcal{N}_{A,\uparrow}^2$.

same graph. By Proposition 6, the WL test on the strictly augmented Hasse graph is equivalent to the CCWL test on the corresponding CC. By Propositions 8-9, the CCWL test on the CC has the same expressive power as CCNNs.

Consequently, we have shown that GCNN are strictly more powerful than CCNNs in distinguishing nonisomorphic CCs. □

Additionally, we construct two combinatorial complexes $\mathcal{C}_1$ and $\mathcal{C}_2$ that are indistinguishable by CCNNs but distinguishable by GCNNs.

Let $\mathcal{C}_1$ and $\mathcal{C}_2$ be two combinatorial complexes with a neighborhood structure $\mathcal{N}_\mathcal{C} = \mathcal{N}_{A,\downarrow}^2$ (down-adjacency of faces). These complexes are illustrated in Figure 9a.

The corresponding strictly augmented Hasse graphs $\mathcal{G}_1$ and $\mathcal{G}_2$ (Fig. 9b) represent the 20 faces of each complex as nodes, where each node has degree 3. Thus:

- Both $\mathcal{G}_1$ and $\mathcal{G}_2$ are 3-regular graphs.
- It is known that regular graphs of the same order are indistinguishable by the WL test (see, e.g., (Kiefer, 2020; Morris et al., 2023)).
- Every pair of graphs with $n$ nodes are distinguishable by the $n$-WL test (Morris et al., 2023).

Since CCWL is equivalent to WL on $\mathcal{G}_\mathcal{N}$ (Proposition 6), the two complexes $\mathcal{C}_1$ and $\mathcal{C}_2$ are indistinguishable by CCWL. Since CCWL has the same expressive power as CCNNs (Propositions 8-9), the two complexes $\mathcal{C}_1$ and $\mathcal{C}_2$ are indistinguishable by CCNNs.

Since $k$-CCWL is equivalent to $k$-WL on $\mathcal{G}_\mathcal{N}$ (Proposition 7), the two complexes $\mathcal{C}_1$ and $\mathcal{C}_2$ are distinguishable by $k$-CCWL. Since $k$-CCWL has the same expressive power as $k$-CCNNs (Propositions 10-11), the two complexes $\mathcal{C}_1$ and $\mathcal{C}_2$ are distinguishable by $k$-CCNNs. Since GCNNs generalize and subsume $k$-CCNNs (Proposition 5), $\mathcal{C}_1$ and $\mathcal{C}_2$ are distinguishable by GCNNs.

Thus, we have constructed two combinatorial complexes $\mathcal{C}_1$ and $\mathcal{C}_2$ that are indistinguishable by CCNNs, but are distinguishable by GCNNs.

## C  TIME COMPLEXITY

To analyze the time complexity (in terms of FLOPs) of the Generalized Combinatorial Complex Neural Network (GCCN), we derive the complexity of its submodule $\omega_\mathcal{N}$ and then compute the complexity of a GCCN layer. We then compare it with GNN and CCNN complexity.

### C.1  KEY DEFINITIONS

- **Message Complexity** ($M$)**:** The complexity of a single message computation along a route (e.g., node $\rightarrow$ node). For example, in a Graph Convolutional Network (GCN), a single message is defined as:
$$m_{x \rightarrow y} = a_{xy} \mathbf{h}_y \Theta,$$
where $\mathbf{h}_y$ is a 1$\times$F vector, $\Theta$ is an $F{\times}F$ weight matrix, and $a_{xy}$ is a scalar. This involves a matrix-vector multiplication, contributing a complexity of $O(F^2)$ per message.

- **Update Complexity** ($U$)**:** The complexity of the update function in the reference GNN. For simplicity, we assume the update is an element-wise function, giving $U = O(|N|)$, where $|N|$ is the number of nodes.

### C.2  COMPLEXITY OF $\omega_\mathcal{N}$

Assuming each $\omega_\mathcal{N}$ submodule is a single-layer GNN, the complexity of $\omega_\mathcal{N}$ can be decomposed into three components: **message computation, aggregation, and update.**

$$C_{\omega_\mathcal{N}} = C_{\text{message}} + C_{\text{aggregation}} + C_{\text{update}}$$

This breaks down as:
$$C_{\omega_\mathcal{N}} = 2|E|M + \sum_{n \in N} \deg(n)A + |N|U,$$

where:

- $|E|$: Number of edges in the graph,
- $M$: Complexity per message ($O(F^2)$),
- $\deg(n)$: Degree of node $n$,
- $A$: Complexity of aggregation (e.g., assuming sum/average, $O(F)$),
- $U$: Complexity of the update function ($O(1)$ per node).

Substituting assumptions for convolutional message passing, summation aggregation, and constant node degree $d$:
$$C_{\omega_\mathcal{N}} = 2|E|F^2 + \sum_{n \in N} \deg(n)F + O(|N|),$$

$$C_{\omega_\mathcal{N}} = 2|E|F^2 + |N|dF + O(|N|),$$

$$C_{\omega_\mathcal{N}} = O(|E|F^2 + |N|dF + |N|).$$

### C.3  COMPLEXITY USING COMBINATORIAL COMPLEX NOTATIONS

Up until now, we have expressed $C_{\omega_\mathcal{N}}$ in terms of the nodes and edges making up the strictly expanded Hassse graph it receives as input. To be able to write the complexity of a whole GCCN layer, we must express $C_{\omega_\mathcal{N}}$ in terms of the original cells represented as nodes in the graph. Specifically, we will denote the source cells (cells sending messages) as cells of rank $r$ and the destination cells (cells receiving messages) as cells of rank $r'$. The relationships governing adjacency between the nodes representing these cells will come from the neighborhood $\mathcal{N}$ to which the submodule $\omega_\mathcal{N}$ is assigned.

Rewriting in terms of combinatorial complex notations, where:

- $\|\mathcal{N}\|_0$: Total number of relationships in $\mathcal{N}$ (i.e. number of nonzero entries in matrix corresponding to $\mathcal{N}$),

- $n_{r'}$: Number of $r'$-cells.

- $d_{r'}$: Assumed constant degree of $r'$-cells,

The complexity becomes:

$$C_{\omega_{\mathcal{N}}} = O(\|\mathcal{N}\|_0 F^2 + \text{nrows}(\mathcal{N})d_{r'}F + n_{r'}),$$

$$C_{\omega_{\mathcal{N}}} = O(\|\mathcal{N}\|_0 F^2 + \text{nrows}(\mathcal{N})d_{r'}F + \text{nrows}(\mathcal{N})).$$

## C.4 COMPLEXITY OF A GCCN LAYER

A GCCN layer is composed of a set of $\omega_{\mathcal{N}}$'s, one for each $\mathcal{N} \in \mathcal{N}_{\mathcal{C}}$. The complexity of a GCCN layer is the sum of all the complexities of its submodules, plus the complexity of the module responsible for aggregating the outputs of each neighborhood, i.e. the inter-neighborhood aggregation. We assume this inter-aggregation to be a sum. The layer complexity is:

$$C_{\text{GCCN}} = \sum_{\mathcal{N} \in \mathcal{N}_{\mathcal{C}}} C_{\omega_{\mathcal{N}}} + C_{\text{inter-agg}},$$

where:

$$C_{\text{inter-agg}} = \sum_{r' \in [0, R']} n_{r'} n_{\mathcal{N}_{r'}} F,$$

and $n_{\mathcal{N}_{r'}}$ is the number of neighborhoods sending messages to $r'$-cells.

## C.5 TAKEAWAYS

- **GNN Comparison:** GCCNs increase complexity compared to traditional GNNs due to :
  - the introduction of multiple neighborhoods. A GCCN considers many $\mathcal{N} \in \mathcal{N}_{\mathcal{C}}$, going beyond the simple node-level adjacency $\mathcal{N}_{\mathcal{C}} = A_0$ of a GNN. This is what allows TDL models (GCCNs and CCNNs) to operate on a richer topological space than GNNs.
  - inter-neighborhood aggregation.
- **CCNN Comparison:** Unlike traditional CCNNs, GCCNs allow per-rank neighborhoods, enabling many smaller possible sets of neighborhoods $\mathcal{N}_{\mathcal{C}}$. This more selective inclusion of neighborhoods reduces redundancy. Concretely, this means the sum $\sum_{\mathcal{N} \in \mathcal{N}_{\mathcal{C}}} C_{\omega_{\mathcal{N}}}$ can be smaller.
- **Tradeoff:** GCCNs' time complexity are a compromise between GNNs and CCNNs. While they do introduce $C_{\text{inter-agg}}$ (like CCNNs) and additional elements to the sum $\sum_{\mathcal{N} \in \mathcal{N}_{\mathcal{C}}} C_{\omega_{\mathcal{N}}}$, they can introduce less elements to this sum than CCNNs.

## D  SOFTWARE

Algorithm 1 shows how the TopoTune module instantiates a GCCN by taking a choice of model $\omega_{\mathcal{N}}$ and neighborhoods $\mathcal{N}_{\mathcal{C}}$ as input. Given an input complex $x$, TopoTune first expands it into an ensemble of strictly augmented Hasse graphs that are then passed to their respective $\omega_{\mathcal{N}}$ models within each GCCN layer.

*Remark.* We decided to design the software module of TopoTune, i.e., how to implement GCCNs, as we did for mainly two reasons: (i) the full compatibility with TopoBenchmark (implying consistency of the combinatorial complex instantiations and the benchmarking pipeline), and (ii) the possibility of using GNNs as neighborhood message functions that are not necessarily implemented with a specific library. However, if the practitioner is interested in entirely wrapping the GCCN implementation into Pytorch Geometric or DGL, they can do it by noticing that a GCCN is equivalent to a *heterogeneous* GNN where the heterogeneous graph the whole augmented Hasse graph, with node types given by the rank of the cell (e.g. 0-cells, 1-cells, and 2-cells) while the edge type is given by the per-rank neighborhood function (e.g. "0-cells to 1-cells" or "2-cells to 1-cells" for $\mathcal{N}_{I,\uparrow}^{0}$ and $\mathcal{N}_{I,\downarrow}^{2}$, respectively).

---

**Algorithm 1** TopoTune

---

**Class** TopoTune(torch.nn.Module):
1: **procedure** INIT(neighborhoods, $\omega_n$, $\omega_n\_params$, layers)
2:     $self.omega\_n\_submodels \leftarrow []$
3:     **for** $l \leftarrow 1$ **to** layers **do**
4:         $layer\_models \leftarrow []$
5:         **for** each $nb$ in neighborhoods **do**
6:             $model \leftarrow \omega_n(\omega_n\_params)$
7:             $layer\_models.append(model)$
8:         **end for**
9:         $self.omega\_n\_submodels.append(layer\_models)$
10:     **end for**
11: **end procedure**
12: **procedure** FORWARD($x$)
13:     **for** each layer in $self.omega\_n\_submodels$ **do**
14:         $outputs \leftarrow []$
15:         **for** each $\omega_n\_model$ in layer **do**
16:             $hasse\_graph \leftarrow self.expand\_to\_strictly\_aug\_hasse\_graph(x)$
17:             $outputs.append(\omega_n\_model(hasse\_graph))$
18:         **end for**
19:         $x \leftarrow self.aggregate\_rank\_wise(outputs)$
20:     **end for**
21:     **return** $x$
22: **end procedure**

    **Example Instantiation:**
23: $neighborhoods \leftarrow [[[0, 0], up\_adjacency], [[2, 1], incidence]]$
24: $\omega_n \leftarrow$ torch_geometric.nn.models.GAT
25: $\omega_n\_params \leftarrow \{num\_layers : 2, heads : 4\}$
26: $layers \leftarrow 4$
27: $model \leftarrow$ TopoTune($neighborhoods, \omega_n, \omega_n\_params, layers$)

---

# E  ADDITIONAL DETAILS ON EXPERIMENTS

In this section, we delve into the details of the datasets, hyperparameter search methodology, and computational resources utilized for conducting the experiments.

## E.1  NEIGHBORHOOD STRUCTURES

In order to build a broad class of GCCNs, we consider X different neighborhood structures on which we perform graph expansion. Importantly, three of these structures are lightweight, per-rank neighborhood structures, as proposed in Section 4. The neighborhood structures are:

$$\left\{\mathcal{N}^0_{A,\uparrow},\mathcal{N}^1_{A,\uparrow}\right\} \quad \left\{\mathcal{N}^0_{A,\uparrow},\mathcal{N}^2_{I,\downarrow}\right\} \quad \{\mathcal{N}_{A,\uparrow},\mathcal{N}_{I,\uparrow}\} \quad \{\mathcal{N}_{A,\uparrow},\mathcal{N}_{A,\downarrow},\mathcal{N}_{I,\downarrow}\} \quad \{\mathcal{N}_{A,\uparrow}\}$$

$$\left\{\mathcal{N}_{A,\uparrow},\mathcal{N}^1_{A,\downarrow}\right\} \quad \{\mathcal{N}_{A,\uparrow},\mathcal{N}_{A,\downarrow}\} \quad \{\mathcal{N}_{A,\uparrow},\mathcal{N}_{I,\downarrow}\} \quad \{\mathcal{N}_{A,\uparrow},\mathcal{N}_{A,\downarrow},\mathcal{N}_{I,\uparrow}\} \quad \{\mathcal{N}_{A,\uparrow},\mathcal{N}_{A,\downarrow},\mathcal{N}_{I,\downarrow},\mathcal{N}_{I,\uparrow}\}$$

## E.2  DATASETS

DATASET STATISTICS

Table 3 provides the statistics for each dataset lifted to three topological domains: simplicial complex, cellular complex, and hypergraph. The table shows the number of 0-cells (nodes), 1-cells (edges), and 2-cells (faces) of each dataset after the topology lifting procedure. We recall that:

- the simplicial clique complex lifting is applied to lift the graph to a simplicial domain, with a maximum complex dimension equal to 2;

- the cellular cycle-based lifting is employed to lift the graph into the cellular domain, with maximum complex dimension set to 2 as well.

Table 3: Descriptive summaries of the datasets used in the experiments.

| Dataset | Domain | # 0-cell | # 1-cell | # 2-cell |
|---|---|---|---|---|
| Cora | Cellular | 2,708 | 5,278 | 2,648 |
| | Simplicial | 2,708 | 5,278 | 1,630 |
| Citeseer | Cellular | 3,327 | 4,552 | 1,663 |
| | Simplicial | 3,327 | 4,552 | 1,167 |
| PubMed | Cellular | 19,717 | 44,324 | 23,605 |
| | Simplicial | 19,717 | 44,324 | 12,520 |
| MUTAG | Cellular | 3,371 | 3,721 | 538 |
| | Simplicial | 3,371 | 3,721 | 0 |
| NCI1 | Cellular | 122,747 | 132,753 | 14,885 |
| | Simplicial | 122,747 | 132,753 | 186 |
| NCI109 | Cellular | 122,494 | 132,604 | 15,042 |
| | Simplicial | 122,494 | 132,604 | 183 |
| PROTEINS | Cellular | 43,471 | 81,044 | 38,773 |
| | Simplicial | 43,471 | 81,044 | 30,501 |
| ZINC (subset) | Cellular | 277,864 | 298,985 | 33,121 |
| | Simplicial | 277,864 | 298,985 | 769 |

DATASET SELECTION AND LIMITATIONS

The datasets employed in this work and other TDL studies are predominantly adapted from the GNN literature. Among these, molecular datasets stand out due to the inherent importance of cycles and

hyperedges, which effectively capture chemical rings and functional groups. These are structures that are naturally represented in topological domains.

While TDL methods are not intrinsically constrained to these datasets, the lifting procedures used to construct higher-order cells introduce computational bottlenecks, particularly in memory usage. For instance, operations such as cycle detection and clique enumeration, required for constructing cellular complexes or simplicial complexes, respectively, become computationally prohibitive for large or densely connected graphs.

To address these limitations, ongoing research is focused on developing scalable lifting procedures that can extend TDL methods to broader datasets, including those with more complex structures or larger scales. For example, Bernárdez et al. (2024) propose innovative topological liftings, paving the way for more scalable and applicable datasets in TDL.

### E.3 HYPERPARAMETER SEARCH

Five splits are generated for each dataset to ensure a fair evaluation of the models across domains. Each split comprises 50% training data, 25% validation data, and 25% test data. An exception is made for the ZINC dataset, where predefined splits are used (Irwin et al., 2012).

To avoid the combinatorial explosion of possible hyperparameter sets, we fix the values of all hyperparameters beyond GCCNs: hence, to name a few relevant parameters, we set the learning rate to $0.01$, the batch size to the default value of TopoBenchmark for each dataset, and the cell hidden state dimension to $32$. Regarding the internal GCCN hyperparameters, a grid-search strategy is employed to find the optimal set for each model and dataset. Specifically, we consider 10 different neighborhood structures (see Section E.1), and the number of GCCN layers is varied over $\{2, 4, 8\}$. For GNN-based neighborhood message functions, we vary over $\{$GCN,GAT,GIN,GraphSage$\}$ models from PyTorch Geometric, and for each of them consider either 1 or 2 number of layers. For the Transformer-based neighborhood message function (Transformer Encoder model from PyTorch), we vary the number of heads over $\{2, 4\}$, and the feed-forward neural network dimension over $\{64, 128\}$.

For node-level task datasets, validation is conducted after each training epoch, continuing until either the maximum number of epochs is reached or the optimization metric fails to improve for 50 consecutive validation epochs. The minimum number of epochs is set to 50. Conversely, for graph-level tasks, validation is performed every 5 training epochs, with training halting if the performance metric does not improve on the validation set for the last 10 validation epochs. To optimize the models, `torch.optim.Adam` is combined with `torch.optim.lr_scheduler.StepLR` wherein the step size was set to 50 and the gamma value to 0.5. The optimal hyperparameter set is generally selected based on the best average performance over five validation splits. For the ZINC dataset, five different initialization seeds are used to obtain the average performance.

### E.4 HARDWARE

The hyperparameter search is executed on a Linux machine with 256 cores, 1TB of system memory, and 8 NVIDIA A100 GPUs, each with 80GB of GPU memory.

# F MODEL SIZE

We provide details on model size for reported results in Section 6.

Table 4: Model size corresponding to results reported in Table 1.

| Model | Graph-Level Tasks | | | | | Node-Level Tasks | | |
|---|---|---|---|---|---|---|---|---|
| | MUTAG | PROTEINS | NCI1 | NCI109 | ZINC | Cora | Citeseer | PubMed |
| **Cellular** | | | | | | | | |
| CCNN (Best Model on TopoBenchmark) | 334.72K | 101.12K | 63.87K | 17.67K | 88.06K | 451.85K | 1032.84K | 163.72K |
| GCCN $\omega_\mathcal{N}$ = GAT | 15.11K | 46.27K | 68.99K | 49.63K | 39.78K | 341.54K | 1677.32K | 344.83K |
| GCCN $\omega_\mathcal{N}$ = GCN | 45.44K | 45.25K | 65.92K | 30.69K | 29.54K | 801.16K | 1507.59K | 443.91K |
| GCCN $\omega_\mathcal{N}$ = GIN | **63.62K** | 23.49K | 49.03K | **66.79K** | **64.35K** | 669.58K | 1674.25K | 211.97K |
| GCCN $\omega_\mathcal{N}$ = GraphSAGE | 44.42K | 76.99K | **47.49K** | 115.17K | 79.71K | 1195.14K | 741.5K | 640.51K |
| GCCN $\omega_\mathcal{N}$ = Transformer | 112.26K | 78.79K | 82.05K | 115.43K | 317.02K | 249.51K | 468.29K | 331.59K |
| GCCN $\omega_\mathcal{N}$ = Best GNN, 1 Hasse graph | 14.98K | 18.88K | 18.05K | 15.91K | 20.83K | 150.12K | 367.88K | 66.50K |
| **Simplicial** | | | | | | | | |
| CCNN (Best Model on TopoBenchmark) | 398.85K | 10.24K | 131.84K | 135.75K | 617.86K | 144.62K | 737.29K | 134.40K |
| GCCN $\omega_\mathcal{N}$ = GAT | 15.11K | 46.27K | 68.99K | 49.63K | 67.42K | 341.45K | 1677.32K | 344.83K |
| GCCN $\omega_\mathcal{N}$ = GCN | 45.44K | 45.25K | 65.92K | 30.69K | 64.35K | 801.16K | 1507.59K | 443.91K |
| GCCN $\omega_\mathcal{N}$ = GIN | 63.62K | 23.49K | 49.03K | 66.79K | 118.11K | 669.58K | 1674.25K | 211.97K |
| GCCN $\omega_\mathcal{N}$ = GraphSAGE | 44.42K | 76.99K | 47.49K | 115.17K | 147.30K | 1195.14K | **741.51K** | 640.51K |
| GCCN $\omega_\mathcal{N}$ = Transformer | 113.15K | 213.70K | 82.05K | 166.24K | 148.83K | 284.58K | 468.29K | 331.59K |
| GCCN $\omega_\mathcal{N}$ = Best GNN, 1 Hasse graph | 19.07K | 14.66K | 31.11K | 15.91K | 29.54K | 150.12K | 367.88K | 66.50K |
| **Hypergraph** | | | | | | | | |
| CCNN (Best Model on TopoBenchmark) | 84.10K | **14.34K** | 88.19K | 88.32K | 22.53K | **60.26K** | 258.50K | **280.83K** |

Table 5: Model sizes corresponding to results in Table 2.

| Model | MUTAG | PROTEINS | NCI1 | NCI109 | Cora | Citeseer | PubMed |
|---|---|---|---|---|---|---|---|
| **SCCN** | | | | | | | |
| TopoBenchmark | 398.85K | 397.31K | **131.84K** | **135.75K** | 155.88K | 782.34K | **457.99K** |
| 1 Hasse graph / $\mathcal{N}$, $\omega_\mathcal{N}$ = Best(GNN) | **852.74K** | **851.97K** | 248.58K | 291.39K | **159.46K** | 791.56K | 510.47K |
| 1 Hasse graph for $\{\mathcal{N}\}$, $\omega_\mathcal{N}$ = Best(GNN) | 104.32K | 153.09K | 71.17K | 54.85K | 143.66K | **741.51K** | 376.58K |
| **CWN** | | | | | | | |
| TopoBenchmark | 334.72K | **101.12K** | 124.10K | 412.29K | 343.11K | **1754.50K** | **163.72K** |
| 1 Hasse graph / $\mathcal{N}$, $\omega_\mathcal{N}$ = Best(GNN) | **350.46K** | 353.54K | 95.75K | 465.28K | 900.23K | 177.10K | 159.56K |
| 1 Hasse graph for $\{\mathcal{N}\}$, $\omega_\mathcal{N}$ = Best(GNN) | 219.65K | 283.91K | **78.85K** | **264.45K** | **138.95K** | 163.94K | 138.95K |

# G    MODEL TRAINING TIME

We provide training times for all experiments reported on in Section 6. We measure these training times by running each experiment on a single A30 NVIDIA GPU. We note that these times include the on-the-fly graph expansion method, which slows down the model forward proportionally to dataset size. We plan on moving this process into data preprocessing in the future.

Table 6: Model training time (seconds) corresponding to results reported in Table 1.

| Model | Graph-Level Tasks | | | | | Node-Level Tasks | | |
|---|---|---|---|---|---|---|---|---|
| | MUTAG (↑) | PROTEINS (↑) | NCI1 (↑) | NCI109 (↑) | ZINC (↓) | Cora (↑) | Citeseer (↑) | PubMed (↑) |
| Cellular | | | | | | | | |
| CCNN (Best Model on TopoBenchmark) | 100 ± 23 | 132 ± 19 | 238 ± 89 | 254 ± 39 | 228 ± 44 | 75 ± 15 | 57 ± 4.4 | 128 ± 50 |
| GCCN $\omega_{\mathcal{N}}$ = GAT | 80 ± 11 | 64 ± 10 | 778 ± 118 | 486 ± 75 | 3173 ± 954 | 46 ± 3 | 63 ± 1 | 202 ± 22 |
| GCCN $\omega_{\mathcal{N}}$ = GCN | 43 ± 7 | 67 ± 16 | 544 ± 40 | 495 ± 108 | 4013 ± 620 | 46 ± 4 | 65 ± 3 | 149 ± 12 |
| GCCN $\omega_{\mathcal{N}}$ = GIN | **61 ± 18** | 59 ± 18 | 523 ± 119 | **386 ± 76** | **3301 ± 440** | 64 ± 8 | 77 ± 2 | 207 ± 33 |
| GCCN $\omega_{\mathcal{N}}$ = GraphSAGE | 43 ± 12 | 43 ± 3 | **691 ± 80** | 364 ± 102 | 2863 ± 262 | 49 ± 2 | 60 ± 3 | 211 ± 25 |
| GCCN $\omega_{\mathcal{N}}$ = Transformer | 50 ± 19 | 786 ± 147 | 1005 ± 27 | 1484 ± 181 | 15320 ± 5386 | 121 ± 20 | 94 ± 20 | 5459 ± 1374 |
| GCCN $\omega_{\mathcal{N}}$ = Best GNN, 1 Aug. Hasse graph | 33 ± 7 | 70 ± 24 | 451 ± 123 | 441 ± 130 | 3162 ± 340 | 47 ± 5 | 72 ± 6 | 194 ± 35 |
| Simplicial | | | | | | | | |
| CCNN (Best Model on TopoBenchmark) | 123 ± 57 | 104 ± 28 | 172 ± 50 | 183 ± 62 | 178 ± 86 | 143 ± 16 | 75 ± 23 | 114 ± 18 |
| GCCN $\omega_{\mathcal{N}}$ = GAT | 25 ± 5 | 70 ± 17 | 755 ± 158 | 794 ± 151 | 2242 ± 275 | 49 ± 3 | 68 ± 2 | 192 ± 38 |
| GCCN $\omega_{\mathcal{N}}$ = GCN | 40 ± 7 | 138 ± 26 | 548 ± 185 | 603 ± 181 | 2428 ± 833 | 49 ± 5 | 67 ± 2 | 167 ± 22 |
| GCCN $\omega_{\mathcal{N}}$ = GIN | 61 ± 7 | 66 ± 21 | 904 ± 180 | 538 ± 39 | 3603 ± 475 | 71 ± 6 | 77 ± 8 | 210 ± 42 |
| GCCN $\omega_{\mathcal{N}}$ = GraphSAGE | 31 ± 3 | 61 ± 27 | 572 ± 124 | 511 ± 74 | 1721 ± 201 | 51 ± 3 | 74 ± 8 | 221 ± 37 |
| GCCN $\omega_{\mathcal{N}}$ = Transformer | 35 ± 5 | 947 ± 333 | 1386 ± 404 | 1360 ± 410 | 7979 ± 1373 | 146 ± 58 | 77 ± 2 | 5281 ± 827 |
| GCCN $\omega_{\mathcal{N}}$ = Best GNN, 1 Aug. Hasse graph | 25 ± 2 | 78 ± 27 | 598 ± 31 | 312 ± 7 | 2681 ± 910 | 52 ± 4 | 72 ± 8 | 156 ± 16 |
| Hypergraph | | | | | | | | |
| CCNN (Best Model on TopoBenchmark) | 127 ± 48 | **96 ± 20** | 220 ± 74 | 128 ± 49 | 387 ± 105 | 121 ± 38 | 48 ± 1 | 177 ± 71 |

Table 7: Model training times (seconds) corresponding to results in Table 2.

| Model | MUTAG | PROTEINS | NCI1 | NCI109 | Cora | Citeseer | PubMed |
|---|---|---|---|---|---|---|---|
| SCCN Yang et al. (2022) | | | | | | | |
| **Benchmark results Telyatnikov et al. (2024)** | 11 ± 2 | 60 ± 18 | **247 ± 65** | **311 ± 83** | 102 ± 39 | 101 ± 41 | **143 ± 35** |
| GCCN, on ensemble of strictly aug. Hasse graphs *2, dig | **14 ± 1** | **75 ± 8** | 413 ± 120 | 298 ± 15 | **121 ± 2** | 172 ± 6 | 285 ± 20 |
| GCCN, on 1 aug. Hasse graph *2, dig | 5 ± 1 | 59 ± 10 | 283 ± 90 | 217 ± 100 | 110 ± 3 | **166 ± 10** | 376 ± 27 |
| CWN Bodnar et al. (2021a) | | | | | | | |
| **Benchmark results Telyatnikov et al. (2024)** | 11 ± 2 | **43 ± 5** | 240 ± 50 | 252 ± 92 | 54 ± 25 | **52 ± 5** | **119 ± 14** |
| GCCN, on ensemble of strictly aug. Hasse graphs *2, dig | **12 ± 1** | 73 ± 10 | 536 ± 38 | 426 ± 90 | 91 ± 17 | 49 ± 1 | 125 ± 19 |
| GCCN, on 1 aug. Hasse graph *2, dig | 11 ± 1 | 62 ± 11 | **573 ± 107** | **410 ± 64** | **96 ± 2** | 46 ± 1 | 130 ± 20 |

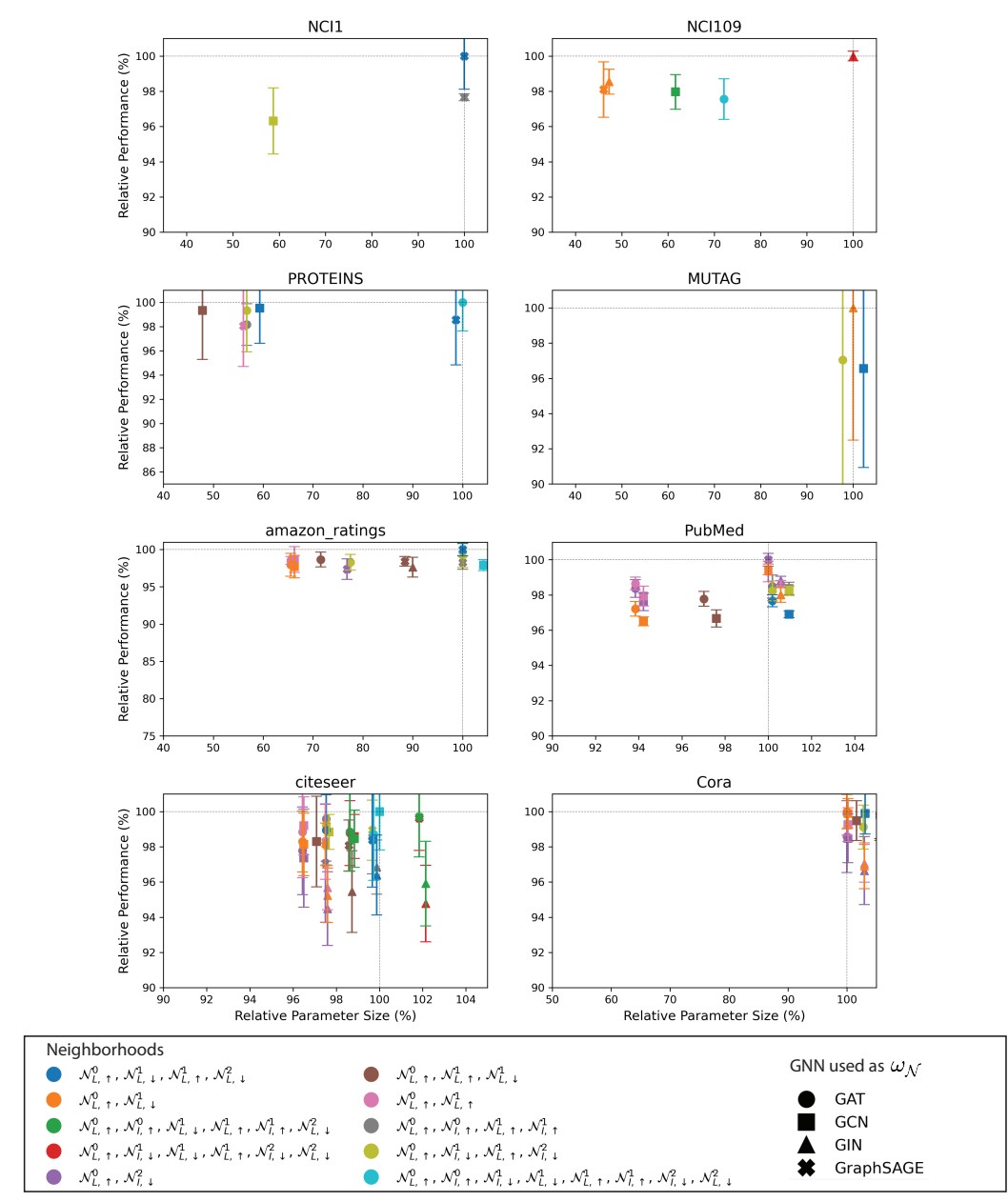

Figure 10: Performance versus size, scaled to best-performing model. The vertical axis range shows models achieving within 10% of the best performance on that dataset.

## H  PERFORMANCE VERSUS SIZE COMPLEXITY

We show the plots similar to Fig. 5 for all datasets. Again here, the best model determines the amount of GCCN layers and GNN sublayers we keep constant.

# I    ADDITIONAL EXPERIMENTS ON LARGER NODE-LEVEL DATASETS

Table 8 additionally presents the experimental results on 4 heterophilic datasets introduced in Platonov et al. (Amazon Ratings, Roman Empire, Minesweeper, and Questions). These represent larger node-level classification tasks than those shown in the main Table 1, with up to 48,921 nodes and 153,540 edges in the case of the Questions graph. Except on this precise dataset, which was not considered in previous TDL literature, we compare the results against CCNNs and hypergraph models from Telyatnikov et al. (2024). We observe that overall GCCNs achieve similar performance than regular CCNNs, and they outperform them by a significant margin on Minesweeper.

|  | **Amazon Ratings** | **Roman Empire** | **Minesweeper** | **Questions** |
| --- | --- | --- | --- | --- |
| Best GCCN Cell | $50.17 \pm 0.71$ | $84.48 \pm 0.29$ | $94.02 \pm 0.28$ | $78.04 \pm 1.34$ |
| Best CCNN Cell | $\mathbf{51.90 \pm 0.15}$ | $82.14 \pm 0.00$ | $89.42 \pm 0.00$ | - |
| Best GCCN Simplicial | $50.53 \pm 0.64$ | $88.24 \pm 0.51$ | $\mathbf{94.06 \pm 0.32}$ | $77.43 \pm 1.33$ |
| Best CCNN Simplicial | OOM | $\mathbf{89.15 \pm 0.32}$ | $90.32 \pm 0.11$ | - |
| Best Hypergraph Model | $50.50 \pm 0.27$ | $81.01 \pm 0.24$ | $84.52 \pm 0.05$ | - |

Table 8: Results on larger node level datasets, each experiment run with 5 seeds. We report accuracy for Amazon Ratings and Roman Empire, and AUC-ROC for Minesweeper and Questions. The values for the best CCNNs and hypergraph models are extracted from TopoBenchmark (Telyatnikov et al., 2024).