# OpenReview forum: "TopoTune: A Framework for Generalized Combinatorial Complex Neural Networks"
_ICLR.cc/2025/Conference — Submitted to ICLR 2025_

### Official Review · Reviewer_kr8N · 2024-10-29

**Soundness:** 2
**Presentation:** 2
**Contribution:** 2
**Rating:** 5
**Confidence:** 3

**Summary:**

The authors tackle the challenge of systematically defining new Topological Deep Learning (TDL) architectures and to enlarge the accessibility of the latter to the broader community. The way they approach this endeavour is by (i) proposing a new class of TDL architectures that generalises previously proposed ones, and by (ii) implementing a software module that encapsulates architectural search over this class.

As for (i), the authors build upon the concepts of “strictly augmented Hasse Graphs” and “Per-rank neighborhoods”. The former ones are employed to model the structure of a combinatorial complex via an ensemble of augmented Hasse graphs, one for each neighbourhood. The latter ones prescribe defining a specific set of neighbourhoods for each rank. The authors propose GCCN as architectures which process ensembles of strictly augmented Hasse graphs with per-rank neighbourhoods with specific neural models and “synchronisation” components.

As for (ii), the module is called TopoTune and is a configuration-oriented component integrated with other TDL frameworks.

Experiments are conducted on graph datasets, lifted to either simplicial or cellular complexes. Results show that GCCNs can outperform standard architectures with a smaller number of parameters or lower computational cost.

**Strengths:**

- The submission tackles an interesting research topic in a timely manner.
- The implemented TopoTune module can be helpful to practitioners and researchers outside of the specific field of TDL.

**Weaknesses:**

- From the perspective of the framework generality, it is not clear how GCCNs would unlock new interesting operations or computational patterns.
    - Eq. 3 and 8 look particularly alike, and it is not evident what kind of advantage the latter brings. In particular, in Eq. 3, the message function $\psi$ can be specific to a particular neighbourhood (and rank), similarly to the neighbourhood message function $\omega$ in Eq. 8 — which, incidentally, is not rank specific.
    - Specific information about ranks and neighbourhoods could be specified by features akin to ”marks” over nodes and edges of an augmented Hasse graph, and a general enough neural architecture could then make use of these for neighbourhood and rank specific updates.
- Proposition 3 appears to be quite trivial given Proposition 1. What is it telling us in addition to that?
- It is not clear how the proposed contributions would help “democratising” TDL, as the authors claim. The proposed approach appears to significantly enlarge the hyper-parameter space by considering a plethora of possible architectural designs arising from the combination of neighbourhood and rank specific neural modules. Although TopoTune lowers the practical effort of searching over these spaces, these large parameter searches may still require large computational capabilities to be satisfactorily performed in a reasonable time frame.
- The value and/or interest of some experimental questions and emerging results is not clear.
    - “GCCNs outperform CCNNs”: It is not clear what the outperformance is due to when comparing to “standard” CCNNs, which could have, potentially, neighbourhood and rank-specific message functions. What is the take-home message for readers?
    - “GCCNs are smaller than CCNNs”: the authors do not explain why this is the case, and it is seemingly the first time this concept emerges in the manuscript
    - “GCCNs improve over existing CCNNs”: the results seem to be merely a matter of additional hyper-parameter search?
    - “Performance-cost tradeoff”: The authors highlight the reduced number of parameters of GCCN models, but they do not expand into how this actually translates into lower computational cost (e.g. because run-time experiments are not discussed in this section).
- Generally speaking, the manuscript would benefit from a clearer and more punctual presentation in regards to the motivations behind the proposed contribution and how these precisely address the research questions put forwards by the authors.

**Questions:**

- Can the authors expand on whether CCNNs can capture GCCNs? Are there functions expressed by GCCNs that cannot be expressed by CCNNs? If the two classes are equivalent, can the authors discuss more in detail what is the effective advantage of considering their proposed GCCN class?
- Can the authors better explain what were the research questions addressed in their experimental section and how their results contribute to answer them?
- Can the authors better discuss how TopoTune goes beyond merely a hyper-parameter search tool?

Please also see weaknesses.

---

> ### Author Response · Authors · 2024-11-20
> **Response to reviewer kr8N**
>
> Thank you for your detailed review and valuable feedback. We believe it has significantly helped to improve our paper. We address your main concerns (weaknesses (W) and questions (Q)) below:
>
> **W1. How GCCNs would unlock new operations and patterns.**
>
> **Bullet 1:** This is an important point. Comparing Equations 3 and 8:
> The collection of neighborhoods considered in CCNNs implicitly exclude per-rank neighborhoods. Specifically, in the definition of CCNNs by Hajij et al 2023 (Eq. 3), the notation "N" refers to the neighborhood N *across all ranks*. This means that, for example, if the incidence neighborhood is considered, all possible 1-hop rank incidences are considered (edges to nodes, faces to edges). It is not possible, for example, to only consider incidence between edges and nodes. GCCNs, on the contrary, are not constrained by this, hence unlocking new possible message-passing paths. This increased flexibility of GCCNs over CCNNs is precisely what allows them to outperform previous architectures. We realize the original text describing Eq. 3 was misleading, thank you for pointing this out.
>
> While $\psi$s in Eq. 3 could potentially be neighborhood- and rank-dependent, by default the vast majority of CCNNs consider all of them to be of the same nature (Papillon et al. 2023). We have clarified the text to reflect this. Moreover, by definition, CCNNs are message-passing architectures, as the function $\psi$ can at most define the message between cells. Contrary to this, nothing forces the $\omega$s of Eq. 8 to be message passing based (e.g., by using a Transformer or MLP architecture). This introduces a completely new landscape of possible TDL models, which have until now been very focused on message passing (see response to Q3 for more).
> We have updated the descriptions of CCNNs (Eq. 3, Section 2) and GCCNs (Eq. 8, Section 4) to further clarify these points.
>
> **Bullet 2:** Regarding the possibility of ‘marking’ node and edge features to specify rank and neighborhood information, it is indeed a valid approach. In fact, it is precisely what simulation works like [1] do (see Section 3, paragraph Retaining expressivity, but not generality). However, the goal of our work is to design a model whose architecture (as opposed to features) naturally incorporates such inductive biases. Indeed, this fulfills TDL’s goal of preserving the topological symmetry of the domain, and requires an inherently different model. To better explain why this idea of incorporating inductive biases goes beyond marking features, we consider an example (now at lines 231-238).
>
> Consider a molecule, represented as a combinatorial complex: bonds are modeled as edges (1-cells) and rings such as carbon rings are modeled as faces (2-cells). Two bonds can simultaneously share multiple neighborhoods. For instance, they could be lower adjacent because they have a common atom (0-cell) and, at the same time, also be upper adjacent because they are part of the same molecular ring (2-cell). Despite their different chemical meaning, the whole Hasse graph (i.e., the approach of [1]) would collapse these two relations (upper adjacent, lower adjacent) into one. Moreover, the resulting GNN would not be able to distinguish anymore which node of the Hasse graph was an atom or a bond or a ring in the original molecule, and would process all the connections with the same set of weights.
>
> Therefore, even if a GNN on the whole augmented Hasse graph of a combinatorial complex is as expressive in a WL sense as a CCNN on the CC, expressivity itself is not enough to employ a GNN rather than a TNN, as the resulting learning models are still inherently very different. In this sense, GCCNs are the first class of models to retain all the properties of [1] while being proper TDL models. In other words, GCCNs still preserve the topological symmetry of the domain.
>
> [1] Jogl et al. “Expressivity-preserving GNN simulation.”
>
> **W2. Stronger expressivity proof.** We agree that Proposition 3 as it stood was not very interesting. We took your feedback to heart and now provide and prove a stronger proposition about expressivity. We show that GCCNs are strictly more expressive than CCNNs, using a higher-order analog to the k-WL test. The full proof is provided in Appendix B3. We provide a brief outline here:
>
> - We define WL-tests related to CCNNs and GCCNs, called CCWL and GCCWL.
>
> - The CCWL test is equivalent to a WL test on a strictly augmented Hasse graph.
>
> - The GCCWL test is at least as powerful as the k-WL test on a strictly augmented Hasse graph, for any k.
>
> - Using the fact that k-WL is more powerful than WL on graphs, we find a pair of two strictly augmented Hasse graphs that cannot be distinguished by WL but can be distinguished by k-WL. This yields two combinatorial complexes that cannot be distinguished by CCWL, but can be distinguished by GCCWL.
>
> - As such, GCCNs are strictly more expressive than CCNNs.
>
> See next reply.

---

> ### Author Response · Authors · 2024-11-20
> **Response to reviewer kr8N (continued)**
>
> **W3. Democratization of TDL.**  Indeed, we believe that our ‘democratization’ claim can be better contextualized. Here are the reasons why TopoTune democratizes TDL.
>
> - **A plug-in approach that removes the need of expert TDL knowledge.** To date, the design of new TDL methods has required a great expertise of the field, as no standardized, base method has been made available to practitioners wishing to apply TDL on their own data. However, by turning these technical steps into a simple component of the architecture –as you noted under “Strengths”-- TopoTune enables both experienced TDL practitioners and newcomers the design of new TDL architectures and message passing pipelines.
>
> - **A plug-in approach that improves upon a given GNN performance.**  TopoTune also democratizes TDL because it relies on GNNs, which are commonly used by deep learning practitioners. The typical use case is as follows. A deep learning researcher is using a GNN model for a given application. This researcher can plug their GNN into TopoTune, which automatically makes it “higher-order” and can potentially improve its performance. You are right that this increases the hyperparameter search space by considering previously fixed traits (topological domain, message function, etc.) as variables. However, we stress that these results with GCCNs were obtained with very minimal “traditional” hyperparameter tuning, keeping parameters like learning rate and embedded dimensions largely fixed across experiments, while the hyperparameters used with CCNNs were the result of an extensive benchmarking study (https://arxiv.org/abs/2406.06642). We now specify this at Section 6.1.
>
> - **A framework that opens the door to a systematic exploration of the TDL landscape.**
> We emphasize that, going beyond a search tool, TopoTune provides an integrated and standardized framework that significantly reduces the implementation overhead associated with TDL. Rather than manually engineering architectures from scratch for each task, users can leverage TopoTune to explore architectural choices within a unified, expressive framework that encapsulates and generalizes previous work. By organizing the vast design space of TDL into a manageable and interpretable framework, TopoTune shifts the challenge from ad-hoc design to systematic exploration, which we believe is critical for advancing the field. We have further clarified this claim in lines 403-405.
>
> **W4. Experimental questions on how GCNNs outperform CCNNs.**
>
> Addressing the different subpoints as follows:
>
> - “GCCNs outperform CCNNs: why?”: This performance is due to a myriad of reasons. First, GCCNs allow for more flexible neighborhood structures (“per-rank” neighborhoods) than CCNNs (see response to W1, Bullet 1). For example, on MUTAG, a cellular GCCN using a per-rank neighborhood significantly outperforms CCNNs. Secondly, GCCNs benefit from the use of $\omega_\mathcal{N}$ modules based on existing, well-studied architectures, rather than relying on the custom, ad hoc message-passing schemes typically implemented in CCNNs. By using existing GNNs in PyTorch Geometric as base $\omega_\mathcal{N}$, for example, GCCNs leverage a much larger body of literature and much more established software. Finally, GCCNs are now shown to be strictly more expressive than CCNNs (see response to W2).
> - “GCCNs are smaller than CCNNs: why?”: After testing many combinations of GCCNs, we found the best performing models to have, in general, small parameter budgets. We found this empirically; it was not intended by design. However, we can hypothesize that this happens because i) GCCNs allow for smaller, more focused neighborhood structures, which we included in our search space and ii) two of the message functions (GCN, GAT) we consider are particularly lightweight, and could introduce less parameters than the ad-hoc message functions of CCNNs.
> - “GCCNs improve over existing CCNNs: a matter of hyperparameter search?”: You are right that GCCNs introduce additional search space by making previously fixed parameters like topological domain and message function variable. However, as discussed in W3 Bullet 2, we stress that all the other potential critical parameters that were optimized while benchmarking CCNNs (learning rate, hidden channels, dropout values, batch size, readout function, etc) were totally fixed in our experiments. We now specify this at lines 420-423. In this way, despite increasing the overall combinatorics, we actually performed a more limited hyperparameter search than TopoBenchmark –yet we obtained better performance. This underlines the methodological contribution of GCCNs as an architecture.
> - “Performance-cost trade-off and running times.” Thank you for bringing up this excellent point. We now provide time complexity analysis in Appendix C as well as training runtimes in Appendix G. Please see our response to all reviewers, points 1A and 1B.
>
> See next reply.

---

> > ### Author Response · Authors · 2024-11-20
> > **Response to reviewer kr8N (continued, part 2)**
> >
> > **W5. Clearer presentation of the motivations.** We refer to point 2B “Contextualizing our contribution” in our main reply to all reviewers. We also refer to our answer to question Q2. At its heart, our work aims to address many open problems in this very young and emerging field. We hope that our previous answers to your raised points W1, W3, and W4 help further inform the value of the contribution, not only as a methodological advance but also as a tool for making the field more accessible and navigable.
> >
> > **Questions**
> >
> > **Q1.** By Proposition 1, proved in Appendix B1, GCCNs formally generalize CCNNs. As such, a CCNN is at most a special case of a GCCN. As a direct consequence, the two classes of architectures are not equivalent. Consider these two examples of how GCCNs go beyond CCNNs:
> > Non-message passing networks (ex.: MLP, Transformer, etc.) cannot be implemented in CCNNs. However, in a GCCN, such models can absolutely be implemented via $\omega_\mathcal{N}$. We provide an example in the paper of this using Transformer. This is a non-message passing architecture that is the building block of this GCCN.
> > CCNNs only explore 1-hop neighborhoods of the Hasse graph in a single layer. GCCNs, on the other hand, can leverage $k$-hop neighborhoods simply by choosing a $\omega_\mathcal{N}$ such as a GNN with $k$ layers. This is the basis of the proof of stronger expressivity provided in our response to W2.
> >
> > If your question about expressed functions refers to universal approximations, this is unfortunately not within the scope of our work. Answering this would require extending universal approximation theory to TDL, which is a whole other line of work that the community has yet to tackle.
> >
> > **Q2.** Our experiment section asks the question: Can GCCNs provide an efficient graph-based alternative to CCNNs that performs just as well or better on many tasks? We believe that Tables 1 and 2 answer that question thoroughly. Our experiment section also provides observations that emerged from the empirical results, such as lottery ticket models and impact of choice of GNN. We have edited these sections for better clarity.
> >
> > **Q3.** We refer to our answer to W3, Bullet 2 as well as W4. We also further elaborate here. While it is true TopoTune provides an easy-to-use hyperparameter search tool, it also advances TDL by opening up a completely new landscape of models. TopoTune defines a rich space of novel architectures through modular components (that are engineered to be easily tunable). These components– like per rank neighborhoods and graph-based, potentially non-message passing submodules (see answers to W1)– enable previously unexplored model classes. These models, GCCNs, are in fact proven (proposition 1, Appendix B1) to generalize existing models (CCNNs). The experimental results obtained with TopoTune demonstrate that these architectural innovations drive performance gains, even though they are obtained without exhaustive training parameter searches (see answer to W3, bullet 2). TopoTune thus represents a fundamental reconceptualization of how topological neural networks can be constructed and composed. We have clarified this under the "Accelerating TDL Research" subsection of Section 5.

---

> > > ### Author Response · Authors · 2024-11-22
> > > **Follow-up on reviewer feedback**
> > >
> > > We are following up on our previous responses to ask if they have been helpful in addressing the weaknesses and questions you laid out. To summarize our responses to these, have:
> > >
> > > - explained how the mathematical definition of GCCNs is different from that of CCNNs and introduces novelty;
> > > - explained how marking edge/node features leads to information loss that a GCCN, on the contrary, avoids;
> > > - provided a stronger expressivity proof showing GCCNs are more powerful than CCNNs
> > > - contextualized our democratization claim with concrete examples;
> > > - explained point by point how GCCNs outperform CCNNs;
> > > - contextualized the contributions within the field's open problems;
> > > - answered questions about generality and how TopoTune is much more than a hyperparameter tool, as it unlocks a whole new landscape of models that are empirically shown, with little to no, training hyperparameter tuning, to perform better.
> > >
> > > We hope that these answers have addressed the Reviewer's concerns, and if there is anything else we can help clarify. Otherwise, in case the Reviewer is satisfied with our response and the clarifications, we'd greatly appreciate it if they could reconsider the rating of our submission.

---

> > > > ### Comment · Reviewer_kr8N · 2024-11-23
> > > > **Answer to the rebuttal 1/2**
> > > >
> > > > I am grateful to the authors for their rebuttal. In the following I will point out related further comments and questions about it.
> > > >
> > > > **W1. How GCCNs would unlock new operations and patterns.**
> > > >
> > > > > This is an important point. […] This increased flexibility of GCCNs over CCNNs is precisely what allows them to outperform previous architectures.
> > > > >
> > > >
> > > > How can the authors be so firm in claiming this architectural modification is *precisely* the reason GCCN outperforms previous architectures? In a follow up response they indeed mention the outperformance can be due to “myriad reasons”. I understand they believe explicitly calling out this additional flexibility to the readers is part of their contributions, but these strong claims requires being substantiated at least by ablation studies in controlled settings.
> > > >
> > > > > […] Contrary to this, nothing forces the ωs of Eq. 8 to be message passing based (e.g., by using a Transformer or MLP architecture). This introduces a completely new landscape of possible TDL models, which have until now been very focused on message passing.
> > > > >
> > > >
> > > > I understand the appeal to seaminglessly introduce transformer layers, but it is also natural to wonder why this is a good idea in the specific TDL use-case. To the best of my understanding, applying fully connected computation via a transformer layer to a strictly augmented Hasse Graph would potentially indiscriminately intermingle representations from cells of different ranks (see e.g. the rightmost depiction in Fig. 3). Why would this be a good idea? Why would it even be relevant to apply Transformers to topological domains? Intuitively this would go in the opposite direction than incorporating “inductive biases” from topological domains as mentioned in the next rebuttal response:
> > > >
> > > > > However, the goal of our work is to design a model whose architecture (as opposed to features) naturally incorporates such inductive biases.
> > > > >
> > > >
> > > > Regarding “marking”, the authors claim that their approach better embodies such inductive biases to respect the “topological symmetry of the domain”. But how are these exactly defined? What do the authors exactly mean by this? Do they refer to “symmetries” as intended in the Geometric Deep Learning and Physics community?
> > > >
> > > > **W2. Stronger expressivity proof**
> > > >
> > > > First, whilst I appreciate the authors’ effort on this, it is due to signal that it constitutes a significant addition, on which I do not believe I can guarantee the same level of attention and depth I provided during the review period.
> > > >
> > > > In any case, I am confused by some statements in the proof. A particularly puzzling point is the claim about the equivalence between the k-CCWL test and the k-dimensional WL test on the strictly augmented Hasse Graph. Why is that the case? It is not obvious to find an intuitive link between the two, especially, because the `k' in the former refers to the number of hops in the neighbourhood, whilst in the latter it refers to the size of tuples whose colours are refined. Generally speaking, deeper neighbourhoods do not obviously enhance discriminative power?
> > > >
> > > > **W3. Democratization of TDL**
> > > >
> > > > > **A plug-in approach that improves upon a given GNN performance** […] This researcher can plug their GNN into TopoTune, which automatically makes it “higher-order” and can potentially improve its performance […]
> > > > >
> > > >
> > > > The claim that TopoTune is “A plug-in approach that improves upon a given GNN performance” can be misleading or incomplete, if anything. In my opinion, the issue with this argument is that it is missing an important ingredient: the “lifting strategy”. I believe this could be a very impactful component in potential improvements obtained when turning a GNN into a higher-order architecture. I agree that a sophisticated search over neighbourhood functions can also contribute, but it is not necessarily the sole responsible for that.
> > > >
> > > > > […] results with GCCNs were obtained with very minimal “traditional” hyperparameter tuning, keeping parameters like learning rate and embedded dimensions largely fixed across experiments, while the hyperparameters used with CCNNs were the result of an extensive benchmarking study.
> > > > >
> > > >
> > > > Can the authors then specify how these “base hyper-parameters” were chosen at the beginning (let me apologise in case I have personally missed this)? The risk is that, if the choice is driven by results obtained from tuning CCNNs, then one should also reason on the fact that prior computation was, somehow, already inherited.
> > > >
> > > > Consistently with the strength I noted, I understand and appreciate that TopoTune can unify interfaces and make TDL exploration more accessible to practitioners, but I believe some claims on its democratising effect are somewhat over-emphasised.

---

> > > > > ### Comment · Reviewer_kr8N · 2024-11-23
> > > > > **Answer to the rebuttal 2/2**
> > > > >
> > > > > **W4. Experimental questions on how GCNNs outperform CCNNs**
> > > > >
> > > > > > “GCCNs outperform CCNNs: why?”: This performance is due to a myriad of reasons. […]
> > > > > >
> > > > >
> > > > > I understand there *could* be many reasons, in principle, but have the authors gathered concrete evidence on some in particular? Or do they believe there is evidence that all of them jointly contribute at the same time as they seem to explicitly claim in their answer? Is there an emerging conclusion they can leave to readers that TopoTune or the GCCN architectural components helped uncover? If the authors believe some components of GCCN are contributing to its performance, in particular, I believe that making these more specific claims could be more informative to readers rather than generally claiming that GCCNs outperform CCNNs.
> > > > >
> > > > > Overall, I am insisting on these points because I am concerned by the way the findings are presented and contextualised. Slightly zooming out a little, but still linking to the next answer the authors provided: some paragraph headings could be misleading. Examples follow. “GCCNs are smaller than CCNNs.” In which sense? Wouldn’t it be more accurate and informative to say that TopoTune helped finding configurations which worked competitively with a smaller number of parameters? “GCCNs improve existing CCNNs.” Would not it be more precise to say that TopoTune allowed searching for better configurations that improved on the original ones? “Lottery Ticket GCCNs” – this heading could be confusing because it refers to a line of research that is completely distinct [1]
> > > > >
> > > > > > dense, randomly-initialized, feed-forward networks contain subnetworks ("winning tickets") that - when trained in isolation - reach test accuracy comparable to the original network in a similar number of iterations
> > > > >
> > > > > [1] The Lottery Ticket Hypothesis: Finding Sparse, Trainable Neural Networks (https://arxiv.org/abs/1803.03635)

---

> > > > > > ### Author Response · Authors · 2024-11-24
> > > > > > **Second response to Reviewer kr8N (1/3)**
> > > > > >
> > > > > > Thank you for considering our response and engaging with it. We really appreciate your continued feedback. We provide responses to points below.
> > > > > >
> > > > > > **W1. How GCCNs would unlock new operations and patterns.**
> > > > > >
> > > > > > - *“Increased flexibility.”* In this response we used the expression of “increased flexibility” to describe the contribution of per-rank neighborhoods that differentiates GCCNs (Eq. 8) from CCNNs (Eq. 3). You are correct that this is not necessarily the singular reason they perform better, but rather one of many – this was poor phrasing on our part. In the paper, we specified this contribution as “increased flexibility over CCNNs” (line 344). In terms of empirically studying the contribution, we refer to our response to “GCCNs outperform CCNNs: why?”.
> > > > > >
> > > > > > - *Transformer.* You are totally correct that using a Transformer message-function leads to a fully-connected computation setting within each strictly augmented Hasse graph, which inherently collapses relationship information at the neighborhood level. Our goal here is to show that GCCNs provide a bridge between traditional TDL architectures and models like Transformers. The simplicity of the bridge we propose does come at the cost of collapsing neighborhood-level relationship information, as would be the case with any non-message passing architecture that doesn’t enforce relationship structure through modified loss, for instance. Solving this issue is an active field of research in the graph domain, with works like [1] proposing a neighborhood contrastive loss to apply MLP on graphs. By seamlessly including Transformer in GCCNs, we hope to make a first step towards generalizing this research to the topological domain. In addition to this, let us also remark that a Transformer for TDL is outlined as an open problem in a recent position paper of the field [2]. While this is not strictly aligned with our research goals –i.e. incorporating meaningful inductive biases in the architectural design, as you pointed out–, TopoTune indirectly provides new possibilities to TDL practitioners interested in this topic. For instance, our solution can easily enable the application of Graphormers [3] to the per-rank neighborhood expansions, which could represent a sweet spot between message-passing TNNs and a pure TDL transformer that collapses all neighborhoods at once.
> > > > > > - *“Topological symmetry”.* Thank you for feedback on clarity – we understand this can be a confusing choice of words. (We have reverted this edit in the main paper). To answer your question, by “preserving the topological symmetry” we mean that the properties of the topological domain are preserved in the same way that the properties of the graph are preserved by GNNs. For example, the architecture is equivariant to permutation of cells within a rank, just as a GNN is equivariant to permutation of nodes. In our previous response to you, we detail exactly how a GCCN goes beyond “marking” features in practice, and what the consequences are in terms of preserving information about the topological structure.
> > > > > >
> > > > > > References
> > > > > >
> > > > > > [1] Hu, Yang, et al. "Graph-MLP: Node classification without message passing in graph." arXiv preprint arXiv:2106.04051 (2021).
> > > > > >
> > > > > > [2] Papamarkou, Theodore, et al. "Position: Topological Deep Learning is the New Frontier for Relational Learning." Forty-first International Conference on Machine Learning. 2024.
> > > > > >
> > > > > > [3] Ying, C, et al. “Do transformers really perform badly for graph representation?.” NeurIPS 2021. Vol 34 p. 28877--28888.
> > > > > >
> > > > > > **W2. Stronger expressivity proof.**
> > > > > >
> > > > > > We are working on improving the proof now, and will respond once more on Monday with an update on this.

---

> > > > > > > ### Author Response · Authors · 2024-11-24
> > > > > > > **Second response to Reviewer kr8N (2/3)**
> > > > > > >
> > > > > > > **W3. Democratization of TDL.**
> > > > > > >
> > > > > > > - *Topological liftings.* We completely agree with you that topological lifting is a very important part of the pipeline. It is in fact what is responsible for assigning higher-order structure to data originating from the graph domain. In fact, a main reason we chose to implement TopoTune inside TopoBenchmark [1] is that this platform hosts many choices of topological liftings and allows for the practitioner to choose the lifting of their choice, just as they would choose a dataset or a model. The authors of this platform even hosted a Topological Deep Learning Challenge at ICML [2] this year which aimed to crowd-source implementations of topological liftings for various purposes. We now specify easy access to topological lifting choice at line 402.
> > > > > > >
> > > > > > > - *Choosing hyperparameters.* Great question! We confirm that our “base hyperparameters” choice was not inherited by the results obtained from tuning regular CCNNs. Instead, we decided to use TopoBenchmark defaults, and for those in which no default was provided (e.g. hidden state dimension) we just selected the lower value that was considered in TopoBenchmark’s grid search. Moreover, it can also be checked in their reproducible scripts [3] that the values of these base hyperparameters (learning rate, hidden state dimension, batch size, readout,...) largely vary across datasets and models. Therefore, we believe that our claim holds, and thanks to your feedback it is better supported in the paper (line 423).
> > > > > > >
> > > > > > > References
> > > > > > >
> > > > > > > [1] Telyatnikov et al. “TopoBenchmarkX: A Framework for Benchmarking Topological Deep Learning.” 2024 https://arxiv.org/pdf/2406.06642
> > > > > > >
> > > > > > > [2] Bernardez et al. “ICML Topological Deep Learning Challenge 2024: Beyond the Graph Domain.” Proceedings of the Geometry-grounded Representation Learning and Generative Modeling Workshop (GRaM) at ICML 2024 https://arxiv.org/abs/2409.05211
> > > > > > >
> > > > > > > [3] https://github.com/geometric-intelligence/TopoBenchmark/blob/main/scripts/reproduce.sh

---

> > > > > > > > ### Author Response · Authors · 2024-11-24
> > > > > > > > **Second response to Reviewer kr8N (3/3)**
> > > > > > > >
> > > > > > > > **W4. Experimental questions on how GCNNs outperform CCNNs**
> > > > > > > >
> > > > > > > > **GCCNs outperform CCNNs: why?**
> > > > > > > >
> > > > > > > > To address this, we summarize in bullet points all of the emerging conclusions that we extract from our experimental results:
> > > > > > > >
> > > > > > > > - Table 1 provides direct evidence that our proposed graph expansion, which leverages many strictly augmented Hasse graphs instead of just one, leads to better performance. (line 464)
> > > > > > > > - Evidence that being able to choose a message-function is helpful appears in:
> > > > > > > >      - Table 2. We reproduce existing CCNNs as GCCNs and vary only the message function. All other parameters are kept fixed. We observe better performance (>1sigma) in 3 datasets for SCCN and in 3 datasets for CWN. (line 484)
> > > > > > > >      - Fig. 5 (and its extended version in Fig. 8). We observe that some datasets like ZINC (line 508) and NCI1 strongly benefit from a specific choice of message-function, in these cases GIN and GraphSage, respectively.
> > > > > > > > - Disentangling the contribution of neighborhood structure towards performance is trickier. While there is a similar parallel to be made with Fig. 5 and 8 – i.e., consider the different choices of neighborhood (color of marker) for a fixed GNN (shape of marker) – it is clear there are many possible options that perform similarly well. In our work, we leverage this observation to remark that some much smaller/more lightweight neighborhood structures lead to similarly performing models, and hence provide a potential solution towards less costly TDL. (line 503)
> > > > > > > >
> > > > > > > > Therefore, even if the overall performance gains in Table 1 rests on a combination of multiple factors, we argue that our evaluation does provide supporting evidence that the various novelties proposed by GCCNs are useful. While we understand this is not as satisfying as a universal statement on how best to implement TDL models given a task, we would like to emphasize that it is the first work of its kind that even considers going beyond one particular choice of message-passing, neighborhood structure, and topological domain. It is only with such a set of results that one could make these emerging observations. We are very excited to continue using TopoTune for application-specific tasks (see future work, line 536) and better understand how to optimize GCCNs in a more focused setting going beyond standard benchmarking. Last, but not least, we truly appreciate your detailed feedback on this critical point; thanks to this ongoing discussion our key findings are much better distilled and contextualized in the revised manuscript.
> > > > > > > >
> > > > > > > > **Paragraph Headings.**
> > > > > > > >
> > > > > > > > Our goal with these paragraph headings was simply to organize the section into short take-home messages. Based on your feedback, we understand that they could be more accurate and descriptive. Here are the new paragraph headings:
> > > > > > > >
> > > > > > > > - “GCCNs are smaller than CCNNs.” becomes “GCCNs perform competitively to CCNNs with less parameters.” Here we would like to stress that GCCNs are a novel, more general class of architectures that go beyond CCNNs, so we kindly avoid referring to them as “configurations that TopoTune has found” (as it could undermine this contribution).
> > > > > > > > - “GCCNs improve existing CCNNs.” becomes “Generalizing existing CCNNs to GCCNs improves performance.” Again, we emphasize that GCCNs are a novel architecture that go beyond a new configuration of previous models.
> > > > > > > > - “Lottery Ticket GCCNs” becomes “TopoTune finds parameter-efficient GCCNs.” We understand that the lottery ticket hypothesis could be a confusing reference here, so both the heading and the paragraph content has been edited to better frame this point.

---

> > > > > > > > > ### Author Response · Authors · 2024-11-26
> > > > > > > > > **Second response to Reviewer kr8N: Proof of expressivity**
> > > > > > > > >
> > > > > > > > > As promised, we are reaching out with an update on the stronger expressivity proof. First, we would like to sincerely thank you for taking the time to review the first version of the proof. We are grateful that you are engaging with our work during this discussion.
> > > > > > > > >
> > > > > > > > > Second, we thank you for catching the oversight on the definition of the neighborhoods Nj(sigma) in the coloring scheme of our k-CCWL test. We meant the j-neighborhood as defined in [Morris et al (2019)] and _not_ the j-hop neighborhood. As you noted, the initialization also needs to be updated to use colors on k-tuples of cells. In fact, we will use k-sets of cells (as opposed to k-tuples of cells) for this proof.  Accordingly, we have updated the coloring scheme of our k-CCWL test correcting both neighborhoods and initialization.
> > > > > > > > >
> > > > > > > > > You are right, deeper neighborhoods do not immediately increase expressivity. However, the higher-order k-GNNs from [Morris et al. 2019] are strictly more expressive than GNNs. As GCNNs can implement k-GNNs on strictly augmented Hasse graphs whereas CCNNs cannot (as the proof shows), GCNNs are strictly more expressive than CCNNs.
> > > > > > > > >
> > > > > > > > > We have added significant details to each step of the proof in the form of auxiliary definitions and propositions as well as references to [Morris et al. 2019], in purple in the updated manuscript. To make the proof easier to follow, particularly given the limited time left in this discussion period and the significant addition it represents, we have added a graphical summary of the definitions and propositions in the new Figure 7.
> > > > > > > > >
> > > > > > > > > Once again, we were truly grateful for your engagement with the first version of our proof. We hope that you will appreciate this refined version. We thank you for the time and effort you are devoting to evaluating our work.
> > > > > > > > >
> > > > > > > > >
> > > > > > > > > Morris, Christopher, Martin Ritzert, Matthias Fey, William L. Hamilton, Jan Eric Lenssen, Gaurav Rattan, and Martin Grohe. "Weisfeiler and leman go neural: Higher-order graph neural networks." In Proceedings of the AAAI conference on artificial intelligence, vol. 33, no. 01, pp. 4602-4609. 2019.
> > > > > > > > >
> > > > > > > > > Grohe, Martin. Descriptive complexity, canonisation, and definable graph structure theory. Vol. 47. Cambridge University Press, 2017.

---

> > > > > > > > > > ### Author Response · Authors · 2024-11-29
> > > > > > > > > > **Follow-up on reviewer feedback (#2)**
> > > > > > > > > >
> > > > > > > > > > Hello reviewer r8N, we are following up about our recent response to your additional questions, which includes:
> > > > > > > > > > - rewordings of the expressions *"increased flexibility"* and *"topological symmetry"* to better explain the advantages of GCCNs;
> > > > > > > > > > - reasoning behind *Transformer* architecture;
> > > > > > > > > > - clarifications and better justifications in the manuscript about democratization claim;
> > > > > > > > > > - list of evidence of empirical results pointing to contribution of GCCNs over CCNNs;
> > > > > > > > > > - reworded paragraph headings in the Results subsection, per your suggestions;
> > > > > > > > > > - updated, stronger expressivity proof with accompanying figure for easy scanning.
> > > > > > > > > >
> > > > > > > > > > We are eager to hear back from you on these points, and address any remaining questions before the upcoming deadline. We believe your feedback has significantly helped improve the manuscript. In light of this, if there are no remaining questions or concerns, we would really appreciate if you considered updating your rating of the paper to reflect this.

---

> > > > > > > > > > > ### Comment · Reviewer_kr8N · 2024-12-01
> > > > > > > > > > > **Second answer to rebuttal 1/2**
> > > > > > > > > > >
> > > > > > > > > > > Thanking the authors for their efforts, I will reply point by point below, and further share additional thoughts.
> > > > > > > > > > >
> > > > > > > > > > > **rewordings of the expressions "increased flexibility" and "topological symmetry" to better explain the advantages of GCCNs**
> > > > > > > > > > >
> > > > > > > > > > > I note the authors' changes and attempt to clarify. I would just signal I have found other points in the current revision which refer to "topological symmetries".
> > > > > > > > > > >
> > > > > > > > > > > I will now reply to the clarifications provided by the reviewers. If I understand correctly, the authors claimed that specific rank and neighbourhood specific encoders is a novel feature of the proposed approach which allows adhering to symmetries in the topological domains such as permutation invariance, contrary to other approaches based, e.g., on marking.
> > > > > > > > > > >
> > > > > > > > > > > I agree that understanding the relative advantage of specific encoders is intriguing and deserves attention, but I still found their claims and arguments not convincing. Some counter-arguments could be: (i) "marking strategies" do not necessarily invalidate these symmetries; (ii) approaches like Transformers, whose application is more easily supported, something underscored by the authors, would blend together representations from cells irrespective of their connectivity structure and rank, seemingly in contrast with the argument that rank-specific modules and neighbourhood structures are a compelling feature; (iii) previous approaches, although less "flexible" can in any case support rank- and neighbourhood specific components, so this does not constitute an inherent architectural limitation that can only be overcome by the proposed architecture.
> > > > > > > > > > >
> > > > > > > > > > > Let me remark: to enquire into the relative gains offered by these architectural patterns is something that could be of interest. However, the authors' claims on the value and novelty of such a methodology are, in my opinion, overemphasised and justified with slightly deceptive arguments.
> > > > > > > > > > >
> > > > > > > > > > > **reasoning behind Transformer architecture**
> > > > > > > > > > >
> > > > > > > > > > > The authors' points are generally reasonable, but these arguments do not seem particularly coherent with others given in the paper for other contributions, and this generates possible confusion.
> > > > > > > > > > >
> > > > > > > > > > > Also, I note that to develop a new transformer architecture for TDL is an open research problem, but the authors do not discuss whether simply plugging transformer models as encoders into their architecture is something justified, or why exactly this would go in the direction of solving the open research problem. This is clearly research out of scope for the present work, and I believe should not significantly weigh in either negatively or positively. As my above comment on the open research problem does not obviously play a role in influencing my evaluation, at the same time I believe the authors' claims on the value of including transformers should not be overemphasised.
> > > > > > > > > > >
> > > > > > > > > > > **clarifications and better justifications in the manuscript about democratization claim**
> > > > > > > > > > >
> > > > > > > > > > > The authors claim researchers "can plug their GNN into TopoTune, which automatically makes it “higher-order” and can potentially improve its performance". Again, I find this argument somewhat confusing – the architecture would use the GNN as an encoder rather than turning into a higher-order method ...? And the ultimate performance of the approach will (also) be determined by the combination of the lifting and the specific choices of the components in the GCCN, other than the specific features of the GNN. What would be the actual conclusion from this a practitioner would gain?
> > > > > > > > > > >
> > > > > > > > > > > So, beyond these arguments, trying to sum up and clarify: eventually a portion of the proposed contribution is simply to provide a software framework that would facilitate experimenting with different relational encoders. Software which, surely could be useful overall, but that, it is also to be noted, has not been provided in the submission yet, or whose software-engineering features, components and design principles have not been illustrated in their detailed specifics (unless I missed that, in which case I apologise).
> > > > > > > > > > >
> > > > > > > > > > > Regarding hyperparameters: "Moreover, it can also be checked in their reproducible scripts [3] that the values of these base hyperparameters (learning rate, hidden state dimension, batch size, readout,...) largely vary across datasets and models." This makes then wonder how they were originally chosen. Do the authors have a pointer for that?
> > > > > > > > > > >
> > > > > > > > > > > It would be reasonable to acknowledge that (better) results were obtained by additionally tuning new specific architectural components. To contextualise: my question and concern is, once again, on the solidity of the authors' arguments, which, in this case are writing "results with GCCNs were obtained with very minimal “traditional” hyperparameter tuning, keeping parameters like learning rate and embedded dimensions largely fixed across experiments, while the hyperparameters used with CCNNs were the result of an extensive benchmarking study (https://arxiv.org/abs/2406.06642)".

---

> > > > > > > > > > > > ### Comment · Reviewer_kr8N · 2024-12-01
> > > > > > > > > > > > **Second answer to rebuttal 2/2**
> > > > > > > > > > > >
> > > > > > > > > > > > **list of evidence of empirical results pointing to contribution of GCCNs over CCNNs** and **reworded paragraph headings in the Results subsection, per your suggestions**
> > > > > > > > > > > >
> > > > > > > > > > > > I note the authors' points and hope that this discussion helps clarifying the contributions of the presented paper.
> > > > > > > > > > > >
> > > > > > > > > > > > **updated, stronger expressivity proof with accompanying figure for easy scanning.**
> > > > > > > > > > > >
> > > > > > > > > > > > After looking into this, I would like to explicitly point out the following. As things stand now in the current revision, the authors are introducing six additional appendix pages with technical definitions and derivations. Importantly, no other reviewer has explicitly acknowledged reviewing this new part. I appreciate the authors' efforts, but I must also share that I do not believe I could (and, to some extent, should) additionally review this new content and alter my evaluations during the current discussion phase based on that. Comments on the soundness, scope, and impact of this additional part are therefore deferred.
> > > > > > > > > > > >
> > > > > > > > > > > > ---
> > > > > > > > > > > >
> > > > > > > > > > > > I will conclude with further comments and questions in respect to addressing some open research questions as per the referenced position paper.
> > > > > > > > > > > >
> > > > > > > > > > > > **Open problem 3**
> > > > > > > > > > > >
> > > > > > > > > > > > The authors of the position paper explain this point as *"Benchmark suites are needed to enable efficient and objective evaluation of novel TDL research"*, and it is clear from the paper they refer to a "minimal collection of higher-order benchmark datasets", "implementations of graph lifting algorithms for generating synthetic datasets, "a taxonomy of higher-order datasets", a "comprehensive set of performance metrics". The authors' contribution can support the action of benchmarking itself, but the claim that it addresses such open problem, that is clearly about data and metrics, is rather far-fetched.
> > > > > > > > > > > >
> > > > > > > > > > > > **Open problem 6**
> > > > > > > > > > > >
> > > > > > > > > > > > Why would be the ability to work across domain a specific feature of GCCNs? Why wouldn’t one be able to apply a standard CCNN to different domains by changing, for example, the lifting function? I do not fully understand why the present contribution would specifically go in the direction of addressing open problem 6.
> > > > > > > > > > > >
> > > > > > > > > > > > **Open problem 9**
> > > > > > > > > > > >
> > > > > > > > > > > > From the original position paper: *”Theoretical foundations have not yet been adequately laid to consolidate the relative advantages of TDL. More theoretical research is needed to shed light on the relevance of topology in deep learning”.*
> > > > > > > > > > > >
> > > > > > > > > > > > The open problem seems to be more about deriving theoretical results on the superiority of “working higher-order / topological” and its general relevance in Deep Learning. Results on the theoretical properties of GCCNs and its comparison with other TDL methods are clearly a welcome contribution, but do not seem to strongly align to address the aforementioned open problem – which is more about the role of TDL in the Deep Learning landscape.
> > > > > > > > > > > >
> > > > > > > > > > > > **Open problem 11**
> > > > > > > > > > > >
> > > > > > > > > > > > I still cannot fully understand: are the authors claiming their model have advantages on building models that could work across topological domains? Why is that? Can they expand a little on this?

---

> ### Comment · Reviewer_kr8N · 2024-12-01
> **Second answer to rebuttal 3/2**
>
> Zooming out a little: I hope my previous comments and questions do not sound gratuitously combative. As an official reviewer, I felt it important to convey my opinion that some claims are somewhat overemphasised, and some arguments and responses have been given in a way I found, in some cases, slightly deceptive.
>
> Other than this, in the above response I had additional comments to share and question to raise, which I invite the authors to concisely respond to.

---

> > ### Author Response · Authors · 2024-12-02
> > **Third response to Reviewer kr8N (1/4)**
> >
> > Thank you for your additional comments. We respond point by point below as concisely as possible.
> >
> > **“Topological symmetry”.** We apologize for the oversight – we have removed the 2 instances we used this term (paragraph heading, introduction), which were both intended to introduce the topic at large rather than delve into it. We introduced this expression during rebuttal as an attempt to respond to Reviewer kr8N’s misunderstandings of GCCNs’ contribution. If the paper is accepted, there will be no mention of it in the camera-ready version.
> >
> > **Claims on value and novelty.**
> >
> > - (i) We think there continues to be a misunderstanding about the fundamental differences between GCCNs and marking-based GNNs. Marking strategies currently proposed by the literature do *necessarily* collapse topological information, as we detailed in our reply entitled “Response to reviewer kr8N” sent Nov 19 and as we explicitly state in the manuscript (original and rebuttal versions) they are *not equivalent* to the CCNNs they seek to emulate.
> > - (ii) This is also not quite right. In GCCNs, Transformer modules only consider cells from one neighborhood at a time, thus avoiding this “blending” between the entire complex. It is true that in the current implementation, the Transformer module does collapse some connectivity, but only *within neighborhoods*, rather than within the whole complex. We mention that this does not perform nearly as well as other choices of $\omega_\mathcal{N}$. We also emphasize that the Transformer-GCCN is meant to be a first step towards non-message-passing models in higher order settings, and not meant to be the main focus of the paper. Indeed, it is only one GCCN parameterization among the many others we consider. We mention the word “Transformer” three times, all only in the context of hyperparameter lists (lines 71,119,418).
> > - (iii) This is not true. Previous architectures are not capable of accommodating per-rank neighborhoods (line 280) and are not constructed with “Lego block” style modules that consider one neighborhood at a time. This novelty is inherently tied to our novel graph expansion mechanism, which is the first to represent a higher-order complex as a collection of strictly augmented Hasse graphs (line 258). As Proposition 1 (line 349) states, GCCNs generalize and subsume previous topological models (CCNNs).
> >
> > >  “However, the authors' claims on the value and novelty of such a methodology are, in my opinion, overemphasised and justified with slightly deceptive arguments.”
> >
> > Our arguments of novelty are supported through a thorough motivation of the work in comparison to existing gaps (Section 3), clear enumeration of novel architectural contributions (lines 258, 280, 309), and proven theoretical statements (line 347) on generality, equivariance, and expressivity. Moreover, we refer to our reply to the Reviewer’s remarks (i,ii,iii) above.
> >
> > **reasoning behind Transformer architecture**
> >
> > We agree that the Transformer-based GCCN architecture differs from the GNN-based GCCNs. We also agree that further developing non message-passing architectures is out of the scope of this work. We agree that this contribution should not be overemphasized, and stress that it is in fact not emphasized anywhere in the paper, beyond a short remark at line 371 that GCCNs imply a possibility for defining non message-passing architectures.

---

> > > ### Author Response · Authors · 2024-12-02
> > > **Third response to Reviewer kr8N (2/4)**
> > >
> > > **clarifications and better justifications in the manuscript about democratization claim**
> > >
> > > There seems to be a big misunderstanding here. Our method **does indeed make any given GNN “higher-order”**, i.e., it produces a model that takes into account higher-order interactions (the neighborhoods) among groups of more than two nodes (the cells) and process them using the same methodological principles of the underlying GNN. This said, the resulting architecture is clearly a product of the lifting, neighborhoods, and GNN choice. However, implicitly using Proposition 1, the reviewer has to agree that most of the architectures in the TDL literature can be recovered using TopoTune with the corresponding graph model. For instance, with appropriate choices of liftings and neighborhoods:
> > > - the Simplicial Complex Convolutional Neural Network [1] can be obtained from TopoTune by setting the GNN to a 1-layer Graph Convolutional Neural Network
> > > - the Generalized Simplicial Attention Networks [2] can be obtained from TopoTune by setting the GNN to a 1-layer Graph Attention Network.
> > >
> > > These constructive examples should make clear that TopoTune represents a significant resource for practitioners in TDL and across diverse related fields. Practitioners can now easily test and benchmark the performance of any GNN on the relational structure induced by an underlying topological domain. Practitioners can also test a new TDL model that can be implemented using a GNN on the set of strictly augmented Hasse graphs. For this reason, we chose TopoBenchmark as the host platform for TopoTune because it allows easy access to the widest available library of topological liftings, some of which are application specific (see white paper on topological lifting challenge organized by TopoBenchmark’s developers [3]).
> > >
> > > **Software**
> > >
> > > Unfortunately, due to anonymity concerns we could not include the link to the open-source repository. (This was stated in the original manuscript but we had to remove it to accommodate rebuttal-related modifications.)
> > > In terms of software features/components/design:
> > > - At line 400, we summarize the tools offered by TopoBenchmark and their benefits for practitioners.
> > > - At line 431, we reference pseudo-code in Appendix D for how a GCCN model forward runs, going from graph expansion to message-passing to aggregation.
> > > - In terms of the broader software, we refer to the manuscript associated with TopoBenchmark [4], which details the design and operation of this extensive platform for lifting/processing data, implementing models, and training them.
> > > - A central focus of our software tool is ease of use and customization. With no better way to prove this due to anonymity and the supplementary materials being frozen, we emphasize that we spent a significant amount of time organizing and thoroughly documenting TopoTune (every function documented with a docstring and parameters description in numpy documentation format) inside TopoBenchmark, as well as providing a descriptive set of instructions on getting started, customizing models, and reproducing experiments. We use ruff as a linting tool that ensures that our software coding style respects software engineering best practices (the associated linting Github Action passes). We use pytest as a unit-testing tool that automatically tests our code so that, together, 88% of TopoBenchmark and TopoTune’s code are unit-tested.
> > > - Going beyond our own software development, an important contribution of the paper is making existing GNN software readily available for Topological Deep Learning (line 395, 537), making much more established software libraries like PyTorch Geometric available to use on topological domains.
> > >
> > > References
> > >
> > > [1] Yang, et al. "Simplicial convolutional neural networks." ICASSP 2022-2022 IEEE International Conference on Acoustics, Speech and Signal Processing (ICASSP). IEEE, 2022.
> > >
> > > [2] Battiloro, et al. "Generalized simplicial attention neural networks." IEEE Transactions on Signal and Information Processing over Networks (2024).
> > >
> > > [3] Bernardez et al. “ICML Topological Deep Learning Challenge 2024: Beyond the Graph Domain.” Proceedings of the Geometry-grounded Representation Learning and Generative Modeling Workshop (GRaM) at ICML 2024 https://arxiv.org/abs/2409.05211
> > >
> > > [4] Telyatnikov et al. “TopoBenchmarkX: A Framework for Benchmarking Topological Deep Learning.” 2024 https://arxiv.org/pdf/2406.06642

---

> > > > ### Author Response · Authors · 2024-12-02
> > > > **Third response to Reviewer kr8N (3/4)**
> > > >
> > > > **Hyperparameters**
> > > >
> > > > According to Appendix C.2 of the TopoBenchmark manuscript [1], TopoBenchmark developers performed a wide grid search across multiple training hyperparameters for each model and dataset. They published the best hyperparameters for each combination in their scripts. In the code, they offer a default configuration that automatically determines training hyperparameters if not customized. We used these “default hyperparameters”, thus avoiding a traditional tuning grid search. In the spirit of increasing clarity as much as possible, line 422 now reads: “While CCNN results reflect extensive hyperparameter tuning by [1], we fix GCCN training hyperparameters using the TopoBenchmark default configuration.” We argue that a practitioner could reasonably do the same thing and avoid a tuning grid search when introducing new choices of GNN and neighborhood structure.
> > > >
> > > > [1] Bernardez et al. “ICML Topological Deep Learning Challenge 2024: Beyond the Graph Domain.” Proceedings of the Geometry-grounded Representation Learning and Generative Modeling Workshop (GRaM) at ICML 2024 https://arxiv.org/abs/2409.05211
> > > >
> > > >
> > > > **updated, stronger expressivity proof with accompanying figure for easy scanning.**
> > > >
> > > > We completely understand this is a lot to review, and we really appreciate the feedback Reviewer kr8N has provided so far. These additional pages (we assume the 6 pages refer to Appendix B3) are in direct response to Reviewer kr8N’s initial feedback about how the expressivity proof statement could be more interesting. We took the initiative to add the accompanying Fig. 7 to convey the information concisely in one spot. Everything we have added has been in direct response to reviewer feedback.
> > > >
> > > > **Open problem 3**
> > > >
> > > > We agree that our work is not about a “dataset benchmark suite”. However, we argue that by building our work directly into TopoBenchmark, the only available benchmarking platform (developed after this position paper came out), we make benchmarking an integral goal and capability of our novel TDL research. Up until now, TDL research has considered one architecture at a time and benchmark datasets have been widely heterogeneous in nature and processing/lifting methodology. By building a tool that can efficiently and objectively evaluate many, many novel models against multiple benchmark datasets that are homogeneously preprocessed/lifted, we argue that we do speak to this open problem.
> > > >
> > > >
> > > > **Open problem 6**
> > > >
> > > > Existing CCNNs only consider one topological domain at a time, as demonstrated in a literature review of the field [1] which organizes models by topological domain for which they were developed. The right column of Table 1 of this review shows that each CCNN is tested either by comparison to a GNN or by comparison to a CCNN of the same topological domain. To our knowledge, GCCNs are the first models to be tested across many domains at publication time. This was made easy due to the engineering of TopoBenchmark that allows for a practitioner to choose a lifting separately from a model. We hope this will pave the way for future TDL research to continue addressing problem 6: standardized implementations divorced from topological lifting, rather than ad-hoc, one-off implementations that include a singular lifting.
> > > >
> > > > [1] Papillon et al. “Architectures of Topological Deep Learning: A Survey of Message-Passing Topological Neural Networks.” 2024. https://arxiv.org/abs/2304.10031
> > > >
> > > > **Open problem 9**
> > > >
> > > > We agree that this open problem focuses on the theoretical advantages of TDL at large. We do not claim to solve this problem, but rather address it through our theoretical contributions which provide general and comparable theoretically grounded inductive biases. By introducing and proving Propositions 1 (generality), 2 (permutation equivariance), and 3 (expressivity) on the combinatorial complex domain--the most general topological domain--, we believe we consolidate TDL advantages in one location and one language. Importantly, since this language is largely based on the GNN theory language (WL tests, permutation equivariance, and so on), we aim for this new, generalized theory of TDL to be as comparable as possible in considering the deep learning landscape. This is the first instance of WL tests (see Appendix B.3) on the combinatorial complex domain.

---

> > > > > ### Author Response · Authors · 2024-12-02
> > > > > **Third response to Reviewer kr8N (4/4)**
> > > > >
> > > > > **Open problem 11**
> > > > >
> > > > > We refer to our response above to Open problem 6. Aditionnally, we specify that GCCNs are advantaged by the facts that
> > > > > - i) their theory is built upon combinatorial complexes (most general domain), making them eligible for any special subcase domain (we note that this theoretically also applies to combinatorial CCNNs, although, in practice, no testing across domains has been performed)
> > > > > - ii) their implementation is built into a platform where models and liftings are divorced, and there are many liftings to choose from.
> > > > >
> > > > >
> > > > > > As an official reviewer, I felt it important to convey my opinion that some claims are somewhat overemphasised, and some arguments and responses have been given in a way I found, in some cases, slightly deceptive.
> > > > >
> > > > > We thank Reviewer kr8N for the detailed response and continued engagement. We appreciate the care they have devoted to our work, and we hope that this response will help clear up the remaining misunderstandings that seem to shape their opinion of the novelty and value of the work.
> > > > >
> > > > > We are more than happy to further clarify any claims through responses or in the paper (in camera ready version), as we have absolutely no intentions of deception. Our answers to the Reviewer’s specific questions should not be confused with the content of the paper which does not highlight nearly as strongly many of the topics discussed here.

---

> > > > > > ### Comment · Reviewer_kr8N · 2024-12-03
> > > > > > **Third answer to rebuttal**
> > > > > >
> > > > > > Thank you. I will leave further comments and questions you could reply to in the time left.
> > > > > >
> > > > > > **Claims on value and novelty.**
> > > > > >
> > > > > > (i) My argument was referring to the symmetries you mentioned in your response, e.g. permutation invariance of cells. Marking strategies do not necessarily invalidate that. As for maintaining a strict hierarchical structure that is more an inductive bias than a symmetry.
> > > > > >
> > > > > > (ii) In the case one chooses an incidence neighbourhood, the strictly augmented Hasse graph would contain cells of different ranks, unless I missed something. See e.g. Fig. 3. Wouldn't a Transformer encoder applied to that intermingle their representations? I hope your answer to this will eventually clear up any outstanding confusion on this.
> > > > > >
> > > > > > (iii) What I meant with this is that nothing prevents one from using a per-rank update function in, say, a CWN msg-passing layer (https://arxiv.org/pdf/2106.12575), regardless of the fact that this is not common practice. At this point, at the same time, such an update function could learn to assign different weights to different messages, and, theoretically, potentially disregard some of them.
> > > > > >
> > > > > > **Open Problems 6, 11**
> > > > > >
> > > > > > Again, why wouldn’t one be able to apply a standard CCNN to different domains by changing, for example, the lifting function? I understand that an easier infrastructure to do that would be helpful for the community, but, again, I am not sure I understand why this is something possible only with your proposed architecture and not with others.

---

> > > > > > > ### Author Response · Authors · 2024-12-03
> > > > > > > **Fourth response to Reviewer kr8N**
> > > > > > >
> > > > > > > Firstly we would like to thank you very much for increasing your rating of the paper. We believe our conversations during this rebuttal have led to many important improvements and we really appreciate that you agree. In response to the remaining questions:
> > > > > > >
> > > > > > > **Claims on value and novelty**
> > > > > > >
> > > > > > > - (i) Thank you for clarifying. To our knowledge, no works using marking strategies have a proven statement on cell permutation invariance or equivariance. They only focus on preserving expressivity.
> > > > > > > - (ii) That is correct. In the case of an incidence neighborhood, the input Hasse graph would be a fully-connected graph between both ranks involved. So yes, in this case there would be “blending” at the level of these two ranks. A way to avoid this would be to exclude such neighborhoods or implement a contrastive loss to partially “de-blend” on the backend. Thank you for the pointer – we  hope this has cleared up the confusion.
> > > > > > > - (iii) Both the message-passing equations and software infrastructure of CCNNs standardly do not support per-rank neighborhoods or per-rank components at large, like update functions. To our knowledge, this is the first work that provides the theoretical and practical implementations that support these. While we agree that adding per-rank capability is a relatively straightforward generalization, we believe its simplicity should not diminish the value it brings as a true generalization.
> > > > > > >
> > > > > > > **Open problems 6,11**
> > > > > > >
> > > > > > > It is true that one could take, for example, a simplicial complex neural network and, if it is implemented in TopoBenchmark, use it along with a cellular complex lifting. However, similarly to (iii) this is not done in CCNN literature, in part because authors run experiments that match their theoretical frameworks, which are often developed for a specific sub-domain (ex.: expressivity in the cellular domain with cellular-level WL tests).
> > > > > > >
> > > > > > > We argue that our theoretical framework at the combinatorial level (including combinatorial-level WL tests), combined with our implementation that provides seamless access to many liftings across different domains, represents a fundamentally distinct approach compared to domain-specific CCNNs.

---

### Official Review · Reviewer_iHAm · 2024-11-02

**Soundness:** 3
**Presentation:** 3
**Contribution:** 2
**Rating:** 6
**Confidence:** 4

**Summary:**

In this work, the authors propose a generalization of Combinatorial Complex Neural Networks (CCNNs) called GCCNs and an accompanying software library called TopoTune, to generalize works on CCNNs into one computational framework and streamline the training and tuning of TDL architectures. Both theoretical and empirical results indicate that the proposed framework is indeed a useful generalization of previous efforts in TDL.

**Strengths:**

- The main strength of this work is that the authors are able to subsume TDL architectures under a single framework.
- The empirical results indicate to me that the framework matches existing works, thus validating the claim that the framework is indeed general.
- The framework allows the use of GNNs, which should bring the two fields closer together and have TDL research benefit from progress in GNNs.

**Weaknesses:**

- L458: The authors state that “GCCNs outperform CCNNs”. Out of the 8 presented datasets, I can only find two instances (NCI1, ZINC) where GCCNs actually perform better than the best CCNN baseline (accounting for one standard deviation). I could be convinced that the benefit of TopoTune is that one must only sweep over the GNN sub-modules to obtain an (at least) on-par model. However, this would still require some effort to find the best sub-module; see question 3 for more on this.
- In L468 and Figure 5, the authors discuss performance vs. number of parameters. However, I don not find this comparison convincing as a smaller number of parameters may not necessarily be more cost-efficient. Instead, I would like to see a comparison in terms of runtime and memory usage of the different models.
- Since the authors argue their approach to be superior to works on higher-order GNNs, a comparison of GCCNs and higher-order GNNs would be very useful. For example, PPGN++ (https://arxiv.org/abs/1905.11136), a higher-order GNN, performs much more on par with the best GCCN on ZINC than most CCNN baselines presented in the paper.

**Questions:**

- In the introduction you say “However, constrained by the pairwise nature of graphs, GNNs are limited in their ability to capture and model higher-order interactions […]”. I would expect that higher-order GNNs (https://arxiv.org/abs/1905.11136, https://arxiv.org/abs/1810.02244, https://arxiv.org/abs/1905.11136) are able to capture higher-order interactions. Could you elaborate on how TDL differs from higher-order GNNs?
- Related to the first question, in L88-L93 you mention the work of Jogl et al. (https://openreview.net/forum?id=HKUxAE-J6lq) on Cell Encodings, which is equivalent to using the standard Weisfeiler-Leman test on a transformed graph, but your argument for the shortcomings of this approach is not clear to me. In particular, you state that “However, although these architectures over the resulting graph-expanded representations are as expressive as their TDL counterparts […] the former are neither formally equivalent to nor a generalization of the latter”. What is “the former”? What is “the latter”? Assuming the former are Cell Encodings and the latter topological GNNs, why is it important that they are formally equivalent or one being a generalization of the other? Are they different in their runtime or memory requirements? Do we expect better learning behavior from TDL methods?
- As outlined in the weaknesses, in Table 1, only on two datasets GCCNs outperform the best CCNN from TopoBenchmarkX. Can you further elaborate on the benefits of TopoTune in this context?
- Related to the third question, can the authors provide an overview over the runtime and memory complexity of the compared CCNNs, as well as GCCNs, possibly in relation to the complexity the underlying GNN submodules?
- Am I correctly assuming that the ZINC dataset used in this work is the full ZINC dataset with 250K graphs, rather than the ZINC (12K) version frequently benchmarked in graph learning?

---

> ### Author Response · Authors · 2024-11-20
> **Response to reviewer iHAm**
>
> Thank you very much for your time and your very helpful feedback. We address your points about weaknesses (W) and questions (Q) below.
>
> **W1. Justifying the summary statement “GCCNs outperform CCNNs” and contextualizing TopoTune’s benefits.**
> It is true that GCCNs only outperform outside 1\sigma on 2 datasets across all domains. However, when considering inter-domain performance, as has been traditionally the case in TDL, GCCNs outperform (beyond 1\sigma) the counterpart best CCNN on 11 out of 16 of the domain/dataset combinations tested. We now specify this at Section 6.2, paragraph GCCNs outperform CCNNs.
>
> Moreover, we stress that these results with GCCNs were obtained with very minimal “traditional” hyperparameter tuning, keeping params like learning rate and embedded dimensions largely fixed across experiments, while the hyperparameters used with CCNNs were the result of an extensive benchmarking study (https://arxiv.org/abs/2406.06642). We have clarified this point in Section 6.1.
> More broadly, we would like to emphasize how powerful these results are in the context of how much easier and simpler TopoTune makes TDL. Each of the CCNNs we benchmark against is the product of a painstaking, one-at-a-time generalization of a specific choice of model to a specific choice of domain, always with a “from scratch” message-passing scheme. In contrast, each GCCN that achieves comparable performance is the result of a structured and systematic exploration of domain, neighborhood, GNN, and related parameters facilitated by TopoTune.
> While it is true that selecting the best GNN sub-modules or parameter combinations requires some effort, TopoTune provides an integrated and standardized framework that significantly reduces the overhead associated with this process. Rather than manually engineering architectures from scratch for each task, users can leverage TopoTune to explore architectural choices within a unified, expressive framework that encapsulates and generalizes previous work. This approach not only accelerates the design process but also ensures consistency and comparability across models. As shown in Fig. 5, TopoTune also provides insight into how design choices—such as the number of parameters, neighborhood configurations, or GNN submodules—affect performance, offering a more principled way to navigate and refine architectural decisions.
> TLDR: By organizing the vast design space of TDL into a manageable and interpretable framework, TopoTune shifts the challenge from ad-hoc design to systematic exploration, which we believe is critical for advancing the field.
>
> **W2. Comparison in runtimes and memory usage.**
> For runtime, we point to bullet point “Training times” in our global answer to all reviewers.
> Due to the large amount of experiments and necessity for distributed training, reporting used memory beyond the number of parameters was not possible in this timeframe.
>
> **W3. Comparison with higher-order GNNs.** This is a very interesting point, thank you for bringing this work to our attention. Indeed, these models perform very well on graph benchmark tasks.
>
> First, let us directly answer your question about how works like PPGN++ are different from TDL, and specifically GCCNs. The main difference between higher-order GNNs and Topological Neural Networks (TNNs) lies at a very fundamental level.
>
> GNNs run on graphs, which are simple topological spaces.  GNNs have often been studied through the lens of Geometric Deep Learning (GDL). GDL (in the sense of [1]) is built on group-theoretic arguments along with the frequent usage of Hilbert Spaces (strictly related to manifold learning and, in general, to metric spaces).
>
> TNNs run on combinatorial topological spaces that are inherently higher-order. TNNs have been studied through the lens of Topological Deep Learning (TDL). TDL is solely built on the modeling assumption of data living on the neighborhoods of a combinatorial topological space and having a relational structure induced by the neighborhoods’ overlap. Further insights can be gained from the thesis in [2]-[3].
>
> This said, higher-order GNNs still run on graphs. As such, they usually use higher-order information to update node embeddings. In contrast, in TNNs (GCCNs included), all the cell embeddings are updated, as each cell is an element of the underlying space. This is not a detail. Indeed, while higher-order information is usually used in GNNs to achieve some desirable property (e.g., improved expressivity) but without any other additional theoretical argument, TNN usually results in improved expressivity while being supported by the sophisticated and powerful machinery of algebraic topology. This fact allows us to leverage additional structure (e.g., spectral theory, homology theory, homotopy theory,...) when combinatorial complexes are particularized to simplicial, cell, or path complexes [4][5][6][7].
>
> See next reply.

---

> > ### Author Response · Authors · 2024-11-20
> > **Response to reviewer iHAm (continued)**
> >
> > **W3, continued**. Finally, on a more practical side, the fact that each cell is an entity in the underlying space allows us to select which higher-order interactions matter rather than relying on arbitrary structures induced by the k-hop neighborhoods of nodes. For instance, let’s consider a molecule to be our data. In the cellular domain, carbon rings of the molecule are directly incorporated as standalone features of the dataset. In a k-hop setting, this is not possible.
> >
> > We believe this reply is also useful to partly clarify that two equally expressive architectures are not equivalent and that expressivity itself is not an exhaustive metric, as we explain more in detail in our reply to Q2.
> >
> > Nevertheless, this very exciting progress in the GNN field (higher-order or not) speaks directly to TopoTune’s goal of making such advances more accessible to TDL.  For example, it could be interesting to use PPGN++ as an $\omega_\mathcal{N}$ message-function (instead of the fairly vanilla GNNs used in the paper) and perform k-hop learning on the topological domain. Another important future research direction with TopoTune is using richer lighting mechanisms that better capture dataset topology (rather than adhering to strict theoretical domains like simplicial and cellular complexes), and thus potentially better highlight the contribution of TDL as a whole.
> >
> > References.
> >
> > [1] Bronstein, M., et al. "Geometric deep learning: Grids, groups, graphs, geodesics, and gauges."
> >
> > [2] Bodnar, C, “Topological Deep Learning: Graphs, Complexes, Sheaves”
> >
> > [3] Battiloro, C, “Signal Processing and Learning over Topological Spaces”
> >
> > [4] Yang, et al. "Simplicial convolutional neural networks."
> >
> > [5] Battiloro, C, et al. "Generalized simplicial attention neural networks."
> >
> > [6] Barbarossa, S, et al. "Topological signal processing over simplicial complexes."
> >
> > [7] Roddenberry, T. M, et al. "Signal processing on cell complexes."
> >
> >
> > **Answers to questions**
> >
> > **Q1.** Answered in W3.
> >
> > **Q2.** Yes, in that sentence the former refers to Cell Encodings, and the latter to CCNNs –apologies for the complex formulation. We have rewritten that sentence in the paper to improve clarity. Additionally, we have expanded that paragraph to address the motivation behind our claim (Section 3, paragraph Retaining expressivity, but not generality). Indeed, as you pointed out, it lacked proper contextualization.
> >
> > When we say that a GNN running on the expanded graph is neither formally equivalent to nor a generalization of its TDL counterparts we straightforwardly mean that they are not the same model and one should not be used as a surrogate of the other. Indeed, a GNN running on the expanded graph does not take into account either the different ranks or the different neighborhoods, resulting in a rank- and neighborhood-independent message function. This collapses many relations induced by the topological domain and applies the same set of weights to the connections that survive the collapse.
> >
> > Take again a molecule as an example, represented as a combinatorial complex: bonds are modeled as edges (1-cells) and rings such as carbon rings are modeled as faces (2-cells). Two bonds can simultaneously share multiple neighborhoods. Two bonds can simultaneously share multiple neighborhoods. For instance, they could be lower adjacent because they have a common atom (0-cell) and, at the same time, also be upper adjacent because they are part of the same molecular ring (2-cell). Despite their different chemical meaning, the whole Hasse graph (i.e., the approach of [1]) would collapse these two relations (upper adjacent, lower adjacent) into one. Moreover, the resulting GNN would not be able to distinguish anymore which node of the Hasse graph was an atom or a bond or a ring in the original molecule, and would process all the connections with the same set of weights.
> >
> > Therefore, even if a GNN on the whole augmented Hasse graph of a combinatorial complex is as expressive in a WL sense as a CCNN on the CC, expressivity itself is not enough to employ a GNN rather than a TNN, as the resulting learning models are still inherently very different. In this sense, GCCNs are the first class of models to retain all the properties of [1] while being proper TDL models.
> >
> > References.
> > [1] Jogl et al. “Expressivity-preserving GNN simulation.”
> >
> > **Q3.** Answered in W1.
> >
> > **Q4.** We refer to reply 1A in our main reply to all reviewers. Memory wise, due to the large amount of experiments and necessity for distributed training, reporting memory complexity beyond the number of parameters was not possible in this timeframe, but would absolutely be important in future work.
> >
> > **Q5.** We’ve clarified this at Section 6.2—thank you for bringing it to our attention. We are using the 12K-graph subset of the ZINC dataset, as further detailed in Table 3, Appendix E2. We chose the subset version because it is most commonly used in graph (and TDL) benchmarking tasks.

---

> > > ### Author Response · Authors · 2024-11-22
> > > **Follow-up for feedback**
> > >
> > > We would like to ask you whether our responses have been helpful in addressing your raised points raised about weaknesses and questions. To summarize our response, have:
> > >
> > > - better contextualized the performance contribution and TopoTune's utility as a whole;
> > > - provided runtimes;
> > > - explained how higher-order GNNs differ from Topological Neural Networks, including GCCNs
> > > - answered questions about the inherent differences with GNNs beyond expressivity
> > >
> > > We would greatly appreciate hearing back from you on these points, as it would allow us to address any remaining issues and further improve the quality of our manuscript. In the case where this response has been helpful, we would kindly ask you reconsider the rating of the paper.

---

> > > > ### Comment · Reviewer_iHAm · 2024-11-23
> > > >
> > > > First of all, I'd like to thank the authors for their extensive answers to my concerns and questions, the latter of which have been answered adequately. I specifically thank the authors for their time to explain in great detail the differences in modeling assumptions between higher-order GNNs and TNNs. Now, to the weaknesses:
> > > >
> > > > * **Benefits of GCCNs vs. CCNNs**: I am largely convinced that your framework unifies many approaches in TDL and hence allows for an easier process for building models in practice. However, regarding the hyper-parameter tuning argument that you employ, **how did you select the hyper-parameters for GCCNs, if not by classical hyper-parameter tuning?**. Perhaps this is stated in your paper and I may have missed it. In any way, I would appreciate if the authors could elaborate on this.
> > > >
> > > > * **Time complexity and runtimes**: I believe this concern is largely addressed. I have just a small follow-up question: You mention that the on-the-fly graph expansion could be instead implemented as a pre-processing step. Do I understand correctly that your current implementation runs the graph expansion for the same data point repeatedly at each training epoch, causing the higher runtime?
> > > >
> > > > * **Comparison between higher-order GNNs and TNNs**: I just now realized that the best performance on ZINC reported in your paper is 0.19 MAE. Is the evaluation procedure here somehow different from that of standard GNNs? I am asking because 0.19 MAE on ZINC would be considered very poor performance in the graph learning setting; see e.g., the PPGN++ paper as an example, where even a simple GIN model surpasses 0.19 MAE. I would appreciate if authors could comment on this. So far I was under the impression that TDL methods perform very strongly on ZINC, maybe this was an oversight on my part.
> > > > Anyway, more generally, while I understand that higher-order GNNs and TNNs use different modeling assumptions, I still think that a comparison between them would be interesting, both to draw more attention to the potential benefits of TDL, as well as to ensure that TDL methods can actually push the state-of-the-art on real-world problems. Considering the dataset choices made in this work and also other TDL works: **Can the authors explain why these datasets are typically used to benchmark TDL methods? Would it, in principle, be possible to apply TDL methods to arbitrary graph datasets to enable a fair comparison between GNNs and TNNs?**

---

> ### Author Response · Authors · 2024-11-24
> **Response to Reviewer iHAm (#2)**
>
> We are happy to hear we were able to address many of your questions! Thank you for engaging with our response. We respond to your additional points below:
>
> **W1. Benefits of GCCNs vs CCNNs.**
> Great question! We chose our hyperparameters using TopoBenchmark defaults, and for those in which no default was provided (e.g. hidden state dimension) we just selected the lower value that was considered in TopoBenchmark’s original grid search, which is recorded in [1]. We now specify this in the paper (line 423); thank you for your feedback.
>
> [1] https://github.com/geometric-intelligence/TopoBenchmark/blob/main/scripts/reproduce.sh
>
> **W2. Time complexity and runtimes.**
> Yes, that’s exactly right. We currently project combinatorial complexes onto strictly augmented Hasse graphs directly inside the forward function of the model, instead of pre-saving the expanded dataset. This introduces higher runtime grossly proportional to dataset size, which explains why we are comparable/faster on smaller datasets and slower on larger datasets.
>
> **W3.  Comparison between higher-order GNNs and TNNs.**
> - We begin by addressing your questions specific to ZINC.
> In terms of the literature, the best reported pure GNN performance on ZINC without edge features is 0.320 MAE using PNA [1]. With edge features (representing bond types), this improves to 0.188 MAE. The best TDL model on ZINC, CIN [2], achieves 0.079 MAE, leveraging edge features. From what we can see, PPGN++ does not include ZINC results in its evaluation.
> In terms of our results, we remark that the current TopoBenchmark implementation disregards edge features. As highlighted in [1], edge features significantly influence performance. This omission likely accounts for the observed performance gap in our results compared to state-of-the-art models.
> You are correct that TDL models generally outperform GNNs on ZINC. We appreciate your observation and will explore incorporating edge features into TopoBenchmark.
>
> - We now address your point about higher-order GNNs.
> We agree that a direct comparison between higher-order GNNs and TDL methods would be valuable, both for highlighting the benefits of TDL and evaluating its potential for real-world problems. Unfortunately, such a comparison belongs outside the scope of this paper, which aims to first and foremost generalize and make it easy for TDL methods to incorporate arbitrary layers. That being said, we absolutely plan to incorporate higher-order GNNs as baselines in future work and help expand the TopoBenchmark framework accordingly.
>
> - To answer your question about choice of dataset: The datasets used in this and other TDL works are largely inherited from the GNN literature. Molecular datasets, in particular, are appealing for TDL due to the significance of cycles and hyperedges in representing chemical rings and functional groups. TDL is not inherently limited to these datasets. However, the lifting procedures used to generate higher-order cells impose computational constraints, specifically on memory. Current implementations (e.g., finding cycles or cliques) do not scale well with large, densely connected graphs. There is ongoing research to design scalable lifting procedures that would enable TDL methods to generalize to broader datasets. For instance, Bernardez et al. [3] propose innovative approaches to extend TDL beyond the graph domain.
>
> All of these aspects have been clarified in the revised manuscript: we provide further details on dataset selection (line 425), excluding edge features(line 430), higher-order GNNs as future work (line 539), and dataset selection + lifting limitations (Appendix E2).
>
> References
>
> [1] Corso, Gabriele, et al. "Principal neighbourhood aggregation for graph nets." Advances in Neural Information Processing Systems 33 (2020): 13260-13271.
>
> [2] Bodnar, Cristian, et al. "Weisfeiler and lehman go cellular: Cw networks." Advances in neural information processing systems 34 (2021): 2625-2640.
>
> [3] Bernardez et al. “ICML Topological Deep Learning Challenge 2024: Beyond the Graph Domain.” Proceedings of the Geometry-grounded Representation Learning and Generative Modeling Workshop (GRaM) at ICML 2024 https://arxiv.org/abs/2409.05211

---

> > ### Comment · Reviewer_iHAm · 2024-11-25
> >
> > Thank you for your response.
> >
> > > We chose our hyperparameters using TopoBenchmark defaults
> > >
> >
> > Doesn't this mean that hyperparameter search is still required, even if you ended up using the same hyperparameters for each GCCN model? Have you compared GCCN performance with the performance of CCNNs using the TopoBenchmark hyperparameter defaults?
> >
> > > From what we can see, PPGN++ does not include ZINC results in its evaluation.
> > >
> >
> > I am referring to Table 3 in https://arxiv.org/abs/2302.11556. If I am not mistaken, PPGN++ uses edge features. Can you confirm that the ZINC (12K) is the same as used in your evaluation?
> >
> > Overall, I am happy to see that you are actively adjusting the manuscript to incorporate the review feedback. I think that this will improve the quality of your work.

---

> > > ### Author Response · Authors · 2024-11-25
> > > **Response to Reviewer iHAm (#3)**
> > >
> > > Thank you for continuing to engage! Your feedback continues to help improve the manuscript and we are happy to hear you agree. To answer your questions:
> > >
> > > - *Hyperparameters.* We did not do a traditional training hyperparameter search in the sense that we only considered one set of training hyperparameters for each combination of GCCN and task. This set of hyperparameters was selected from the defaults proposed by TopoBenchmark. If there was no default available, we picked the lowest value considered in Top Benchmarks reported grid search. As such, we compared :
> > >      - GCCN performance obtained with a combination of TopoBenchmark defaults/smallest-grid-search-value
> > >      - with
> > >      - CCNN performance obtained with TopoBenchmark tuned hyperparameters.
> > > Please let us know if that is more clear!
> > > - *ZINC.* Thank you for specifying the Table, we were looking at the other PPGN++ paper (https://arxiv.org/pdf/1905.11136). Indeed, it does use edge features (p. 14, Appendix A1). Yes, we use the same ZINC-12k in our evaluation.
> > >
> > > Thank you again for your valuable feedback. We kindly ask that you consider updating your score to reflect the improvements your feedback has informed.

---

> ### Author Response · Authors · 2024-11-26
> **Follow-up for Feedback (#2)**
>
> Hello! We are following up to ask if our previous response has addressed your remaining questions about the selected training hyperparameters and the nature of the ZINC dataset. We would appreciate being able to incorporate any additional feedback into the manuscript before the deadline tomorrow.
> In the event you have found our responses to be satisfactory, we would really appreciate an updated rating. We believe your feedback has significantly helped improve the manuscript. Thank you!

---

> > ### Author Response · Authors · 2024-11-29
> > **Follow up for Feedback (#3)**
> >
> > Hello, happy Friday! We are following up once more to ask if our previous response addressed remaining questions about hyperparameters and ZINC. We would love to answer any additional questions by the deadline. As we said in our previous follow-up, we believe your feedback has significantly helped improve the manuscript. As such, in the event you do not have any questions left, we would really appreciate an updated rating that reflects these improvements.

---

> > > ### Author Response · Authors · 2024-12-02
> > > **Follow-up for Feedback (#4)**
> > >
> > > Hello Reviewer iHAm. We are following up once more to ask if our previous response was satisfactory in answering the two remaining questions about hyperparameters and ZINC. Your feedback has helped inform many improvements, including **runtimes, better contextualization of the contribution, and clarifications to the text.** We kindly ask that if the Reviewer has no remaining questions, they consider updating the rating of the paper. We would really appreciate the acknowledgement.

---

> > > ### Comment · Reviewer_iHAm · 2024-12-02
> > >
> > > Regarding ZINC, I think this is largely addressed. In GDL it is more common to use edge features but what is important for this work is that the comparison between TDL models is fair, which seems to be the case.
> > >
> > > Regarding the hyperparameters, I definitely see the authors' point in that they did not need specialized hyperparameters but I want to emphasize that the authors still needed to rely on the hyperparameter defaults of TopoBenchmark and hence, especially for new tasks, a hyperparameter tuning may still be necessary.
> > >
> > > That being said, I don't think that this is a central issue of this work. Further, I appreciate the many improvements the authors have made in the manuscript. Hence, I decided to raise my score.

---

> > > > ### Author Response · Authors · 2024-12-02
> > > > **Thank you!**
> > > >
> > > > We are happy to hear that we answered your remaining questions. Absolutely for ZINC, excluding edge features helps with comparisons. For hyperparameters, you are correct that tuning a new GNN could require extra tuning. That said, in our experiments, we used the same defaults across four separate GNNs (which TopoBenchmark did not include in their own grid search) and got good results.
> > > >
> > > > Thank you very much for increasing your score. We believe our manuscript is now better because of your feedback and are very happy you agree.

---

### Official Review · Reviewer_rJiq · 2024-11-03

**Soundness:** 4
**Presentation:** 4
**Contribution:** 2
**Rating:** 8
**Confidence:** 3

**Summary:**

The authors propose a general topological deep learning (TDL) architecture called Generalized Combinatorial Complex Network (GCCN). It aims to unify prior work on TDL under a common mathematical framework.
Additionally, the authors provide the TopoTune library, a reusable software implementation of the proposed GCCN method.
The experiments show that the flexibility of the GCCN framework allows it to match or outperform previously proposed TDL methods while, oftentimes, requiring fewer model parameters to do so.

**Strengths:**

First, the proposed GCCN architecture (while fairly straight-forward) provides a useful framework for describing a large variety of TDL methods and it enlarges the design space for such methods.
The experiments illustrate how this simplifies the optimization of TDL models and improving upon the state-of-the-art.
Additionally, the authors show that GCCN can match or even outperform previously proposed approaches while requiring fewer parameters to do so.

Second, the provided TopoTune implementation of GCCN integrates with existing GNN and TDL libraries.
This simplifies the exploration of novel TDL architectures and, as stated by the authors, could help accelerate research on TDL.
However, since I am not deeply familiar with the current literature on TDL and open problems, I can not confidently assess the relevance of this contribution.

Last, I want to highlight the presentation. The paper is well structured and written. The figures are of high quality and helpful.

**Weaknesses:**

In Section 4 the authors show a number of theoretical properties of their proposed GCCN framework.
While certainly desirable, the value of those properties is limited.
As stated by the authors themselves in the proofs in the supplement, those properties are, for the most part, fairly straight-forward.
As far as I can tell, the GCCN framework is an intuitive generalization of prior work which only provides relatively small theoretical insights.
The overall value of the contribution therefore seems to depend on the relevance of the previously described strengths of the paper, in particular, on the relevance of the provided TopoTune implementation.
However, as mentioned, I cannot fully assess this aspect.
Thus, one potential general concern might be the overall relevance of the paper.

Apart from this point I have only minor suggestions for improvement:
1. I would have found a (brief) explanation of the evaluated types of combinatorial complexes (cellular vs simplicial) to be helpful.
2. There seem to be two small errors in the formal definitions in Section 2:
	- p. 3 (127): At $\mathcal{P}(S) \setminus \{\emptyset\}$ it should probably read $\mathcal{V}$ instead of $S$.
	- p. 3, eq. 2 (146): $\mathrm{rk}(\tau)$ after $\exists\ \delta$ should be probably $\mathrm{rk}(\delta)$.

**Questions:**

1. In Figure 5, it did not become entirely clear to me why the parameter size is reduced by changing the neighborhoods. I would expect that the total number of parameters of the GNN modules are independent of the specific types of neighborhood used. However, as shown in Figure 5 this does not appear to be the case. Can you elaborate on what exactly you mean by parameter size and how it relates the the choice of neighborhoods?
2. It is not clear to me how exactly the GCCN models are parameterized in the different experiments. In particular, which intra- and inter-neighborhood aggregators were used for the different experiments?
3. In the conclusion, you state that you hope that TopoTune might help "bridge the gap with other machine learning fields". Apart from the connection GNNs (and possibly Transformer models), are there any specific fields you envision that might profit from such a connection?

---

> ### Author Response · Authors · 2024-11-20
> **Response to reviewer rJiq**
>
> We wish to express deep gratitude for your time and your thoughtful review. We are happy to read that the figures were helpful. We address the raised points about weaknesses and questions below.
>
> **W1 Contextualizing the contribution for TDL newcomers.** (Response 2B in main reply to all reviewers above). We completely understand that TDL might be an unfamiliar field, and appreciate your feedback about better contextualizing TopoTune’s contributions with respect to the field’s open problems. As such, we have baked into the paper’s stated contributions and conclusions a stronger connection with a position paper authored by many of the field’s leaders (https://arxiv.org/abs/2106.04051) which defines 11 open problems. This work fully or partially tackles 7 of these problems, which we elaborate upon in our second main reply to all reviewers above.
>
> **W2 Clarifying the simplicial vs. cellular domains.** We have incorporated in Appendix A a brief overview of the different topological domains of TDL. This addition is now referenced in Section 2 Background, and helps make this work more self-contained.
>
> **Math typos.** Thank you for pointing these out –we have fixed both.
>
>  **Questions**
>
> 1. Since a GCCN is made up of message-passing blocks, each individually assigned to a given neighborhood, modifying the amount of neighborhoods necessarily modifies the amount of message-passing blocks, which in turn modifies parameter size. You are correct that each GNN independently does not change in size (whether it be assigned to node-level adjacency, or face-to-edge incidence, for example), but the amount of GNNs used certainly does affect total size. We have also made sure to better clarify this in the paper in Section 6.2.
>
>
> 2. Each GCCN model is parametrized by choice of neighborhood structure, choice of topological domain, choice of message function (i.e. choice of GNN to use as building blocks), and choice of graph expansion method (we consider both ensembles of strictly augmented graphs and one augmented Hasse graph). For the purposes of this work, we only consider one set of intra- and inter- neighborhood aggregations, just to reduce the scope of comparisons already introduced by the previously mentioned parameters. Specifically, intra-neighborhood aggregation is left up to the choice of GNN, and inter-neighborhood aggregation is a sum. Considering and comparing various aggregators would be an interesting future research direction.
>
>
> 3. Architecture wise, this opens the door to newer advances in the graph learning field, going beyond “standard” GNNs, such as graph-based MLP models (https://arxiv.org/abs/2106.04051), and diffusion models (https://arxiv.org/abs/2106.10934). Application wise, we envision TopoTune to be a tool for newcomers from the various fields which have already been shown to benefit from learning on higher order spaces, as outlined in Appendix B of TDL’s most recent position paper (https://arxiv.org/abs/2106.04051). These include: data compression, natural language processing, computer vision, chemistry, virus evolution, and more. Much of this interdisciplinary work is early and its future success is in part deeply tied to the accessibility of TDL, something we hope TopoTune will accelerate.

---

> > ### Author Response · Authors · 2024-11-22
> > **Follow-up on Reviewer Feedback**
> >
> > We would like to ask you whether our response addressed your concerns, weaknesses, and questions so far. Also, we’d like to know whether you have any other questions? To summarize our first response, have:
> > - better contextualized the contribution in the context of the field;
> > - clarified domain definitions;
> > - fixed typos;
> > - provided clarity on model size, parametrization, and applications.
> >
> > We would greatly appreciate a prompt feedback, as it would allow us to clarify any remaining issues and further improve the quality of our manuscript.

---

> > > ### Comment · Reviewer_rJiq · 2024-11-26
> > >
> > > **Re Q1:** Thank you for clarifying the cause of the parameter size reductions in Fig 5. I am however still not quite sure I fully understand the reported differences. On the ZINC dataset you compared three GIN-based GCCN variants (orange: three per-rank neighborhoods, dark green: two neighborhoods, gray: three neighborhoods). With one $\omega_{\mathcal{N}}$ block per neighborhood $\mathcal{N}$, I would expect the dark green variant to have the lowest parameter count. I assume that the difference is due the parameter count varying for per-rank and "cross-rank" neighborhoods. Is this assumption correct and, if yes, how do the GNN modules vary between different types of neighborhoods?
> > >
> > > **Re Q2:** Thank you! This fully answers my question. I agree that it would indeed be interesting to consider other inter-neighborhood aggregators as well.
> > >
> > > **Re Q3:** Thank you for providing additional references. The added context (re W1) and how your work relates to open problems in the field was also helpful.
> > >
> > > Apart from the minor clarification request regarding the parameterization of the GNN modules, I have no further questions.

---

> ### Author Response · Authors · 2024-11-26
> **Reponse to Reviewer rJiq (#2)**
>
> We are happy to hear that our answers were helpful! Addressing the additional question about GNN module parameterization:
>
> - *Small clarification 1.* On the ZINC dataset we in fact compared every single possible combination of GNN and neighborhood structure listed in lines 415–418. Figure 5 only shows three of these combinations because of the plot’s range, which only shows models that performed within 10% of the best model and within 40–100% of the best model’s parameter size. On ZINC, there are only 3 models, all GIN-based, who belong in this range.
> - *Small clarification 2.* The neighborhood notation in Fig. 5 is not an accurate reflection of neighborhood size. The key here is that while the orange neighborhood is written in terms of per-rank neighborhoods (notation includes rank superscripts, see example at line 283), the green and gray neighborhoods are written in terms of regular, rank-independent neighborhoods (notation does not include rank superscript, see example at line 303).  We were hoping to write out these neighborhoods in terms of their per-rank notation, but for some reason Open Review is having a very hard time displaying the math -- so sorry about this. We write them as clearly as possible in text:
>
> {N_{A, \uparrow}, N_{A, \downarrow}} = {N_{A, \uparrow}^0, N_{A, \uparrow}^1, N_{A, \downarrow}^1, N_{A, \downarrow}^2}
>
> N_{A, \uparrow}, N_{A, \downarrow}, N_{I, \downarrow}\} =  N_{A, \uparrow}^0, N_{A, \uparrow}^1, N_{A, \downarrow}^1, N_{A, \downarrow}^2, N_{I, \downarrow}^1, N_{I, \downarrow}^2
>
> We completely understand this could be a confusing choice of notation. To better clarify this distinction but still avoid having space issues, we have added to Fig. 5’s legend a symbol indicating which neighborhoods can only be expressed as per-rank, and mentioned it in the caption (line 526).
>
> Thank you again for your continued feedback. We appreciate engaging with you and improving the paper through your input.

---

> > ### Comment · Reviewer_rJiq · 2024-11-26
> >
> > Thank you! This fully answers my question. I agree that a brief explanation of the notation would help to avoid confusion.

---

> > > ### Author Response · Authors · 2024-11-26
> > >
> > > Thank you for your patience! We were struggling with MathJax. The updated manuscript with updated Fig. 5 and caption is now uploaded.
> > >
> > > Please let us know if you have any more questions or feedback that could help improve the manuscript before the deadline on Wednesday!

---

### Official Review · Reviewer_TxXe · 2024-11-07

**Soundness:** 3
**Presentation:** 3
**Contribution:** 3
**Rating:** 6
**Confidence:** 3

**Summary:**

The paper focuses on the topological deep learning (TDL) models in particular CCNNs and proposes a new powerful graph-based methodology for new TDL architectures, named GCCNs. The paper proves that GCCNs generalize and subsume CCNNs. The paper conducts extensive experiments and shows that the GCCN architectures achieve comparable performance with CCNNs. An efficient toolkit, TopoTune, is also introduced to accelerate the development of TDL models.

**Strengths:**

1. The paper proposes a new method to generalize any neural network to TDL architectures.
2. The proposed GCCNs formally generalize CCNNs and have the same expressiveness as CCNNs.
3. A new toolkit, TopoTune, has been developed to make it easy to design and implement GCCNs.

**Weaknesses:**

1. For node-level tasks, the paper only considers three very small datasets, which might limit the application of the method.
2. The complexity analysis of the method is missing and the paper does not report any training time in the experiment.
3. The experiment of "performance versus size" is not well analyzed especially for the graph-level datasets (i.e., PROTEINS, ZINC).

**Questions:**

1. Could the authors use larger node-level datasets for experiments?
2. What is the time complexity of the proposed GCCNs compared with CCNNs?
3. The GNN models perform very different results in Figure 5. More analysis is needed.

---

> ### Author Response · Authors · 2024-11-20
> **Response to reviewer TxXe**
>
> Thank you for taking the time to provide such valuable feedback, we truly appreciate it.
> Let us address each of the points you raised in Weaknesses:
>
> **W1 Test larger datasets on node-level tasks.** We now provide additional experiments on 4 larger node-level benchmark datasets (Amazon Ratings, Roman Empire, Minesweeper, Questions) that our machine can support memory-wise –please see results in Table 8 (Appendix I), as well as below. Specifically, all TDL models are constrained by the currently available topological liftings, as large graph-based datasets significantly increase in size in the process. (Note: developing better, more lightweight liftings is a current field of research, see https://arxiv.org/pdf/2305.16174). As an example, Tolokers dataset (11,758 nodes, 519,000 edges) raises OOM issues when storing either cliques (simplicial) or cycles (cell) due to its extremely dense connectivity, as was reported in Table 1 of (https://arxiv.org/pdf/2406.06642). We have added Questions dataset (48,921 nodes, 153,540 edges), not evaluated in TopoBenchmark, to show that our GCCNs are applicable as long as the lifting procedures are feasible. Regarding the results on the other three datasets, we observe that GCCNs achieve similar performance than regular CCNNs, outperforming them by a significant margin on Minesweeper.
>
> |                       | Amazon Ratings | Roman Empire | Minesweeper  |   Questions  |
> |-----------------------|----------------|--------------|--------------|--------------|
> | Best GCCN Cell        |  50.17 ± 0.71  | 84.48 ± 0.29 | 94.02 ± 0.28 | 78.04 ± 1.34 |
> | Best CCNN Cell        |  51.90 ± 0.15  | 82.14 ± 0.00 | 89.42 ± 0.00 |       -      |
> | Best GCCN Simplicial  |  50.53 ± 0.64  | 88.24 ± 0.51 | 94.06 ± 0.32 | 77.43 ± 1.33 |
> | Best CCNN Simplicial  |       OOM      | 89.15 ± 0.32 | 90.32 ± 0.11 |       -      |
> | Best Hypergraph Model |  50.50 ± 0.27  | 81.01 ± 0.24 | 84.52 ± 0.05 |       -      |
>
>
>
> **W2 Give time complexity and training times.** We refer to response 1B in the main reply to all reviewers above.
>
>
> **W3 Analyze performance vs size.** (2A in the main reply). We have made sure to better clarify this contribution in Section 6.2 by separating into two renamed subsections: “Lottery Ticket GCCNs” and “Impactfulness of GNN choice is dataset specific.” There is indeed large variance in performance (and size) between choices of GNN. Such a variation in performance between GNNs is to be expected, as some message-passing functions are better suited to certain tasks/datasets, as has been studied in the GNN field through extensive benchmarking (see for example https://arxiv.org/abs/2003.00982). In the context of Topological Deep Learning, we are interested in understanding how these differences in performance couple with choice of neighborhood, and how we can optimize these hyperparameters for maximal performance at minimal cost. We also kindly refer to Figure 7 which includes performance versus size results on all tested datasets.
>
> Figure 5 specifically aims to show “lottery-ticket” models with high performance and low parameter cost (ie, size). This figure only considers models performing within 10% of the best model. In the case of ZINC, both the best performing model and the within-10% models use GraphSAGE. On the other hand, PROTEINS and Citeseer achieve within-10% performance with models using GAT, GCN, and GraphSAGE. This indicates that for some choices of dataset/task, the choice of message-passing function is much more consequential than in other cases. Both sets of results in PROTEINS and Citeseer show that cutting down on parameter complexity can be as simple as choosing a lighter message-passing function (such as GAT or GCN) while keeping choice of neighborhood constant. For example, both in the PROTEINS and Citeseer cases, choosing GAT over GIN or GraphSAGE leads to equivalent performance for a given choice of neighborhood structure (purple).
>
> If you have additional ideas on how to deepen this analysis, we would be happy to hear them during the rest of the discussion period.
>
> **Questions**
> For Q1, we refer to point 1 above. For Q2 we refer to point 2 above. For Q3 we refer to point 3 above.

---

> > ### Author Response · Authors · 2024-11-22
> > **Follow-up on Reviewer Feedback**
> >
> > We would like to follow up to ask if our response addresses the reviewer’s concerns, weaknesses, and questions. To summarize our response, we have :
> > - added results on 4 large node-level tasks;
> > - provided time complexity;
> > - provided runtimes;
> > - re-contextualized performance analysis.
> >
> > We would greatly appreciate a prompt feedback, as it would allow us to clarify any remaining issues and further improve the quality of our manuscript. In case the Reviewer is satisfied with our response and the clarifications, we would kindly ask them to reconsider the rating of our submission.

---

> > > ### Author Response · Authors · 2024-11-25
> > > **Follow-up on Reviewer Feedback (#2)**
> > >
> > > We are following up once again to ask if the Reviewer has any additional questions. We would greatly appreciate to hear back, such that we can address any remaining issues before the deadline.
> > >
> > > If the Reviewer does not have any additional concerns since we have answered their questions, edited the manuscript, and added additional experiments, we would kindly ask that they consider updating the rating of the submission to reflect this.

---

> > > > ### Author Response · Authors · 2024-11-26
> > > > **Follow-up on Reviewer Feedback (#3)**
> > > >
> > > > We are hoping to hear back from the Reviewer before the deadline for making edits to the manuscript on Wednesday. We would love to know if our previous response has addressed all concerns before this deadline. Otherwise, if the Reviewer is satisfied, we kindly ask they reconsider the rating of the paper. Thank you!

---

> ### Author Response · Authors · 2024-11-29
> **Follow-up on Reviewer Feedback (#4)**
>
> Hello Reviewer TxXe. We are following up a fourth time to ask if our rebuttal has addressed your concerns. Your input has informed many new improvements to the paper including **adding new experiments, runtimes, time complexity, and clarifications to the main text**. We would really, really appreciate hearing back from you about your thoughts on this. That way, if there are remaining issues, we’d be happy to clarify before the deadline.
>
> Otherwise, if you’re satisfied, we kindly ask that you reconsider the rating of our updated submission.

---

> > ### Author Response · Authors · 2024-12-02
> > **Follow-up on Reviewer Feedback (#5)**
> >
> > Hello again. We would really appreciate if Reviewer TxXe would reply to our rebuttal, and ask any remaining questions, if any. We repeat that your feedback has informed **adding new experiments, runtimes, time complexity, and clarifications to the main text**. We would really appreciate an updated review that reflects these improvements asked for by Reviewer TxXe.

---

> > > ### Author Response · Authors · 2024-12-03
> > > **Follow-up on Reviewer Feedback (#6)**
> > >
> > > We are once again following up to ask that you please consider replying to our response, or, if you have no further concerns or questions, consider updating your rating of the paper. It is really important to us and to the peer review system that we engage with you. The deadline is coming up fast, so please consider responding soon.

---

### Author Response · Authors · 2024-11-20
**Response to all reviewers**

We thank the reviewers for their time and thoughtful comments. Reviewers found our work to be a meaningful and timely contribution to the field of Topological Deep Learning (TDL), which we find encouraging. Specifically, reviewers appreciated the generalization of prior methods into a unified framework, the practical utility of our TopoTune toolkit, and the extensive experiments showing competitive performance over previous TDL methods.

All reviewers identified similar areas for improvement, particularly regarding runtimes and complexity, as well as better clarifying and contextualizing our contribution. Here, we provide answers to these major comments shared by reviewers. We additionally upload the pdf of our revised paper and Appendix where we have directly addressed all comments (in blue in the text). A response to each individual reviewer’s comments is provided in the thread of the associated review.

**1. Complexity and training times**
We agree with reviewers TxXe and iHAm that complexity and training times are important features of any deep learning model. We thank both reviewers for catching this, as we believe that these additions strengthen our argument.

- **A. Time complexity:** We now provide a time complexity analysis in Appendix C (referred to in main text at lines 475-478). This analysis shows that GCCNs achieve a compromise between time complexities of standard GNNs and of CCNNs. This is specifically due to GCCNs’ accommodation of lightweight per-rank neighborhood structures.

- **B. Runtimes:** In our revised manuscript, we now report both model sizes and training times (Appendix G) on all experiments presented in Section 4, and briefly discuss them in Section 6.2 (paragraph GCCNs are smaller than CCNNs). We find that for smaller datasets (ex: MUTAG, Cora, Citeseer), GCCNs already train comparably or faster than CCNNs. For larger datasets however (ex: ZINC, PubMed) GCCNs train slower than CCNNs, but this is only due to an artifact of our implementation, which performs the on-the-fly graph expansion process before each forward pass of the model. We will move this expansion process into the preprocessing step, computing it only once and significantly speeding up the forward pass (see time complexity analysis).

**2. Clarifying specific advantages of GCCNs**

- **A. Analysis of performance versus size:** Reviewers TxXe and iHAm proposed to clarify the analysis of performance versus size provided in Figure 5. We edited the text to better contextualize this empirical observation. We also point to Fig. 8 in Appendix H which provides the same plots for all datasets. We respond directly to reviewers’ questions in the individual replies.

- **B. Contextualizing the contribution:** Reviewers rJiq and kr8N asked for better contextualization of the research questions addressed by our work, and its usefulness in the context of TDL. We now clearly specify in our stated contributions the open problems of TDL that we address (see next reply), leveraging a recent position paper written by many of the field’s leaders. We also better describe current gaps in the field, and how our work builds upon previous GNN simulation works by respecting topological symmetry. We emphasize this contribution of respecting conserving topological symmetry in responding to Reviewer iHAm about previous works in higher-order GNNs. We also emphasize how TopoTune goes beyond a hyperparameter search tool by unlocking a new class of models (ex: per-rank neighborhoods, message functions from the GNN world) that we empirically validate to be helpful without an extensive training hyperparameter search. Specifically, hyperparameters like learning rate, batch size, embedded dimensions, readout function, and more are kept largely fixed. This is now specified at lines 420-423. (Continues in the next comment)

---

> ### Author Response · Authors · 2024-11-20
> **Response to all reviewers (continued)**
>
> **2.B. Contextualizing the contribution (continued)**
> Below, we outline the 7 open problems of TDL we partially or fully address in this work. These problems were defined by many of the field’s leading authors in a recent position paper (https://arxiv.org/pdf/2402.08871).
>
> - Open problem 1: Need for adaptation of TDL in practice and making TDL more accessible for relation learning in real-world applications. TopoTune removes much of the software-based barrier to entry and makes the choice of base model flexible. Hence, TopoTune makes TDL a more attractive choice for practitioners from the several fields that stand to benefit from relational learning, such as neuroscience, protein engineering, chip design, and so on (see https://arxiv.org/abs/2106.04051 at Appendix B).
>
> - Open problem 3: Need for benchmarking. As TDL evolves quickly across an increasingly wide landscape of models, it becomes challenging to understand which model is best suited for a particular application. Through its integration in TopoBenchmark and its structure, TopoTune makes basic components of TDL such as choice of topological domain, neighborhood structure, and message-passing function easily comparable across many benchmarking tasks. This is a fundamental change of perspective compared to previous literature, which always considered one new model with one set of such choices at a time.
>
> - Open problem 4: Limited availability of software. On one hand, TopoTune directly integrates with the fully open-source TopoBenchmark platform, which itself leverages other open source TDL tools from packages like TopoModelX and TopoNetX. On the other hand, TopoTune makes the much wider and more well-established platform of software tools for GNNs (such as PyTorch Geometric) directly applicable to TDL.
>
> - Open problem 5: Need for a cost-benefit analysis framework. By exploring a large model space systematically, TopoTune makes it much easier to assess cost (model size, training time) in comparison to performance. We show an example of such an analysis in Figure 5.
>
> - Open problem 6: Building scalable TDL models that work across domains. To our knowledge, GCCNs are the first models empirically tested across many higher-order domains. This feature is enabled by baking the implementation into TopoBenchmark, which already contains the data liftings for many topological domains.
>
> - Open problem 9: Consolidating relative advantages of TDL through theoretical foundations. Beyond proving that GCCNs generalize and subsume CCNNs, we also provide expressivity and permutation equivariance proofs that are cross-domain applicable. To our knowledge, the WL-tests and related definitions of expressivity that we define on combinatorial complexes (in the updated manuscript) are the first definitions existing on combinatorial complexes. Further, we leverage these definitions to prove that GCCNs are strictly more expressive than CCNNs (in the updated manuscript), which is  the first proof of expressivity for combinatorial complexes.
>
> - Open problem 11. Developing a transformer architecture for TDL models that “lays a unified foundation for TDL across different higher-order domains.” As presented in Table 1, TopoTune offers the possibility of using a vanilla transformer as a “fully-connected” message-passing function. However, TopoTune offers a more general answer to this problem, in that it provides a foundation, both theoretical and practical, for building models across topological domains.

---

### Meta-Review · Area_Chair_tDzK · 2024-12-20

**Metareview:**

This paper proposes a new framework for topological deep learning that addresses several open problems in a recent position paper.
All reviewers agree this is a very valuable contribution, and the paper is well-structured and well-written.
However, during the discussion period, one of the reviewers pointed out some technical issues with definitions and derivations. These technical issues were addressed by the authors but required significant changes to the manuscript. Any substantial revision should be resubmitted for a fresh assessment by the reviewers, as it’s unlikely that most of them had time to review the new material.
This is why, after a conversation with the senior area chair, despite the good scores, I recommend rejecting it so a new set of reviewers can assess the paper in its current form.

**Additional Comments On Reviewer Discussion:**

There is a long discussion between the authors and reviewer kr8N. I believe this discussion helped the authors improve the paper significantly.

In the end, the reviewer concluded by saying that some claims in the paper are somewhat overemphasized, and some arguments and responses during the rebuttal were slightly deceptive.

As I said above, due to the changes in the paper during the rebuttal period and the fact that this reviewer was very engaged in the discussion and wasn't happy with the final result, I would suggest to reject.

---

### Decision · Program_Chairs · 2025-01-22

Reject